



# Millennial-scale atmospheric CO₂ variations during the Marine Isotope Stage 6 period (190─135 kyr BP)

Jinhwa Shin[1], Christoph Nehrbass-Ahles[2,4], Roberto Grilli[1], Jai Chowdhry Beeman[1], Frédéric Parrenin[1], Grégory Teste[1], Amaelle Landais[3], Loïc Schmidely[2], Jochen Schmitt[2], Thomas F. Stocker[2], Hubertus Fischer[2] and Jérôme Chappellaz[1]

[1]CNRS, Univ. Grenoble-Alpes, Institut des Géosciences de l'Environnement (IGE), Grenoble, France
[2]Climate and Environmental Physics, Physics Institute, & Oeschger Centre for Climate Change Research, University of Bern, Switzerland
[3]Laboratoire des Sciences du Climat et de l'Environnement, LSCE/IPSL, CEA-CNRS-UVSQ, Université Paris-Saclay, 91191, Gif-sur-Yvette, France
[4]Godwin Laboratory for Palaeoclimate Research, Department of Earth Sciences, University of Cambridge, Cambridge,UK

*Correspondence to*: Jérôme Chappellaz (jerome.chappellaz@univ-grenoble-alpes.fr)

**Abstract.** Understanding natural carbon cycle / climate feedbacks on various time scales is highly relevant to reliably predict future climate changes. During the last two glacial periods, climate variations on millennial time scales were observed but the background conditions and duration of climate variations are different. Here we make use of contrasting climatic boundary conditions during the last two glacial periods to gain insight into the co-occurring carbon cycle changes. We reconstruct a new high-resolution record of atmospheric $CO_2$ from the EPICA Dome C (EDC) ice core during Marine Isotope Stage (MIS) 6 (190 to 135 kyr BP). During long stadials in the North Atlantic (NA) region, atmospheric $CO_2$ appears to be associated with the coeval Antarctic temperature changes at millennial time scale connected to the bipolar seesaw process. However, during one short stadial in the NA, atmospheric $CO_2$ variation is negligible and the relationship between temperature variation in EDC and atmospheric $CO_2$ is unclear. We suggest that the amplitude of $CO_2$ variation may be affected by the duration of perturbations of the Atlantic Meridional Overturning Circulation (AMOC). In addition, similar to the last glacial period, in the earliest MIS 6 (MIS 6e and 6d, corresponding to 189 to 169 kyr BP), Carbon Dioxide Maxima (CDM) show different lags with respect to the corresponding abrupt $CH_4$ jumps, the latter reflecting rapid warming in the Northern Hemisphere (NH). During MIS 6e at around 181.5±0.3 kyr BP, CDM 6e.2 lags abrupt warming in the NH by only 200±360 yrs. During MIS 6d which corresponds to CDM 6d.1 (171.1±0.2 kyr BP) and CDM 6d.2 (175.4±0.4 kyr BP), the lag is much longer, i.e., 1,400±375 yrs on average. The timing of $CO_2$ variations with respect to abrupt warming in the NH may be affected by a major change in the organization of the AMOC from MIS 6e to MIS 6d.



# 1 Introduction

Ice core studies allow us to considerably extend our knowledge about natural climate-carbon cycle feedbacks by directly reconstructing atmospheric $CO_2$ from gas preserved in Antarctic ice sheets (Lüthi et al., 2008; Petit et al., 1999). Comparing atmospheric $CO_2$ records from Antarctic ice cores with proxies of paleoclimate helps us to understand how atmospheric $CO_2$

was controlled by carbon exchange with the ocean and land reservoirs on orbital to centennial time scales (Ahn and Brook, 2008, 2014; Bereiter et al., 2012; Higgins et al., 2015; Lüthi et al., 2008; Marcott et al., 2014; Petit et al., 1999).

Previous works on polar ice core records revealed that temperature variations in Greenland and Antarctica on millennial time scales appear to be a pervasive feature during the last glacial period. While Antarctic temperature varied gradually, Greenland temperature changes occurred abruptly. A phase difference can be observed between millennial-scale variations of temperature

in the NH and SH (Northern and Southern hemisphere, respectively), which is referred to as the bipolar seesaw phenomenon (Broecker, 1998; Stocker and Johnsen, 2003). Potential triggers for this climatic variability on the millennial scale are fresh water inputs into the North Atlantic (NA) or alterations of sea ice extent, surface temperature and salinity in NA (Bond et al., 1992; Broecker et al., 1992; Heinrich, 1988; McManus et al., 1998), which may reduce the strength of the Atlantic Meridional Overturning Circulation (AMOC). This would cause a reduction in heat transport from the SH to the NH, which leads to an

abrupt cooling in the NH and a gradual warming in the SH (Stocker and Johnsen, 2003) and the oppoiste behaviour when AMOC is strengthened.

In response to the millennial temperature perturbations, existing $CO_2$ records show the presence of millennial-scale oscillations on the order of 20 ppm over the last glacial period (Ahn and Brook, 2008, 2014; Bereiter et al., 2012), which generally co-vary with the major water isotope (δD) variations in Antarctic ice cores reflecting Antarctic temperature variations (Ahn and

Brook, 2008; Bereiter et al., 2012) (Figure 1). During cold periods in the NA, referred to as NA stadials, atmospheric $CO_2$ increased continuously and in parallel to Antarctic temperature increase. Again, at the onset of warming in Greenland, atmospheric $CO_2$ started to decrease (Ahn and Brook, 2008; Bereiter et al., 2012), generally in line with a co-occurring, slow Antarctic cooling. However, the $CO_2$ decrease did not always start at exactly the same time as the onset of the DO warming, and the lag itself varied. For example, during Marine Isotope Stage (MIS) 3, atmospheric $CO_2$ maxima lagged behind abrupt

temperature change in Greenland by 870±90 yrs. During MIS 5, the lag of atmospheric $CO_2$ maxima with respect to abrupt temperature warming in the NH was only about 250±190 yrs (Bereiter et al., 2012). Atmospheric $CO_2$ variations on millennial scales are thought to be related to the role of the Southern Ocean in carbon uptake and deep ocean ventilation on millennial timescales (Fischer et al., 2010; Marcott et al., 2014; Sigman and Boyle, 2000; Toggweiler et al., 2006). In addition, atmospheric $CO_2$ can be affected by changes in the AMOC, which affects the ventilation of carbon from the deep ocean

(Denton et al., 2010; Sigman et al., 2007). However, the mechanisms responsible for these oscillations are still under debate, and therefore require further studies.

Comparing $CO_2$ changes on millennial time scales during the past two glacial periods, MIS 3 (MIS 3, 60–27 kyr BP) and early MIS 6 (early MIS 6, 185–160 kyr BP) can provide us with a better understanding of the carbon mechanisms at work, due to

the similarities but also differences of the climate changes during the last two glacial periods (see Figure S1 in SI (Supplement Information)). The bipolar see-saw mechanism has been observed to be active also during the early MIS 6 period (Cheng et al., 2016; Jouzel et al., 2007; Margari et al., 2010), however, climate boundary conditions during MIS 6 were slightly different from those during MIS 3. For example, events of iceberg discharge into the NA, which impact oceanic circulation, appear to

be much more frequent during MIS 3 than during MIS 6 (Margari et al., 2010; Margari et al., 2014). During the time period around 175 kyr BP, iceberg discharge was muted; the intertropical convergence zone (ITCZ) is thought to have shifted northward, intensifying monsoon systems in low latitude regions, such as in Asia, the Appenine Peninsula and the Levant (Ayalon et al., 2002; Bard et al., 2002; Cheng et al., 2016), and leading to a generally weaker AMOC (Margari et al., 2010). The duration of NA stadials during the early MIS 6 is longer than that during MIS 3. In addition, the AMOC cell during MIS

6 might have been shallower than that during MIS 3 (Margari et al., 2010), which may have been caused by intensified hydrological cycles in low latitude regions (Ayalon et al., 2002; Bard et al., 2002; Cheng et al., 2016).

In order to investigate whether the different boundary conditions between MIS 6 and MIS 3 could have impacted the relationship between atmospheric $CO_2$ and climate, we reconstructed atmospheric $CO_2$ concentrations from the EPICA Dome C (EDC) ice core (75°06'S, 123°24'E) with 150 new data points spanning MIS 6 (190 to 135 kyr BP), significantly improving

existing records previously obtained from the Vostok ice core (Petit et al., 1999). With our new reconstruction, the average temporal resolution during MIS 6 reaches 360 yr as compared to 1,000 yr in the Vostok record. Over the early time period of MIS 6, when millennial-scale variability is observed in the Antarctic climate record (185–160 kyr BP), a total of 100 new $CO_2$ measurements provide a temporal resolution of ~280 yr. We also improved the relative age uncertainties between ice and gas in the EDC core using $\delta^{15}N$-based estimates of firn column thickness to better constrain leads and lags between $\delta D$ composition

in EDC and atmospheric $CO_2$ concentrations during the early MIS 6, and we established our new chronology using new $\delta^{15}N$ data during the early MIS 6 in this study and published data from Landais et al. (2013). Finally, we improved the temporal resolution of existing $CH_4$ data from EDC (Loulergue et al., 2008) from ~600 yr to ~350 yr to calculate the shift of Carbon Dioxide Maxima (CDM) relative to the rapid climate change in the NH. To avoid the age uncertainties between proxy data and atmospheric $CO_2$ data, $CH_4$ measurements were used as a time marker of rapid warming in the NH. Over the last glacial

period, $CH_4$ and Greenland temperature were found to be essentially synchronous with a mean lag of $CH_4$ of not more than about 50 years (Baumgartner et al., 2014; Rosen et al., 2014).

## 2 Methods

### 2.1 CO₂ measurements

Atmospheric $CO_2$ was reconstructed using the Ball Mill dry-extraction system coupled to a gas chromatograph at the Institut

des Géosciences de l'Environnement (IGE), France (Schaefer et al., 2011). Each data point presented in this study corresponds to a single 40 g ice sample which was measured five times by gas chromatography (five consecutive injections of the same



extracted gas). Approximately 5 mm of ice were trimmed from the ice core surfaces before extraction in order to remove the external part that could be potentially contaminated with drilling fluid or might have been subject to gas loss during storage in the freezer (Bereiter et al., 2009). The $CO_2$ measurements were referenced to a secondary gas standard (synthetic air from Air Liquide (Alphagaz 28416000)) containing 233.7±0.4 ppm of $CO_2$ in dry air, which was referenced to two primary standards

(238.34±0.04 from NOAA (CB09707) and 260.26±0.2 from CSIRO (CSIRO1677)).

Blank tests using 40 g of artificial bubble-free ice were conducted every 10 measurements to quantify the precision of the system and to correct for the $CO_2$ contamination caused by the crushing process. Blank tests are conducted in two steps: First, to validate the baseline of the system, a gas standard with 233.7±0.4 ppm is injected over the bubble-free ice in the cell. The gas is then left equilibrating in the cell for 10 minutes. Then the gas is analysed twice by successive injections into the extraction

line and sample loop. Afterwards, the bubble-free ice is crushed and the gas is analysed 5 more times. The difference between the results before and after crushing was considered as the contamination effect caused by the crushing process. These values were used to estimate the precision of the system. Measured $CO_2$ should be corrected for contamination caused by the analytical procedure by comparing measured $CO_2$ in blank tests with the standard gas value. However, it is not feasible to correct $CO_2$ concentrations directly. The $CO_2$ mixing ratio is calculated as the ratio between partial pressure of $CO_2$ and total pressure in

the measurement line, which is a relative value. Thus, $CO_2$ concentration, and the concentration of $CO_2$ contamination, are dependent on the total pressure. To avoid this dependence, we correct the absolute value (partial pressure) of $CO_2$ in the air by the expected partial pressure of $CO_2$ contamination, as estimated from blank tests.

For this study, we used 4 different chambers to hold the ice core samples during crushing and measurement. Each chamber showed different contamination levels. Therefore, blank tests are conducted on each chamber. The data was corrected by the

average of each chamber. For these measurements, a precision of the system of at ~1 ppm on average was obtained. On average, the blank correction corresponds to a reduction of the measured $CO_2$ concentration by 1.7±1.0 ppm (1σ). In addition, the $CO_2$ record was corrected for gravitational fractionation, using the $\delta^{15}N$ isotope ratio (Craig et al., 1988). To this end, 88 new data points together with existing $\delta^{15}N$ measurements (Landais et al., 2013) covering the late MIS 6 (156.4–139.2 kyr) were used. $\delta^{15}N$ data were linearly interpolated in age to each corresponding $CO_2$ data point. On average, the correction corresponds to

removing 1.2±0.1 ppm from the measured $CO_2$ concentration. Thus, in total, an average correction of 2.9±1.0 ppm was applied to the raw $CO_2$ data. 150 individual ice samples were measured for $CO_2$ in the depth range from 2036.7 to 1787.5 m along the EDC ice core, corresponding to the time period from 189.4 to 135.4 kyr BP on the AICC2012 chronology (Bazin et al., 2013), thus leading to a mean temporal resolution of 360 yr (280 yr over the time period 185–160 kyr BP).

## 2.2 CH₄ measurements

We measured the atmospheric $CH_4$ content of 63 ice core samples, using the wet extraction method at IGE described in detail in Spahni et al. (2005). This allowed us to improve the temporal resolution of existing $CH_4$ data (Loulergue et al., 2008) from ~600 yrs to 360 yrs on the AICC2012 chronology (see Figure S2 in SI).



The previous $CH_4$ dataset (Loulergue et al., 2008) from EDC was produced at both IGE and Climate and Environmental Physics (CEP), Physics Institute, University of Bern, Switzerland. A systematic offset of 6 ppb between IGE and CEP was observed (Loulergue et al., 2008). A new data correction was applied to data measured at IGE in this study, and the systematic offset between the existing (corrected) data points and their nearest neighbours in the new data set is 3.0±4.6 ppb (n=11). In addition, the mean difference between the new data and the existing data points (Loulergue et al., 2008) is 1.7±2.4 ppb (n=63, the standard error of the mean) during MIS 6, which is within data error. Thus, in this study, we did not take this systematic offset between the $CH_4$ records into account or revise the Bern/IGE offset correction of the previous data.

### 2.3 Nitrogen isotopes

Isotopes of molecular nitrogen in air bubbles were measured by a melting technique at the Laboratoire des Sciences du Climat et de l'Environnement (LSCE), France. The gas was extracted from the ice by a wet extraction technique and the released air was analysed by a dual-inlet mass spectrometer (Delta V Plus; Thermo Scientific). The analytical method and data correction are described in detail in Bréant et al. (2019). In total, 151 samples from 88 depth intervals (63 duplicates) between the depths of 2124.7 and 1875.0 m below the surface were measured, corresponding to 205 to 154 kyr BP (Figure S3 in SI). The average resolution on the AICC2012 chronology is ~580 yrs.

### 2.4 Ice age revision by estimating $\Delta$depth from $\delta^{15}N$ in air of ice core

The water isotopic signature ($\delta D$), is, unlike $CO_2$, measured on the ice matrix. Air enclosed in an ice core moves through the porous firn layers at the top of the ice sheet by molecular diffusion, and becomes trapped in the ice at the so-called Lock-In Depth (LID, around 100 m below the surface in the case of the EDC ice core). An age difference thus exists between the ice and the air at a given depth. For the conditions of the EDC ice core, the age difference can reach 5 kyr and is associated with a large uncertainty (several hundred years (Loulergue et al., 2007)). $\delta^{15}N$ of molecular nitrogen in air bubbles can be used to determine the LID and to calculate the depth difference between synchronous events in the ice matrix and air bubbles, called delta depth ($\Delta$depth), thus creating a more precise relative chronology of the gas- with respect to the ice-phase (Parrenin et al., 2013). We use the $\Delta$depth calculation to adjust the gas chronology of EDC, while the ice chronology from AICC2012 remained unchanged.

We calculate the height of the firn column, $h$, from $\delta^{15}N$ of $N_2$ measurements (Craig et al., 1988; Dreyfus et al., 2010; Sowers et al., 1989), by using the following Eq.:

$$h = h_{\text{conv}} + (\delta^{15}N - \Omega (T) \Delta T_{\text{diff}}) \left(\frac{\Delta m \cdot g \cdot 1000}{RT}\right)^{-1}, \qquad (1)$$

In this equation, $\Delta T_{\text{diff}}$ is the temperature difference between the top and the bottom of the diffusive zone as estimated by the Goujon/Arnaud model, where surface temperature and accumulation are estimated from the stable water isotope record (Loulergue et al., 2007). $\Omega (T)$ is the thermal diffusion sensitivity, which has been estimated from laboratory measurements by Grachev and Severinghaus (2003). $\Delta m$ is the mass difference between $^{14}N$ and $^{15}N$ (kg mol$^{-1}$), g is the gravitational



acceleration (9.825 m s$^{-2}$ for Antarctica) (Parrenin et al., 2013), and R is the universal gas constant (8.314 J mol$^{-1}$K$^{-1}$). Finally, $h_{\text{conv}}$ is the height of the convective zone at the top of the firn column, which is considered negligible at EDC according to current observations (Landais et al., 2006). The variation of the convective height is related to changes in wind stress. According to Krinner et al. (2000), wind on the East Antarctic plateau varied little during the LGM (last glacial maximum),

which is analogous to late MIS 6, and the convective zone was confirmed to be very thin during the last deglaciation by Parrenin et al. (2012). Thus, we assume that $h_{\text{conv}}$ is negligible during MIS 6. Δdepth is calculated from the height of the air column using a constant firn average density and a modelled vertical thinning function suggested by Parrenin et al. (2013).

Raw $\delta^{15}$N data cannot be used directly to calculate Δdepth, because bubbles are not trapped at exactly the same time for a given depth in the ice sheet, leading to cm-scale age scale inversions (Fourteau et al., 2017). Although the layer inversions

should not strongly affect our record as its resolution is larger than the scale of inversion events (Fourteau et al., 2017), the change in Δdepth between two different depths (z) in the ice core, denoted $\partial\Delta\text{depth}/\partial z$, deduced from the raw $\delta^{15}$N data, shows values higher than 1, that could correspond to stratigraphic inversions in the gas phase (Figure 2). Therefore, a 3-point moving average weighted by the time difference between a point and its two neighbours was applied to the $\delta^{15}$N dataset. The weights for the three points are equal if the time difference is less than or equal to 500 years, which is close to the average sampling

resolution of the $\delta^{15}$N dataset. Otherwise, the neighbouring data points are weighted by 500/ΔT, where ΔT is the time difference. This avoids assigning too much weight to neighbouring points where the resolution in the record is lower, which would lead to local over-smoothing.

Δdepth as estimated using the 3-point moving average weighted by the time difference is shown in Figure 2. The difference of the Lock-In Depth in Ice Equivalent (LIDIE) calculated in AICC2012 (Bazin et al., 2013) and deduced from $\delta^{15}$N in this study

is 0.5±3.0 m, or 0.7±5.6 % on average (see Figure S4 in SI). The AICC2012 LIDIE was calculated using a background scenario derived from δD, using a linear relationship between δD and $\delta^{15}$N. Our results show this relationship to be relatively unbiased but not entirely exact (Figure 2). The Δdepth values are used to update the EDC gas chronology from the original AICC2012 one, while the ice chronology remains unchanged. For MIS 6, this method significantly reduces the relative age uncertainty between air and ice to 900 yrs on average with respect to the original AICC2012 chronology (see Figure S5 in SI).

**2.5 Age scale of the MD01–2444 marine sediment record**

The MD01–2444 Marine sediment core from the Iberian margin (Margari et al., 2010) is an important proxy for the interpretation of our $CO_2$ record, as it is well–resolved during MIS 6, and it provides high-resolution benthic and planktonic foraminiferal records. In order to make optimal use of the record, its relative chronology with respect to EDC requires additional tuning. The original MD01–2444 age scale was built by matching the benthic $\delta^{18}$O record (Shackleton et al., 2000)

to the δD record from EDC on the EDC03 ice age scale (Parrenin et al., 2007) and using 11 tie points selected by Margari et al. (2010) as shown in Table 1. The age of the sediment record was linearly interpolated between the tie points, and the age of the tie points recalculated using the AICC2012 chronology.





## 2.6 Definition of NA stadial duration

Due to the absence of a Greenland temperature record for MIS 6, the durations of the six NA stadials identified during the MIS 6 period were previously defined by Margari et al. (2010) using the interval between the stadial transitions. Between the maximum and the preceding minimum of δD in the EDC record is defined as the stadial transitions. However, this approach relies on the assumption that the bipolar seesaw was present during MIS 6. Here we instead use another approach which was also proposed by Margari et al. (2010).

The durations of the six NA stadials were defined using $\delta^{18}O$ of planktonic foraminifera and tree pollen in MD01–2444, which reflect temperature variability in the NH (Margari et al., 2010). The interval between the maximum and the preceding minimum of both $\delta^{18}O$ of planktonic foraminifera and tree pollen in MD01–2444 were defined as the inflection points. The time interval between two inflection points were defined as the NA stadial duration. In this approach, small variations of the two records may bias the calculation of the duration of short stadials in the NH. However, the average age difference between the durations identified using the two methods is only 205 years, which is less than the sampling resolution of MD01–2444 during MIS 6. The stadials identified for MIS 6 are shown in Figure 3 and Table 2. The uncertainty of the duration of each stadial was estimated as half of the temporal difference between maxima and minima of $\delta^{18}O$ of planktonic foraminifera.

## 3 Results

### 3.1 The new high-resolution and high precision CO₂ record during MIS 6

Figure 4 shows the new atmospheric $CO_2$ record from EDC during MIS 6, compared to the existing $CO_2$ data from Vostok (Petit et al., 1999). Both were measured using the ball mill extraction system at IGE, and both are reported on the AICC2012 air age scale (Bazin et al., 2013). The error bars indicate the standard deviation of five consecutive injections of the gas extracted from each sample into the gas chromatography added to the precision of the measurement estimated by the reproducibility of the blank tests (~0.8 ppm) using a quadratic sum. The average standard deviation is ~1.1 ppm for EDC (this study) and ~3 ppm for the Vostok dataset (Petit et al., 1999).

The lowest atmospheric $CO_2$ concentration observed in our new MIS 6 dataset is 179.5±0.9 ppm at approximately 156.3 ±0.3 kyr BP, and the highest concentration is 211.7±0.3 ppm at 181.3±0.2 kyr BP (Figure 4). Atmospheric $CO_2$ variability in the EDC ice core shows similar general patterns as the one retrieved from the Vostok ice core, although new features appear thanks to the improved temporal resolution. However, $CO_2$ concentrations from Vostok appear systematically higher than those from EDC by 4.0 ± 5.6 ppm on average and the difference grows to more than 10 ppm at the beginning of Termination 2 at around 135 kyr BP. Two main arguments may explain this offset: contamination due to the extraction system was assumed as being negligible at the time of the Vostok sample measurements at IGE (Petit et al., 1999), while from the measurements conducted during this study it possibly amounts on average to an additional reduction of ~1.7 ppm. Some of the differences in the two records might also be related to age scale uncertainties (Figure 4), due to the limited number of stratigraphic tie points between



the two cores (Bazin et al., 2013). Lastly, the atmospheric $CO_2$ measurements from the Vostok ice core are affected by a larger analytical uncertainty than the EDC ones. Here we constrain our interpretation to millennial changes in our new $CO_2$ record, and our conclusions are largely independent of the absolute $CO_2$ level.

### 3.2 Relationship between EDC δD and atmospheric $CO_2$

Margari et al. (2010) suggested that MIS 6 can be divided into three sections depending on the degree of climatic variability observed in δD (indicative of Antarctic climate variability) and $CH_4$ (which may reflect NA climate variability) in EDC : early (185.2−157.7 kyr BP), transition (157.7−151 kyr BP) and late MIS 6 (151−135 kyr BP) (Figure 5 and Figure 6). Climatic oscillations on millennial time scales are pervasive during the early MIS 6 period (185−160 kyr BP) (Barker et al., 2011; Cheng et al., 2016; Jouzel et al., 2007; Margari et al., 2010; Margari et al., 2014), which is similar to MIS 3 (Figure 1, 5 and

Figure S11 in SI). However, during the late MIS 6 period / penultimate glacial maximum (PGM), millennial variability is subdued, and resembles climate variability on millennial time scales during MIS 2 (Figure 1 and Figure 6). During the transitional period from 157-151 kyr BP, δD in EDC slowly increased (Jouzel et al., 2007). Like δD in EDC, $CO_2$ variations on millennial time scales are pervasive during the early MIS 6 period (185−157.7 kyr BP). During the transitional period from 157–151 kyr BP, atmospheric $CO_2$ increased slowly, while, during the late MIS 6 period, $CO_2$ variation is subdued.

Between these two periods, we can identify one low-amplitude $CO_2$ peak at around 150 kyr BP, which can be another potential CDM. This atmospheric $CO_2$ variation is of triangular shape, which follows the δD pattern. The change of direction is also associated with a $CH_4$ peak. This variation is similar to MIS 4 and MIS 10 (Barker et al., 2011; Nehrbass-Ahles et al., in review).

For the full MIS 6 interval, the new EDC $CO_2$ record and δD in EDC are synchronous within age uncertainty (160±900 yrs
using a maximum lag correlation coefficient estimate, and taking into account the relative age uncertainty between ice and air).

### 3.3 Atmospheric $CO_2$ variability on millennial time scale

While there was indication of millennial $CO_2$ variability in the Vostok record, we now present clear evidence of millennial variability of $CO_2$ concentrations during MIS 6 that are associated with AIM events, thanks to the improved time resolution
and precision of the obtained $CO_2$ data (Figure 5). To better discuss millennial variability in MIS 6, a Savitsky Golay filter with a 500 yr cut−off period was selected to filter out centennial−scale variability and noise (see Figure 5, Figure S10 and Table S1 in SI). Five prominent and one subdued $CO_2$ variations were detected in atmospheric $CO_2$ during early MIS 6 (Figure 5). The five prominent peaks are observed at 160.6±0.3 (CDM 6c.1), 164.2±0.3 (CDM 6c.2), 169.6±0.2 (CDM 6d.1), 174.1±0.2 (CDM 6d.2) and 181.3±0.2 (CDM 6e.2) kyr BP (the errors given here reflect the uncertainty with respect to the
position of each maximum, and do not include the age uncertainty of the ice core, in each case around 3.0 kyr, 1 sigma). Each CDM is associated with an Antarctic Isotope Maximum (AIM) event. The short AIM 6e.1 event corresponds to CDM 6e.1 at around 178 kyr BP (with an age uncertainty of 3.0 kyr, 1 sigma), whose amplitude was estimated by a single data point of





atmospheric $CO_2$ maximum because $CO_2$ variations becomes even less pronounced after filtering. CDM 6e.1 has an amplitude of only ~4 ppm. If this point is excluded, atmospheric $CO_2$ and $\delta D$ composition in EDC appear completely decoupled, though it is likely that the amplitude of the CDM would have been reduced by smoothing during the gas trapping process (Fourteau et al., 2017). This conclusion seems confirmed when considering the relationship between atmospheric $CO_2$ change and the

duration of NA stadials calculated using tree pollen and the $\delta^{18}O$ composition of planktic foraminifera in a Iberian margin core (Margari et al., 2010) for MIS 6 (Figure 3 and Table 2), and using isotopic records from Greenland ice cores (Rasmussen et al., 2014) for MIS 3 (further details can be found in the SI), as shown in Figure 7. The magnitude of atmospheric $CO_2$ change is generally correlated with the NA stadial duration (r=0.7, n=6) during the early MIS 6 period.

We note a similar correlation between the NA stadial duration and atmospheric $CO_2$ change during MIS 3 (r=0.85, n=14).

When the NA stadial duration was shorter than 1,500 yrs, atmospheric $CO_2$ varied less than 5 ppm (Ahn and Brook, 2014; Bereiter et al., 2012) as is the case for CDM 6e.1. Margari et al. (2010) note one exception, AIM 14 during which ice discharge may have led to a stronger perturbation to AMOC.

Both Bereiter et al. (2012) and Ahn and Brook (2014) observe that during short NA stadials which last less than 1,500 yrs, the $CO_2$ maxima do not appear to have a consistent phase relationship with AIMs and $CO_2$ and $\delta D$ anomalies are not correlated

(Ahn and Brook, 2014). On the other hand, in both MIS 3 and 6 periods, $CO_2$ is highly correlated with $\delta D_{ice}$ anomalies during the long stadials (r = 0.84 on average) with a stronger increase in $CO_2$ (Ahn and Brook, 2014; Bereiter et al., 2012).

We observe that during the last two glacial periods, the amplitude of $CO_2$ is highly determined by the NA stadial duration (r=0.83, n=20). This observation implies that despite the different climate boundary conditions during the last two glacials, the behavior of atmospheric $CO_2$ was similar (see Figure S11 in SI).

**3.4 Leads and lags between $CO_2$ and the abrupt warming in NH**

To better understand the link between the bipolar see–saw mechanism and atmospheric $CO_2$ variability on millennial time scales, we calculated the varying time lag for each CDM following abrupt warming events in the NH (see Figure S6–S9 in SI) (Bereiter et al., 2012). Due to the lack of a direct temperature proxy record in Greenland covering MIS 6, and due to the fact that the $\delta^{18}O$ composition of planktic foraminifera in NA sediment cores, which could be used in principle as an indicator

for abrupt warming in the NH (Shackleton et al., 2000), cannot be placed with sufficient precision on a common chronology with the EDC ice core, in this work $CH_4$ measurements performed on the EDC ice core were used as a time marker of rapid warming in the NH (Baumgartner et al., 2014; Brook et al., 1996; Huber et al., 2006). Because $CH_4$ and $CO_2$ signals are both imprinted in the air bubbles, there is no chronological uncertainty when comparing the timing of changes of those two signals. The only remaining uncertainty is related to analytical uncertainties and to the temporal resolution of the two records. We pick

intervals when $CH_4$ increases rapidly by at least 50 ppb over a time period of less than 1 kyr that correspond to Antarctic isotope maxima (Buizert et al., 2015; Loulergue et al., 2008). The timing of abrupt $CH_4$ increases was defined as the midpoint between the beginning of the increase of $CH_4$ and its maximum. The age uncertainty of the midpoint is defined by the time difference between the midpoint and either of the two endpoints.



Figure 8 shows the shifts of CDM with respect to the onset of the abrupt warming in the NH. During the MIS 6 period, three abrupt NH warmings (as inferred from the $CH_4$ signal) at 181.5±0.3, 175.4±0.40 and 171.1±0.2 kyr BP (2σ) were found. These events correspond to CDM 6e.2, CDM 6d.2 and CDM 6d.1, respectively. CDM 6c.1, CDM 6c.2 and CDM 6e.1 do not have corresponding rapid changes in the methane record; this may be due to slow gas trapping as compared to interglacial periods,

which could completely smooth out smaller changes. A synthetic Greenland temperature record (Barker et al., 2011) shows abrupt temperature jumps at CDM 6c.1, CDM 6c.2 and CDM 6e.1 as well. However, this record is calculated using EDC $\delta D_{ice}$, and the large chronological uncertainty (~900 yrs on average) between ice and gas phases does not allow us to make any conclusions about leads and lags using this record, since these tend to be on the order of 1 kyr. We therefore exclude these events.

From MIS 6e to MIS 6d, the lag of $CO_2$ with respect to abrupt warmings in the NH, which were identified from this chronological comparison between EDC $CH_4$ and $CO_2$, becomes larger. During the earliest MIS 6, atmospheric $CO_2$ increases rapidly (by ~4.2 ppm in 200±360 yrs) following the abrupt $CH_4$ increase at 181.5±0.3 kyr BP. The peak of CDM 6e.2 is nearly synchronous with the onset of the NH abrupt warming (non- significant lag of 200±360 yrs, Figure 8). During MIS 6d which corresponds to CDM 6d.1 and 6d.2, $CO_2$ concentrations show a much slower increase over a duration of ~3.3 kyr. Here, $CO_2$

lags behind the onset of the NH abrupt warming by 1,500±280 yrs and 1,300±450 yrs, respectively (1,400± 375 yrs on average, with the error calculated by propagation of the uncertainties in the individual events). Interestingly, these two CDM events occurred during MIS 6d, when iceberg discharge was muted and the intertropical convergence zone (ITCZ) is thought to have shifted northward, intensifying monsoon systems in low latitude regions, such as in Asia, the Appenine Peninsula and the Levant (Ayalon et al., 2002; Bard et al., 2002; Cheng et al., 2016), and leading to a generally weaker AMOC (Margari et al.,

2010). Therefore, the two apparent $CO_2$ lag timescales with respect to abrupt warming in NH during MIS 6 might be explained by this difference in background climate conditions.

Indeed, these features during MIS 6 appear fully compatible with those observed during the last glacial period. To compare $CO_2$ variations with respect to abrupt warming in NH during the last two glacial periods, in this study we also re-estimated the abrupt $CH_4$ jump during the last glacial period with the same tool. Figure 8 shows the $CO_2$ evolution during the onset of abrupt

warming in the NH during MIS 3, and MIS 5. This figure shows the whole sequence of rapid events of the last glacial period (Ahn and Brook, 2014; Bereiter et al., 2012). Atmospheric $CO_2$ during MIS 3, as shown in Figure 8, was reconstructed from the Talos Dome ice core (TALDICE). For MIS 5 it was obtained from Byrd and the EPICA Dronning Maud Land (EDML) ice core (Ahn and Brook, 2008; Bereiter et al., 2012). In Bereiter et al. (2012), both TALDICE and EDML records were used during MIS 3 and compared to the onset of abrupt warming in the NH. However here, we only use data from TALDICE which

are more accurate due to the narrower age distribution of the gas trapped in the LID (Bereiter et al., 2012). Using the same method, the average value of CDM lag with respect to the abrupt warming in NH was calculated. The average CDM lags with respect to the abrupt warming in the NH for the MIS 3 and 5 periods are 770±180 and 280±240 yr (Bereiter et al., 2012). Thus, over the course of the last glaciation, the lag of $CO_2$ maxima with respect to the abrupt NH warming events significantly increased. We observe the same trend through the millennial events depicted during MIS 6, albeit with different absolute lags.





While the earlier $CO_2$ maximum corresponding to CDM 6e.2 during MIS 6 shows no significant lag (thus close to the small lag of $190 \pm 110$ yr observed during MIS 5), the lag between the abrupt warming in NH and CDM 6d.1 and 6d.2 is longer than the lags of CDMs during MIS 3.

## 4 Discussion

### 4.1 Atmospheric CO₂ variability on millennial time scale

We found that the amplitude of atmospheric $CO_2$ variations is well related to the NA stadial duration during MIS 6, which implies that the amplitude of $CO_2$ variations might be affected by the duration of AMOC disruption during the early MIS 6 period (Margari et al., 2010). This hypothesis is also supported by a recent study using oceanic sediment cores from the Southern Ocean (Gottschalk et al., 2019b). The authors report that respired carbon levels in the deep South Atlantic decrease when AMOC is weakened during both glacial periods, and the amount of carbon loss in the deep South Atlantic is highly correlated with the duration of NA stadials.

The hypothesis that perturbations of the AMOC might ultimately lead to changes in atmospheric $CO_2$ concentrations during MIS 6 is supported by numerical simulations (Menviel et al., 2014; Schmittner and Galbraith, 2008). Menviel et al. (2014) report that when large amounts of low-density fresh water are released into the NA, the AMOC is shut down, which strongly reduces heat transport from the south to the north, and moisture transport from the Atlantic to the Pacific is reduced (Leduc et al., 2007; Peterson et al., 2000; Richter and Xie, 2010). In response, sea ice retreats in the Southern Ocean. On the other hand, the NA is cooled, and the ITCZ is shifted to the south and Summer Monsoon intensity in East Asia is reduced (Wang et al., 2008). Changes in precipitation connected to ITCZ shifts (mainly in tropical Africa and northern South America) may cause variations in terrestrial carbon stocks (Bozbiyik et al., 2011; Köhler et al., 2005; Menviel et al., 2008; Obata, 2007). Due to the reduction of Summer Monsoon intensity in East Asia, salinity at the surface of the Pacific Ocean is increased. Thus, AABW and North Pacific Deep Water (NPDW) transport is enhanced (Menviel et al., 2014). Enhanced NPDW transport ventilates deep Pacific carbon via the Southern Ocean which may lead to atmospheric $CO_2$ increases.

As mentioned above, atmospheric $CO_2$ on millennial timescales can be controlled by $CO_2$ exchange between the ocean and the atmosphere, as well as changes of terrestrial carbon stocks. Coupled climate carbon cycle models reported that the variations of atmospheric $CO_2$ concentration on millennial timescales are mainly dominated by deep ocean inventory, requiring a few millennia to react to climate change (Schmittner and Galbraith, 2008). These variations of $CO_2$ concentration might be compensated by fast changes in the terrestrial biosphere (Bouttes et al., 2012; Menviel et al., 2014; Schmittner and Galbraith, 2008). The initial response of the terrestrial biosphere and deep ocean to AMOC perturbations are opposite in the CLIMBER-2 model (Bouttes et al., 2012).

Based on the comprehensive review of Gottschalk et al. (2019a), some models show a $CO_2$ increase during stadial conditions and others show a decrease, with a preference for those showing an increase. During short stadials, a change in the carbon stock of the terrestrial biosphere can compensate the slow response of the deep Ocean (Gottschalk et al., 2019a; Menviel et





al., 2014; Schmittner and Galbraith, 2008). This might explain our observation that atmospheric $CO_2$ variation is rather muted during short stadials, even though other SH climate proxies like dust flux and $\delta D$ composition in EDC show significant variations (Figure 1). However, during long stadials, modelling studies indicate that the deep ocean inventory continues to respond to climate changes on timescales much longer than the response time of the terrestrial biosphere (on the order of a few

centuries), and atmospheric $CO_2$ varies significantly (Gottschalk et al., 2019a).

Another possible reason for the difference between $CO_2$ changes during short and long stadials may be related to a stronger reduction of the NADW during long stadials (Henry et al., 2016; Margari et al., 2010), which would cause a stronger upwelling of deep water in the Southern Ocean (Menviel et al., 2008; Schmittner et al., 2007). When large amounts of low-density fresh water are released into the NA, NADW formation can be slowed down. These events may reduce stratification in the Southern

Ocean due to an increase in salinity of the surface waters and a relative freshening of the deep water (Schmittner et al., 2007). As a result, atmospheric $CO_2$ can be increased due to upwelling and outgassing of $CO_2$ in the Southern Ocean (Schmittner et al., 2007). The co-occurring upwelling in the SO during AIMs for the last termination has been examined (Anderson et al., 2009) but, due to the lack of proxy data with precise age scale for upwelling in the Southern Ocean, this hypothesis cannot be confirmed during MIS 6. During the short stadial in MIS 6 (AIM 6e.1) and the short stadials in MIS 3, the duration and strength

of AMOC disruption are similar. This is supported by the marine proxy data for upwelling in the Southern Ocean which do not show strong variations during short stadials for both MIS periods (Anderson et al., 2009).

According to the model results of Margari et al. (2010), the six AIM events of MIS 6 were likely affected by AMOC perturbations of similar strength. During these events, there is no clear evidence for freshwater perturbation in the NA (Figure S12 in SI), and the strength of the associated AMOC perturbations is estimated to be similar to that during non–Heinrich (no

ice discharge) or non-classic Heinrich (different source signature for IRD, occurring at transitions in ice volume) AIM events of MIS 3. The durations of NA stadials during early MIS 6 (except for AIM 6e.1) appear to be longer than those during non–Heinrich AIM events of MIS 3, which might be caused by the different boundary conditions during MIS 3 and the increase of hydrological cycle during MIS 6.e. A longer duration of the AMOC disruption may require more time to recover AMOC strength, which may impact the amplitude of $CO_2$ variations (Bouttes et al., 2012; Menviel et al., 2008). Considering that the

timescale of the AMOC recovery may be affected by climate background conditions (Bouttes et al., 2012; Menviel et al., 2008), this observation suggests how different climate background conditions may impact atmospheric $CO_2$.

The strength of AMOC perturbations appears to also be an important factor in determining the amplitude of $CO_2$ variations. For example, the duration of the Heinrich events in MIS 3 (AIM 8, 12 and 14) is shorter than any of the MIS 6 events except for 6e.1, but atmospheric $CO_2$ varied significantly in all three.

The relationship between the amplitude of atmospheric $CO_2$ variations and the NA stadial duration is explained by the duration of AMOC disruption during the early MIS 6 period. However, the temporal resolution of $\delta^{18}O$ composition of planktonic foraminifera in MD01–2444 and the precision of the age scale were too low to precisely define the duration of stadials during MIS 6. Additional proxy data providing information about climate change in the NH are needed to confirm the relationship between atmospheric $CO_2$ variations and the NA stadial duration (Figure 3).



The limited available proxy data permit only an exploratory discussion of the mechanisms responsible for $CO_2$ variability during MIS 6. To compare the behaviour of the bipolar see saw with atmospheric $CO_2$ variations, additional investigations about AMOC disturbances and their associated climate responses are needed.

**4.2 Why did $CO_2$ lag the abrupt warming in the NH during MIS 6d?**

Two different lags of $CO_2$ variations with respect to NH warming are present in the MIS 6 period (Figure 8). CDM 6e.2 is nearly synchronous with the abrupt warming in the NH (no significant lag of 200±360 yrs), while the lags for CDM 6d.2 (1,300±450 yrs) and CDM 6d.1 (1,500±280 yrs) are much longer. Two modes of $CO_2$ variations are also observed during the last glacial period. As the last glaciation progressed from MIS 5 to MIS 3 (Figure 8), the lag of $CO_2$ maxima with respect to NH millennial-scale warming significantly increased. This observation may be explained by the different AMOC settings in

MIS 5 and MIS 3 (Bereiter et al., 2012). As observed during the last glacial period, a mode change of oceanic circulation between MIS 6e to MIS 6d might be also the cause of the change in the time lags between NH abrupt warming events and $CO_2$ variations during the early MIS 6.

During MIS 3, the oceanic circulation in the Atlantic was in a "glacial" state, with shallower NADW and carbon–rich AABW extended to the north, while during MIS 5 circulation was similar to the present, in what can be referred to as a "modern–like"

state. At the onset of Dansgaard–Oeschger (DO) events, AMOC is thought to accelerate rapidly, delivering heat to the north and resuming the formation of NADW. When the NADW cell expands, AABW is withdrawn and the upwelling of carbon–rich deep water in the Southern Ocean is enhanced. Essentially, over the time scale of ocean overturn, part of the previously expanded carbon-rich southern sourced water is converted to carbon-poor northern sourced water.

Therefore, due to the northward expansion of AABW before the onset of DO events in the MIS 3 period, additional carbon

may have been added to the atmosphere by deep water exchange from carbon enriched AABW to the NADW. Thus, $CO_2$ continues to be released into the atmosphere for another 500 to 1,000 years while NADW formation continues (Bereiter et al., 2012). However, during MIS 5 where AABW was not expanded in the NA, less $CO_2$ is thought to be released from the ocean to the atmosphere.

The temporal resolution of proxy data related to oceanic circulation during MIS 3 and 5 is unfortunately not sufficient to

validate from the marine realm itself whether the two different modes of $CO_2$ variations reflect the hypothesized mechanism described above. Modelling studies of the carbon stock in AABW and NADW during MIS 5 and 3 have been attempted. However, dependent on the chosen model, the modes of atmospheric $CO_2$ variation are different (Gottschalk et al., 2019a). Some studies, for example, Menviel et al. (2008) generate an atmospheric $CO_2$ decrease when the AMOC is reduced. Others, for example Bouttes et al. (2012) confirm an atmospheric $CO_2$ release from the ocean when the AMOC resumed.

It is possible that a similar change in the AMOC may explain the presence of two different lags in MIS 6. The density of the NA surface water was low during 180–168 kyr BP due to the higher intensity of the low latitude hydrological cycle (Ayalon et al., 2002; Bard et al., 2002; Cheng et al., 2016; Mélières et al., 1997). From MIS 6e to MIS 6d, NADW became shallower and the lag of $CO_2$ maxima with respect to NH millennial-scale warming significantly increased. Moreover, the AMOC cell





appears to have been generally shallower during the early MIS 6 (180–168 kyr BP) relative to MIS 3 (Margari et al., 2010), and so Bereiter et al. (2012)'s hypothesis would indicate an extended $CO_2$ release from an expanded AABW reservoir, potentially explaining the millennial–scale delays with respect to abrupt NH warming events. Wilson et al. (2015) show that circulation conditions in the Atlantic Ocean during the earliest MIS 6 period were similar to those during MIS 5, with deeper

NADW and AMOC than during MIS 6d. This may explain the shorter lag between abrupt NH warming and CDM 6e.2. However, in order to explain the much longer lags of CDM 6d.1 and 6d.2, NADW should be even shallower than in MIS 3. During MIS 6d, the lag between the abrupt warming in NH and CDMs (1,400±375 yrs on average) is longer than the lags of CDMs (770±180 years on average) during MIS 3, which may be related to the observation that the NADW during MIS 6.d might have been shallower than that during MIS 3 (Margari et al., 2010). However, our study has lower temporal resolution

compared to $CO_2$ data set during MIS 3. To confirm the larger lag of $CO_2$ with respect to abrupt warming in NH, additional $CO_2$ reconstructions during MIS 6.d are needed.

To further investigate the applicability of the shallow–AMOC glacial hypothesis to the $CO_2$ variations observed in this study, higher resolution proxy data to estimate variations in the strength of the AMOC during the MIS 6 period are needed. However, as is the case for proxy data during the last glacial period as well, the temporal resolution is unfortunately not sufficient to

quantify the lag between AMOC resumption and temperature increase at the surface during DO warmings (Henry et al., 2016). In addition, because of the low accumulation at EDC, the estimation of the exact timing of CDM from the EDC ice core might be less accurate compared to that from the TALDICE ice core, for example, due to the narrower gas age distribution of TALDICE (Bereiter et al., 2012). To further investigate the exact relationship between CDM and abrupt warming in the NH, additional $CO_2$ measurements from a higher accumulation site could be helpful.

**5 Conclusion**

We measured 150 samples of the EDC ice core to reconstruct atmospheric $CO_2$ during the MIS 6 period (189–135 kyr BP), with an unprecedented time resolution and analytical precision. Five millennial-scale atmospheric $CO_2$ changes are revealed during the early part of MIS 6 (189–160 kyr BP), with amplitudes ranging between 15 to 25 ppm, mimicking similar trends in Antarctic δD variations. During the shortest stadials in the NA, atmospheric $CO_2$ variations are negligible and decoupled with

δD in EDC, probably because the duration of upwelling in the Southern Ocean was not sufficient to impact atmospheric $CO_2$, in line with Ahn and Brook (2014). In the earliest MIS 6 (MIS 6e and 6d, corresponding to 189 to 169 kyr BP), a change of $CO_2$ lags with respect to NH warming – as deduced from atmospheric $CH_4$ changes – is revealed. During MIS 6e, CDM 6e.2 (at ~182 kyr BP) is nearly synchronous with the abrupt warming in the NH (non-significant lag of 200±360 yr), while the lags during MIS 6d corresponding to CDM 6d.1 and 6d.2 (at ~171 and ~175 kyr BP, respectively) are much longer, 1,400±375 yrs

on average. The change in lag time might be related to a change in the organization of the AMOC from MIS 6e to MIS 6d. Similar observations are drawn for the time period covered by our study in comparison with previous studies on MIS 3 and MIS 5 periods, although the lag of $CO_2$ with respect to NH warming reaches larger values during MIS 6d. However, the limited





available proxy data from the marine realm only permits an exploratory discussion of the mechanisms responsible for $CO_2$ variability during MIS 6. Because the boundary conditions of the last glacial period cannot be applied to MIS 6, additional proxy data and multiple modelling studies conducted during MIS 6 period are needed.

**Acknowledgement**

5    This work is a contribution to the "European Project for Ice Coring in Antarctica" (EPICA), a joint European Science Foundation/European Commission scientific program, funded by the European Union and by national contributions from Belgium, Denmark, France, Germany, Italy, The Netherlands, Norway, Sweden, Switzerland, and the United Kingdom. The main logistic support was provided by IPEV and PNRA. This is EPICA publication no. XX. It also received funding from the European Community's Seventh Framework Programmes ERC-2011-AdG under grant agreement no. 291062 (ERC

10   ICE&LASERS). As part of the PhD work of J. Shin, it was also supported by the LabEX OSUG@2020 project of the Grenoble Observatory of Sciences of the Universe (OSUG). Swiss authors also acknowledge long-term financial support of the ice core research at the University of Bern by the Swiss National Science Foundation under grants 200020_159563, 200020_172745, 200020_172506 and 20FI21_189533. The authors would like to thank G. Aufresne for assistance in the additional methane measurements, as well as X. Fain and K. Fourteau for help on the $CH_4$ analytical system and D. Raynaud for the profitable

15   discussions. We would like to thank J. Gottschalk for discussions about $CO_2$ variability and marine sediment core data avaialable for the last two glacial periods.



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





**Table 1:** Ages of the tie points of MD01–2444 record (Margari et al., 2010) on the EDC03 (Parrenin et al., 2007) and AICC2012 ice age scales (Bazin et al., 2013).

| MD01–2444 (m) | Previous age tie points by Margari et al. (2010) (EDC03 age scale, kyr BP) | New age tie points (AICC2012, kyr BP) |
|---|---|---|
| 22.02 | 136.100 | 135.761 |
| 22.50 | 141.686 | 141.677 |
| 23.70 | 149.586 | 150.287 |
| 24.48 | 159.105 | 160.327 |
| 24.72 | 162.476 | 163.878 |
| 25.32 | 168.273 | 170.349 |
| 25.71 | 172.009 | 174.649 |
| 25.95 | 175.461 | 178.423 |
| 26.01 | 177.065 | 180.033 |
| 27.03 | 188.009 | 190.229 |
| 27.30 | 192.231 | 194.186 |



**Table 2:** The minima and maxima defining the duration of each NA stadial using the AICC2012 chronology.

|  | MIS 6c.1 | | MIS 6c.2 | | MIS 6d.1 | | MIS 6d.2 | | MIS 6e.1 | | MIS 6e.2 | |
|---|---|---|---|---|---|---|---|---|---|---|---|---|
|  | Min. | Max. | Min. | Max. | Min. | Max. | Min. | Max. | Min. | Max. | Min. | Max. |
| Age (kyr BP) | 160.1 | 161.9 | 163.2 | 166.0 | 170.2 | 173.5 | 174.2 | 176. 8 | 177.2 | 178.3 | 179.9 | 182.2 |
| Uncertainty (kyr) | 0.2 | 0.2 | 0.2 | 0.2 | 0.2 | 0.2 | 0.2 | 0.2 | 0.2 | 0.1 | 0.1 | 0.1 |



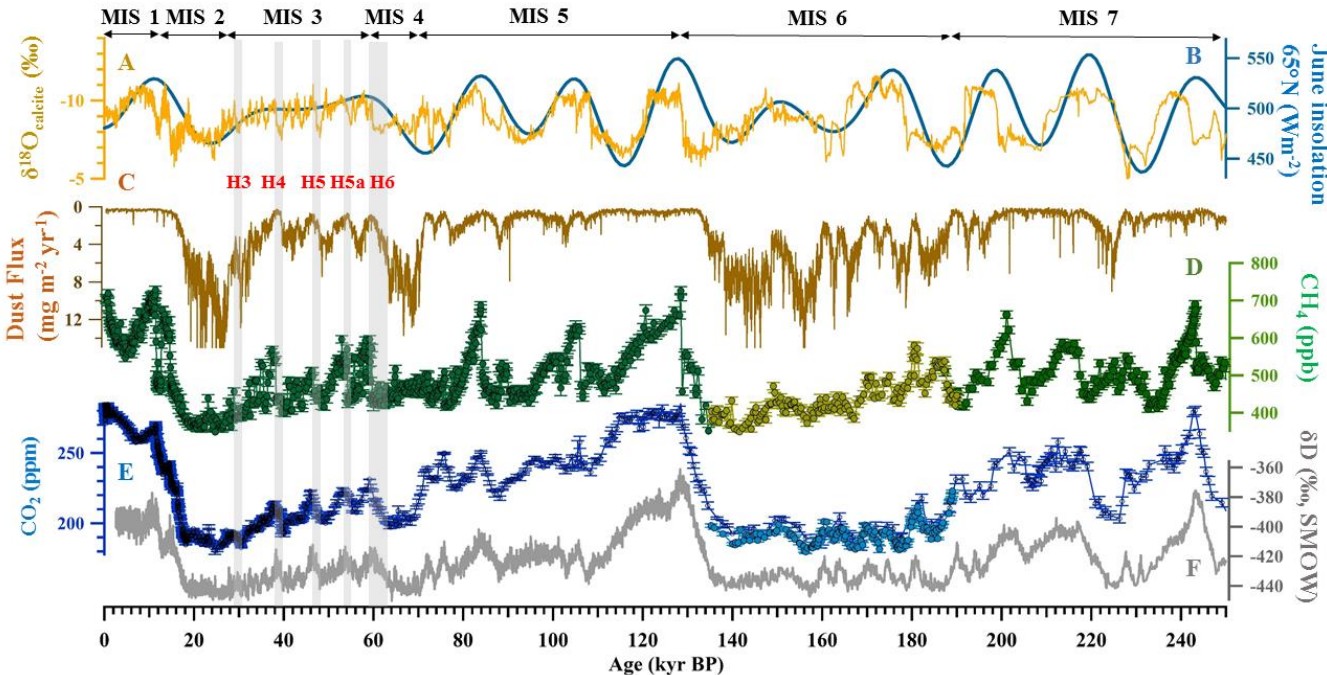

**Figure 1:** Proxy data during 250 kyr BP. A: $\delta^{18}O_{calcite}$ from Sanbao cave, corresponding with the strength of the East Asian monsoon (Cheng et al., 2016). B: 21 June insolation for 65°N (Berger, 1978). C: Dust flux in EDC (Lambert et al., 2012). D: Atmospheric $CH_4$ in EDC (green dots) (Loulergue et al., 2008) and Atmospheric $CH_4$ in EDC in this study (light yellow dots). E: Atmospheric $CO_2$ from EDC in this study (light blue dots) and composite $CO_2$ from Antarctic ice cores (dark blue dots) (Bereiter et al., 2015). F: $\delta D$ composition in EDC, Antarctica (Jouzel et al., 2007). Vertical grey bars indicate the timing of Heinrich events.



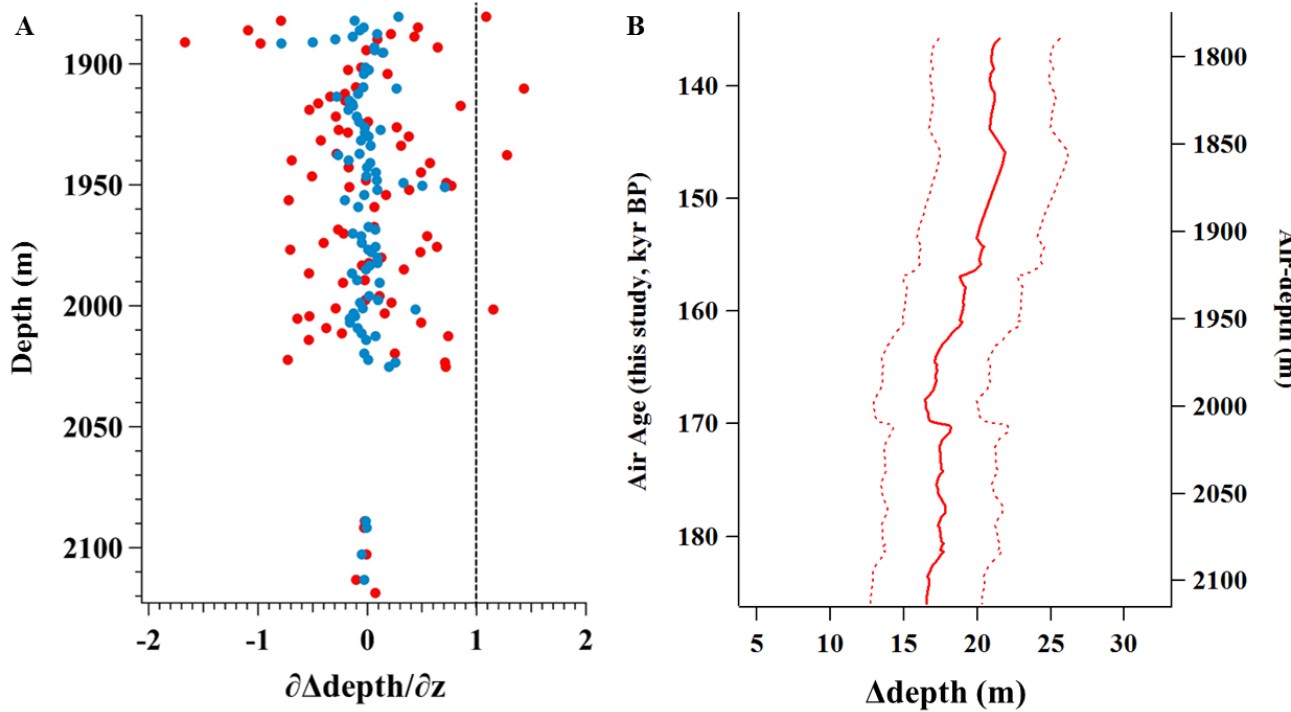

**Figure 2:** A: $\partial\Delta depth/\partial z$ as a function of depth. Red dots from the raw $\delta^{15}N$ measurements. Blue dots from a 3-point running mean weighted by 500/dT. Vertical dashed line indicates when $\partial\Delta depth/\partial z$ function is 1. B: $\Delta depth$ (bold line) for EDC from 1787.5 to 1870.2 m below the surface, deduced from $\delta^{15}N$ and the thinning function calculated in this depth range. The two dash lines correspond to the analytical uncertainties.



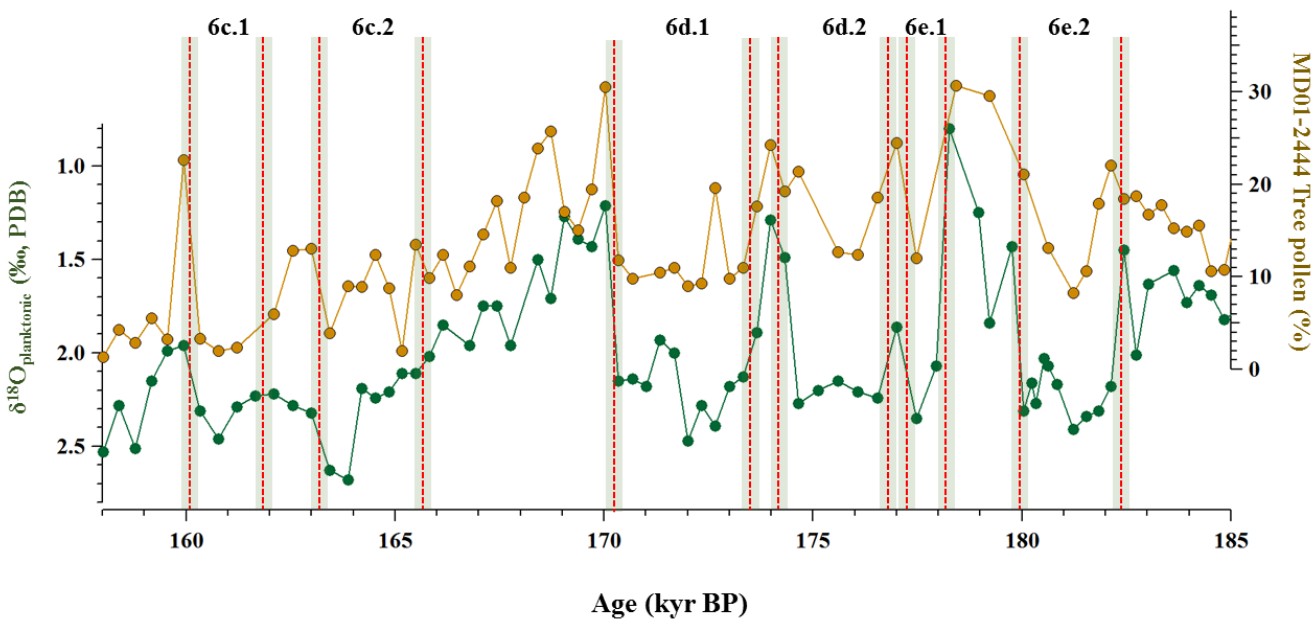

**Figure 3:** Six variations on millennial time scales of tree pollen percentage (top) and $\delta^{18}O$ of planktonic foraminifera (bottom) in the MD01-2444 selected by Margari et al. (2010).





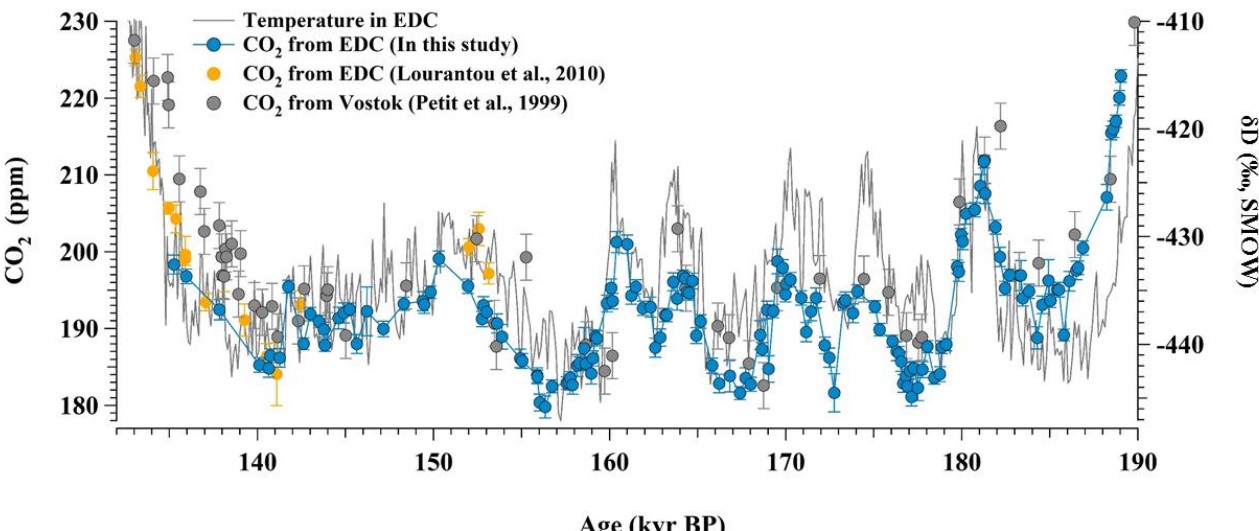

**Figure 4:** Atmospheric $CO_2$ from EDC and Vostok ice cores, compared to the δD of water at EDC (temperature proxy) during 190–135 kyr BP. Blue dots: Atmospheric $CO_2$ from EDC (this study). Yellow dots: Atmospheric $CO_2$ from EDC (Lourantou et al., 2010). Grey dots: Atmospheric $CO_2$ from the Vostok ice core (Petit et al., 1999). Grey line: δD of water at EDC (Jouzel et al., 2007).





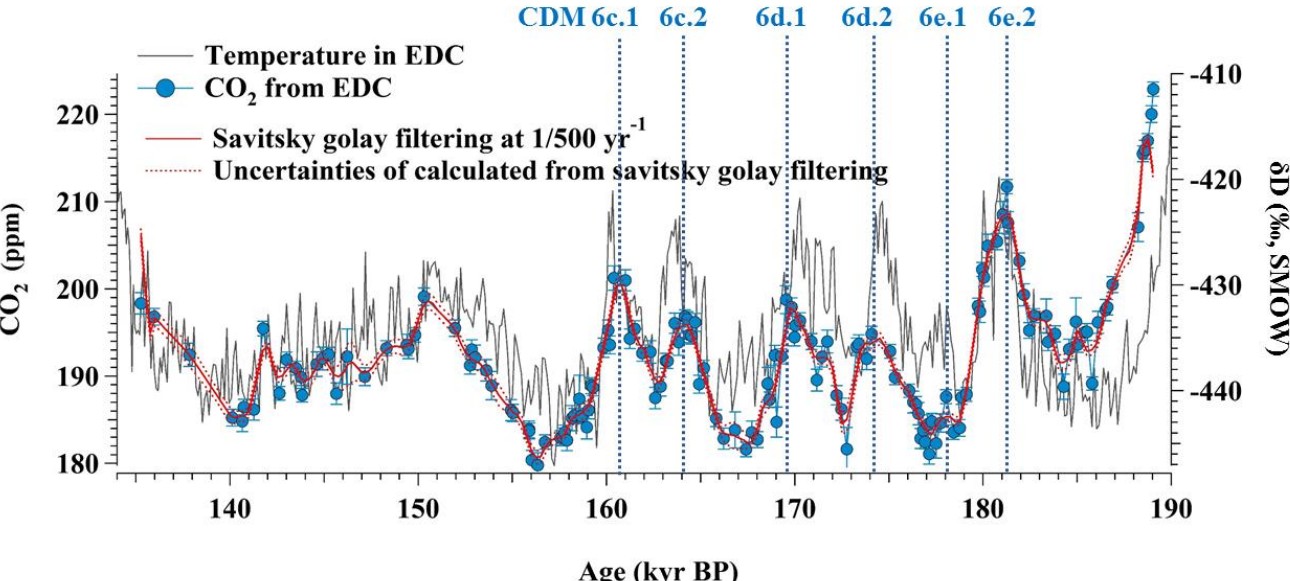

**Figure 5:** Atmospheric CO₂ from EDC (this study) and EDC water isotopic record (Jouzel et al., 2007). Red line indicates Savitsky Golay filtering curve made with a 500 yr cut-off period (red dotted line). Vertical blue dotted lines indicate the six CDM events that we identify during the early MIS 6.



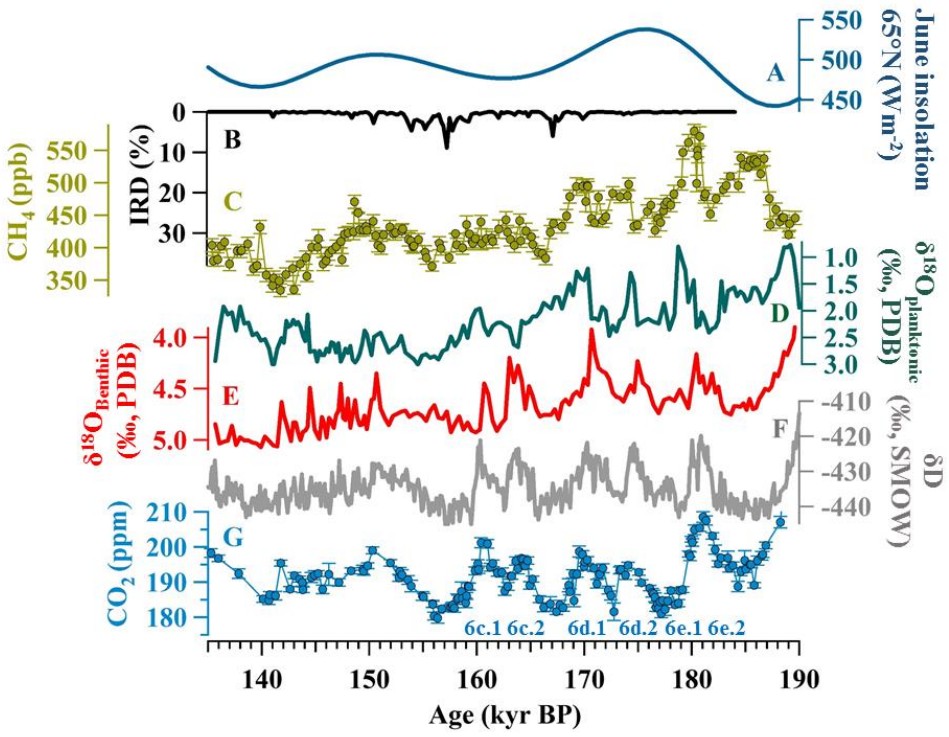

**Figure 6:** A comparison of climate with atmospheric $CO_2$ during MIS 6 period. A: 21 June insolation at 65°N (Berger, 1978). B: Ice-rafted debris (IRD) input in the Iberian margin core MD95–2040 (de Abreu et al., 2003). C: Atmospheric $CH_4$ in EDC (Loulergue et al., 2008; this study). D: $\delta^{18}O$ of planktonic foraminifera in the Iberian margin marine Core MD01–2444 (Margari et al., 2010). E: $\delta^{18}O$ of Benthic foraminifera in the Iberian margin marine Core MD01–2444 (Margari et al., 2010). F: Temperature in Antarctica from $\delta D$ composition of the EDC ice core (Jouzel et al., 2007). G: Atmospheric $CO_2$ in EDC during MIS 6 period (in this study). The numbers of CDM events are written at the bottom.





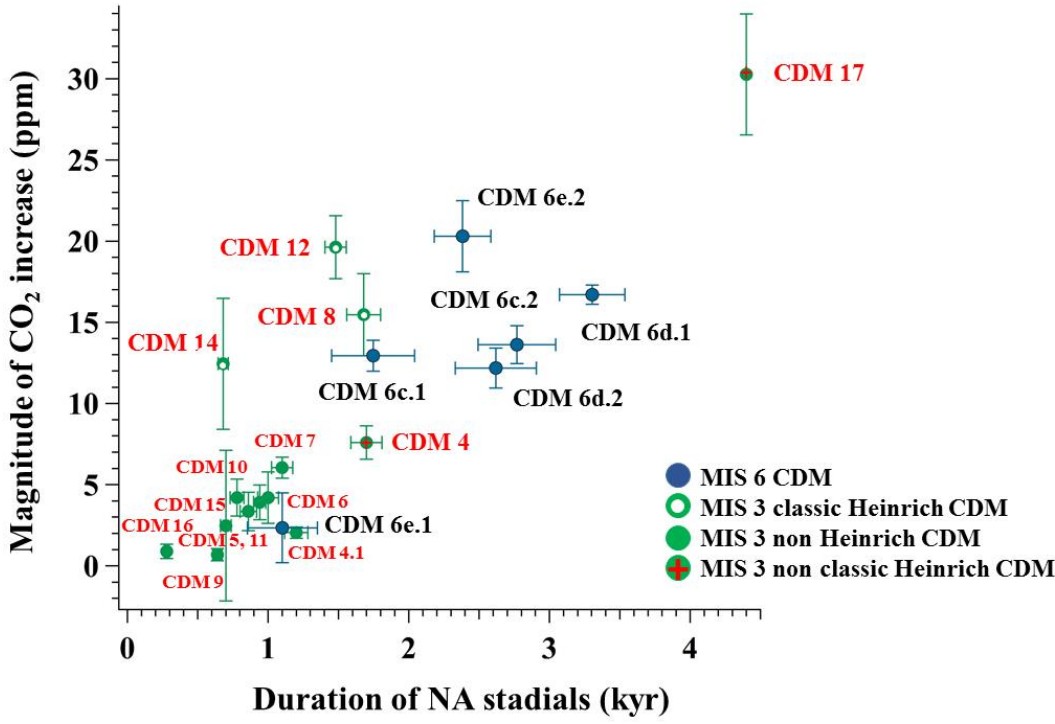

**Figure 7:** The relationship between NA Stadial duration and magnitude of $CO_2$ increase during the early MIS 6 period. Green dots indicate non Heinrich CDM events during MIS 3, green dots with a white dot in the middle indicate classic Heinrich CDM events during MIS 3, and green dots with a red cross in the middle indicate non classic Heinrich CDM events during the MIS 3 period. Blue dots indicate CDM events during MIS 6 respectively.





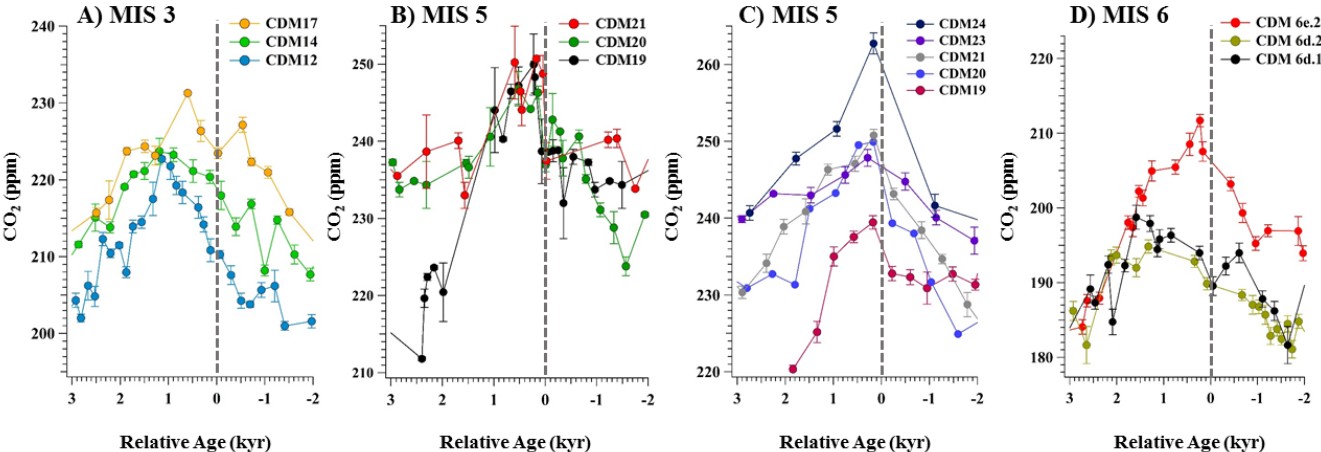

**Figure 8:** CDM lags relative to abrupt temperature increase in NH. Grey dotted lines indicate when climate changes abruptly in NH. A: Atmospheric $CO_2$ was recorded from TALDICE (MIS 3), B: Atmospheric $CO_2$ was recorded from Byrd (MIS 5), C: Atmospheric $CO_2$ was recorded from EDML (MIS 5), D: Atmospheric $CO_2$ was recorded from EDC (MIS 6). For MIS 6, we selected the 3 CDMs that correspond to an abrupt methane increase; the other CDMs do not correspond to an abrupt change. The scale of the y-axis is not the same for the four panels