# Peer review of "Millennial-scale atmospheric CO2 variations during the Marine Isotope Stage 6 period (190–135 kyr BP)"

_Climate of the Past, 2019_

## Referee Comment (RC1) · Anonymous Referee #1 · 17 Jan 2020

Review of manuscript cp-2019-142:

General Comments:

The manuscript by Shin and others presents new, high-resolution measurements of CO2, CH4, and $\delta$15N in EDC ice core samples spanning the glacial period, MIS 6. The new data resolve millennial-scale variations in CO2 and CH4. The authors independently identified MIS 6 stadial durations in tree pollen % and planktonic $\delta$18O in the Iberian Margin marine sediment core MD01-2444. The authors also revised the MIS 6 gas age chronology of the EDC ice core (previously AICC 2012) using new estimates of $\Delta$depth from the $\delta$15N data. The revised EDC age scale, along with the timing of

climate variations observed in the sediment core, provides the authors with a temporal framework for understanding millennial-scale CO2 variations during the penultimate glacial period.

The authors specifically analyze the timing of the CO2 changes relative to changes in CH4, considered here a proxy for NH warming, identifying leads/ lags between the two records. They also discuss differences between the CO2 features in MIS 6 and analogous features that occurred in MIS 3. The authors also observe differences in the magnitudes of CO2 maxima during MIS 6. They identify a relationship between the amplitude of CO2 change and the duration of the preceding stadial event, offering the hypothesis that the amplitude of CO2 variations depends on the duration of AMOC perturbations. They also identify a shift in the lag of CO2 maxima from MIS 6e to MIS 6d and suggest that this may be due to a change in the organization of AMOC.

This manuscript is well written, organized, and clearly presented, the science is in my opinion sound, and the new datasets represent important contributions that will be of interest to others in the field. The work is appropriate for the journal Climate of the Past, and I recommend this paper for publication after minor revisions. Below I list specific comments that, if addressed, will aid in the clarity of the paper and hopefully strengthen the analyses therein. I also list technical corrections below.

Specific Comments:

INTRODUCTION – P3L9 – Can you provide a reference for the longer duration of stadials in early MIS 6?

– P3L15 – There are more pre-existing CO2 measurements from late MIS 6 besides those from Vostok (Lourantou et al., 2010; Schneider et al., 2013).

METHODS – P3L31 – Did the measured CO2 concentration depend on the amount of air injected? (Presumably, the pressure in the sample loop depleted across the 5 individual injections. Was there a linearity effect?)

– P4L18-19 – Was the amount of contamination in each chamber consistent from day to day? Did it depend on the length/ amount of crushing?

– Did you run replicate CO2 measurements on ice samples from the same depths? In my opinion, this would be a better estimate of the true system precision.

– P4L25 – Can you state briefly how the new, corrected CO2 record compares to the preexisting CO2 data?

– P4L32 – Can you state the precision of the CH4 measurements?

– P5L3 – What do you think are possible reasons for the systematic offset? Please describe briefly.

– P6L20 – Figure S4 in the SI does not have a label to distinguish blue from red.

– P6L22 – I do not follow how Figure 2 supports the claim that the previous method was "relatively unbiased but not entirely exact."

– P6L13 – In Figure 3 it appears that the midpoints in the transitions are somewhat ambiguously defined. Sometimes they fall between a local max and min for d18O, sometimes for pollen %. The markers are chosen as midpoints between local maxima/ minima, but sometimes it is unclear where those max/ min data points are. 6d.2, for example, could easily be shorter (i.e., it looks like the end marker at 174.2 ka could be defined at an older age). 6c.2 is a particularly ambiguously defined stadial – I do not see which maximum and minimum pair defines the older marker. Could you define the stadial durations more objectively? The ambiguity and subjectivity in picking the stadial transitions lead me to believe that they were defined while also considering the ice core data. That's not necessarily a bad thing, but perhaps you should just be forthright and show the gas data in Figure 3 along with the sediment core data.

RESULTS – P8L3 – You should mention the known phenomenon of CO2 offsets between different ice cores (e.g. WAIS versus Law Dome). The co-author Christoph could certainly comment on this.

– P25Fig5 – It is unclear how the blue CDM events were defined. Do they relate somehow to the stadial duration markers you defined previously? If not, please clarify how you identified them (or provide proper reference to SI).

– P26Fig6 – Shading or vertical lines would help to delineate the CDM's in Figure 6. Right now the text floats at the bottom and is unclear exactly what the labels refer to.

– One result that strikes me as interesting, and not discussed in the paper, is that the lowest $CO_2$ and Antarctic temperature values occur in the early/ middle part of MIS 6, not the latest part (as in MIS 2). $CH_4$, on the other hand, reaches the lowest values during late MIS 6, right before the termination, as does peak glaciation as inferred from the benthic d18O. This is unlike MIS 2, which is characterized by low $CO_2$, low Antarctic temperature, low $CH_4$, and peak glaciation occurring simultaneously. Can you speculate why $CO_2$ is higher in late MIS 6 relative to earlier in MIS 6, despite full glacial extent?

– P28Fig8 - The authors compare the timing of $CO_2$ maxima relative to the onset of NH warming. The $CO_2$ measurements come from different ice cores with different age scales (to my knowledge at least, Byrd is not synchronized to the AICC 2012 as EDML, EDC, and TALDICE are). What is the bias or uncertainty in the analysis due to age offsets? Why not exclusively use the EPICA cores on a unified age scale for this analysis?

DISCUSSION – P11L26&31 – When you say that the terrestrial biosphere can "compensate" for the slow response of the deep ocean, do you mean in terms of its timing or in terms of the direction of $CO_2$ change? Please clarify. "Compensate" may not be the best word to use in case it is confused with carbonate compensation.

– P13 – After the discussion of AMOC and deep ocean ventilation, I realized there was no discussion entertaining productivity fluctuations as a possible mechanism for millennial-scale $CO_2$ variability (Ziegler et al., 2013; Gottschalk et al., 2016; Anderson et al., 2014; Martinez-Garcia et al., 2014).

– P13L13-18 – Need more references in this paragraph.

– P14 – After reading this section it strikes me that there is a large amount of discussion about AMOC changes without actually showing any AMOC data. The discussion is very "AMOC-centric." Indeed, we believe that AMOC changes are probably key to explaining the MIS 3 $CO_2$ changes, but to assume the same mechanism operates in MIS 6 without data to suggest so, and then to make assertions about the AMOC based on the $CO_2$ trends at least requires some qualification in my mind. It is okay to speculate, but please say explicitly that you are doing so and that it is based on extrapolation of the relationships observed in MIS 3.

CONCLUSIONS – P14L22 – "Unprecedented" strikes me as too strong of a word.

– I think the conclusion section should contain less about the AMOC. The primary contributions of the paper (in my mind) are the new data, the revisions to the EDC gas age scale, and perhaps the observations of leads/ lags relative to abrupt $CH_4$ changes. The differences in the organization of AMOC between and within MIS 6 and MIS 3, as well as the relationships between stadial length and AMOC perturbation should be left out here. They are interesting hypotheses, but they are not supported by data. See also my note above about rewording the discussion to be more explicitly speculative.

Technical Corrections:

– Section 2.4 is titled "Ice age revision. . ." but the gas chronology, not the ice chronology, is what is actually revised. It might be confusing, so consider titling this section "Gas age revision. . ."

– In Figure 8 the authors show various $CO_2$ maxima plotted against the lead/lag with respect to the onset of Northern Hemisphere warming. It would be helpful to clarify, for example, "CDM 12" corresponds to DO 12, etc.

– P2L10 – Capitalize "Hemisphere" in "Northern and Southern hemisphere, respectively."

– P2L15 – "opposite"

– P2L17 – I suggest leaving out "In response to the millennial temperature perturbations,"

– P2L32-33 – No need to repeat "MIS 3" and "MIS 6" in parentheses. Just state the age ranges.

– P2L32 – Why just "early MIS 6?" The data also span some of late MIS 6, younger than 160 kyr.

– P3L10 – I think the sentence about a shallower AMOC cell can be combined with the preceding discussion about weaker AMOC.

– P12L8-9 – You already said this in the previous sentence (NADW can be slowed down after freshwater forcing). I think it can be omitted.

– P25Fig5 – There is a typo in the legend. "Uncertainties of calculated from savitsky golay filtering." I am not certain exactly what it is supposed to say.

– SI P7FigS7 – The caption says "Two boxes..." but there are five.

Anderson, R. F., Barker, S., Fleisher, M., Gersonde, R., Goldstein, S. L., Kuhn, G., Mortyn, P. G., Pahnke, K., and Sachs, J. P.: Biological response to millennial variability of dust and nutrient supply in the Subantarctic South Atlantic Ocean, Philosophical Transactions of the Royal Society a-Mathematical Physical and Engineering Sciences, 372, 17, 10.1098/rsta.2013.0054, 2014.

Gottschalk, J., Skinner, L. C., Lippold, J., Vogel, H., Frank, N., Jaccard, S. L., and Waelbroeck, C.: Biological and physical controls in the Southern Ocean on past millennial-scale atmospheric CO2 changes, Nat. Commun., 7, 11, 10.1038/ncomms11539, 2016.

Lourantou, A., Chappellaz, J., Barnola, J. M., Masson-Delmotte, V., and Raynaud, D.: Changes in atmospheric CO2 and its carbon isotopic ratio during the penultimate deglaciation, Quat. Sci. Rev., 29, 1983-1992, 10.1016/j.quascirev.2010.05.002, 2010.

Martinez-Garcia, A., Sigman, D. M., Ren, H. J., Anderson, R. F., Straub, M., Hodell, D. A., Jaccard, S. L., Eglinton, T. I., and Haug, G. H.: Iron Fertilization of the Subantarctic Ocean During the Last Ice Age, Science, 343, 1347-1350, 10.1126/science.1246848, 2014.

Schneider, R., Schmitt, J., Kohler, P., Joos, F., and Fischer, H.: A reconstruction of atmospheric carbon dioxide and its stable carbon isotopic composition from the penultimate glacial maximum to the last glacial inception, Climate of the Past, 9, 2507-2523, 10.5194/cp-9-2507-2013, 2013.

Ziegler, M., Diz, P., Hall, I. R., and Zahn, R.: Millennial-scale changes in atmospheric $CO_2$ levels linked to the Southern Ocean carbon isotope gradient and dust flux, Nature Geoscience, 6, 457-461, 10.1038/ngeo1782, 2013.

---

## Referee Comment (RC2) · Anonymous Referee #2 · 24 Jan 2020

Shin et al. present new records of ice core CO2, CH4, and d15N from the EPICA dome C core during MIS 6. They show that the CO2 maxima tend to lag Antarctic d18O maxima, by an amount that increases thought the glacial period. The magnitude of CO2 increase scales with the duration of North Atlantic stadial period, suggesting a key role of AMOC variations in millennial-scale CO2 variability.

The data and analysis are obviously of interest to the broader paleoclimate community, and this paper should be published with only minor corrections. I believe that the manuscript can be clarified in some places. The main conclusion seems to be that MIS6 behaves very similarly to the last glacial period (as expected). Owing to this

similarity, the discussion is somewhat long and involves a lot of speculation – much of which has already been said in earlier work (for example Bereiter et al. 2012).

Throughout the paper the authors present speculative climatic mechanisms, or uncorroborated results from individual model simulations as established fact. One example: "Due to the reduction of Summer Monsoon intensity in East Asia, salinity at the surface of the Pacific Ocean is increased. Thus, AABW and North Pacific Deep Water (NPDW) transport is enhanced (Menviel et al., 2014). Enhanced NPDW transport ventilates deep Pacific carbon via the Southern Ocean which may lead to atmospheric CO2 increases." While this is not a bad description of Menviel 2014, I think this would be better presented with some caution because while possibly correct, this is in no way a consensus view.

Throughout the paper, the authors compare MIS 3 and MIS6. In several places the authors write that MIS3 and MIS6 had different "background conditions". I am not sure what is meant by that. In what way are they really different? Both periods represent a range of orbital conditions, sea ice volumes, ITCZ positions, Heinrich events etc. So there are many places where they are very similar. I would advise the authors remove this idea that these two glacial periods are somehow very different – I don't think they have made the case that they are (and their data surely suggest that the carbon cycle responds in a very similar manner).

Due to the historical convention the last ice age is actually MIS2-4, rather than just MIS3. So a more meaningful comparison would be MIS2-4 to MIS6. Also, the authors also include MIS5 in their analysis (Fig. 8). I think the paper would be a lot simpler if the authors just claim to be studying millennial-scale CO2 variability, rather than focus on Marine Isotope Stage distinctions that may not be relevant.

In all figures I would appreciate a more clear demarcation of the sub-sections. I am not very familiar with the MIS6a-6e definitions. Do they follow precession/Benthic sequences like in MIS5, and who has defined these? Could you please add the MIS5 and

MIS6 (and MIS7?) sub-stage numbering into figures 1, 4 and 5. Also, for consistency you should mark the H-events of stage 6 in Fig 1. Where does the event numbering 6.e1 etc. come from? I have seen alternative numberings elsewhere in the literature.

Could you please add the synthetic Greenland reconstruction from Barker et al. 2011 to Figure 3, to see how it compares?

The authors do not address the CO2 offset between the records enough. It is up to 10 ppm with Vostok, which is quite large. They explain this as due to the blank correction, which is only around 1.7 ppm and therefore insufficient. Such offsets are seen more often in comparing CO2 from different cores, and may actually be in the ice. Can you explain the EDC CO2 offset between this work and Lourantou?

Specific line-by-line comments:

P1L16: I don't think you can argue that the background conditions are different. That hasn't been established.

P2L11: Broecker does not talk about the bipolar seesaw, but a seesaw in deepwater formation. Other references to consider are Blunier & Brook (2001); Pedro et al. (2018).

P3L6: Normally a stronger monsoon is not associated with a weaker AMOC. How does this work?

P6L9: Add or replace with Etheridge et al (1992); this idea is much older.

P5L5: The "assumption" that the bipolar seesaw was present is a pretty obvious one, and I don't think it needs to be questioned. My personal choice would have been to use Antarctic isotopes to define the stadials and interstadials (see e.g. Kawamura et al., 2017), rather than NA sediments that have much poorer age control.

P7L31: the offsets persist in periods of stable CO2, suggesting there is more than chronological error going on. Please discuss offsets between the cores.

P10L19: again, the link between monsoon and AMOC does not make sense to me

P12L7: upwelling or ventilation /de-stratification?

P12L15-16: Anderson does not cover MIS6, plus those records lack the resolution to investigate short stadials.

P13L22: yet the $CO_2$ variations of MIS5 are larger than those in MIS3?

P14L22: remove "unprecedented". Some ice core $CO_2$ records have decadal precision.

P21: Could you add the H-events of Stage 6 also (or perhaps an IRD record that spans the full period)? Could you mark the MIS6a-6e substage numbering? (I am not familiar with this nomeclature).

P26: could you add the DO onsets you infer from $CH_4$ as vertical bands?

P27: Why did you not add the Stage 5 events here?

References:

Blunier, T., & Brook, E. J. (2001). Timing of millennial-scale climate change in Antarctica and Greenland during the last glacial period. Science, 291(5501), 109-112.

Etheridge, D. M., Pearman, G. I., & Fraser, P. J. (1992). CHANGES IN TROPO-SPHERIC METHANE BETWEEN 1841 AND 1978 FROM A HIGH ACCUMULATION-RATE ANTARCTIC ICE CORE. Tellus Series B-Chemical and Physical Meteorology, 44(4), 282-294. doi:10.1034/j.1600-0889.1992.t01-3-00006.x

Kawamura, K., Abe-Ouchi, A., Motoyama, H., Ageta, Y., Aoki, S., Azuma, N., . . . Yoshimoto, T. (2017). State dependence of climatic instability over the past 720,000 years from Antarctic ice cores and climate modeling. Science Advances, 3(2). doi:10.1126/sciadv.1600446

Pedro, J. B., Jochum, M., Buizert, C., He, F., Barker, S., & Rasmussen, S. O. (2018). Beyond the bipolar seesaw: Toward a process understanding of interhemispheric coupling. Quaternary Science Reviews, 192, 27-46. doi:10.1016/j.quascirev.2018.05.005

---

## Referee Comment (RC3) · Anonymous Referee #3 · 1 Apr 2020

This manuscript presents new and important atmospheric CO2 concentration data from the penultimate glacial period, also known as MIS6. The data concern so-called millennial-scale climate change, which has been well documented from Greenland ice cores. Because the Greenland ice cores do not extend back into MIS6, the natural archive in which to study millennial-scale climate for this period is Antarctic ice. The data appear to be of high quality and the discussion is appropriately oriented to the question of the temporal lag of peak CO2 behind millennial-scale warm intervals. The lag is found to be larger in the colder intervals than in the warmer intervals, much as was previously found for the more recent period of MIS 3 to MIS 5.

[Figure]

The one major thing I find lacking in this paper is replication of CO2 data points from the same depth in the ice core. Replication of gas measurements in ice cores is fundamental in order to have confidence in the accuracy of the data. Furthermore, the authors should calculate a pooled standard deviation from the means of replicates cut from the same depth in the ice core. This is widely viewed in the field as the most reliable indicator of the overall precision of the measurement, including potential issues arising from the ice itself (such as in-situ CO2 production). It is now well known that bacteria living in the ice can and do produce CO2. The only question is, how much? So it is absolutely essential to replicate CO2 analyses on pieces of ice cut from the same depth (and therefore presumably the same age, and having been exposed to the same atmospheric gas concentrations).

Therefore the authors must return to the laboratory and measure essentially another 150 pieces of ice, before this manuscript can be published in CP. The authors must also quote their value they have found for the pooled standard deviation.

I also did not notice any mention of the number of samples that were rejected (but perhaps I just missed it). The authors must mention this number clearly in the main text (not in the Supplement).

Another problem with the manuscript as it stands is the large amount of speculation in the discussion. This doesn't add to the value of the paper and can be mostly cut out, or clearly labelled as speculation in the text.
* * *

---

## Referee Comment (RC4) · Anonymous Referee #4 · 3 Apr 2020

Jinhwa Shin and colleagues present new measurements of CO2 trapped in bubbles of ice at the EPICA Dome C (EDC) site in Antarctica during the penultimate glaciation. They reconstruct a high-resolution record of atmospheric CO2 changes and compare its variations to climatic signals from Antarctica and the North Atlantic region. For the early part of their glacial record, atmospheric CO2 and CH4 display contrasting lags, shifting from hundreds to more than one thousand years. The authors interpret this shift in terms of a reorganization of the Atlantic meridional overturning circulation, and also conclude that the amplitude of CO2 variations may be influenced by the duration of AMOC perturbations.

[Figure]

The new data are welcome, nearly tripling the existing CO2 record from Vostok and nearly doubling the existing CH4 record from EDC, and the resulting discussions are worthwhile. This is a potentially valuable new contribution and can be considered for publication following revision that should include addressing the following points.

1) The new CO2 data are offset to lower values from previous data, and are not replicated. Is there an explanation for the first point and a justification for the latter?

2) The authors state that they "make use of contrasting boundary conditions during the last two glacial periods to gain insight into the co-occurring carbon cycle changes." They then note that those boundary conditions are only "slightly different", and in the end, never explain what those differences are or why they might be expected to matter. This undermines the rationale for proceeding with this study and should be much better explained in a revised manuscript.

3) It appears that a potentially significant conclusion of the manuscript derives from the observations associated with a single small millennial event. This hardly seems justified and should be bolstered either by theoretical arguments or indications of similar behavior in existing data from another time interval.

4) The division and labeling of sub-events is neither referenced nor adequately described, much less explained. Such division is understandable and can be helpful, but only if clearly delineated and consistently applied. Are the divisions related to marine oxygen isotopes, and should they be, or to something else? What is the justification for 6c, 6d, and 6e, when there is no 6a or 6b?

Smaller points to be considered and addressed:

Page 2 line 15 – "opposite behaviour"

Page 2 line 17 – Do the authors really infer that CO2 changes are "in response" to temperature?

Page 7 line 8 - The half cycle between minimum and maximum values is not the definition of an inflection point, nor is it any point at all.

Page 14-15 – Data do appear to be limited, although Helmke, 2003, Kandiano, 2003, Obrochta 2014, Mokkeddem 2016, and Barker 2015 come to mind.

Page 15, line 16 – "available"

Figure 3 – What are the "Six variations on millennial time scales. . ."?

Figure 5 – Golay should be capitalized in the legend.

---

## Editor Comment (EC1) · Denis-Didier Rousseau (Editor) · 6 Apr 2020

Dear authors,

I received the necessary reviews I was looking for and I apologize if some of them were released with some delays. However, I sincerely regret that you didn't use the opportunity offered by the journal's format to already reply and comment the early reviews you got. As the discussion is by now closed, no more comment can be submitted by the community and the reviewers. You are invited to publish your replies to these detailed reviews prior any invitation from my side about the final decision.

[Figure]

Looking forward to reading from you

All the very best

denis-didier Rousseau (Climate of the Past co-editor in chief)

———————————————————

---

## Author Comment (AC1) · 1 Jun 2020

Dear reviewers,

We are grateful for the detailed suggestions, and we believe that these suggestions will considerably improve our study. Below, we address the comments in blue and the revised texts in the manuscript in green. All the best, on behalf of all co-authors,

Jinhwa Shin

Please also note the supplement to this comment:

[Figure]

https://www.clim-past-discuss.net/cp-2019-142/cp-2019-142-AC1-supplement.pdf

[Figure]

**Supplement:**

Dear reviewers,

We are grateful for the detailed suggestions, and we believe that these suggestions will considerably improve our study. Below, we address the comments in blue and the revised texts in the manuscript in green.

5  All the best, on behalf of all co-authors,

Jinhwa Shin

**Anonymous Referee #4**

10

Jinhwa Shin and colleagues present new measurements of CO2 trapped in bubbles of ice at the EPICA Dome C (EDC) site in Antarctica during the penultimate glaciation. They reconstruct a high-resolution record of atmospheric CO2 changes and compare its variations to climatic signals from Antarctica and the North Atlantic

15  region. For the early part of their glacial record, atmospheric CO2 and CH4 display contrasting lags, shifting from hundreds to more than one thousand years. The authors interpret this shift in terms of a reorganization of the Atlantic meridional overturning circulation, and also conclude that the amplitude of CO2 variations may be influenced by the duration of AMOC perturbations.

20  The new data are welcome, nearly tripling the existing CO2 record from Vostok and nearly doubling the existing CH4 record from EDC, and the resulting discussions are worthwhile. This is a potentially valuable new contribution and can be considered for publication following revision that should include addressing the following points.

25  1)  The new CO2 data are offset to lower values from previous data, and are not replicated. Is there an explanation for the first point and a justification for the latter?

**CO₂ offsets:** The ball mill system has a different extraction efficiency depending on the presence of bubbles and/or clathrates in the ice sample, which may cause an accuracy for reconstructing absolute mean $CO_2$ level. When the air is extracted from an ice core sample where bubble and clathrates co-exist, different dry extraction

30  methods with different extraction efficiencies on bubbly and clathrate ice may lead to biased $CO_2$ concentrations (Lüthi et al., 2010; Schaefer et al., 2011). During clathrate formation, the gas is partitioned into clathrates due to the different gas diffusivities and solubilities (Salamatin et al., 2001). $CO_2$ has consistently been observed to be depleted in bubbles and enriched in clathrates (Schaefer et al., 2011). Degassing from clathrates during extraction takes much longer than air release from bubbles; thus, if air from the clathrate ice is not extracted entirely, $CO_2$

35  measurement will be lower than the true value.

The ball mill shows extraction efficiencies of ~62% for bubbles and ~52% for clathrates on average (Schaefer et al., 2011). If the ball mill is used to reconstruct $CO_2$ in Bubble–Clathrate Transformation Zone (BTCZ), $CO_2$ concentrations can be biased. $CO_2$ concentrations from EDC were reconstructed from 150 depth intervals that cover 2036.7 to 1787.5 m along the EDC ice core, which consist of clathrate ice. There exists true small scale

40  variability in $CO_2$ concentrations in the ice below the Clathrate Zone (Lüthi et al., 2010). Due to the diffusion effect, this small variation of atmospheric $CO_2$ is smoothed. Thus, $CO_2$ concentrations in these depth intervals

might represent the initial mean atmospheric concentration. However, the EDC ice core for MIS 6 was drilled in 1999 and, the ice core has been stored for ~20 years in cold rooms at -22.5 ± 2.5°C before the gas is analysed. More than 50% of the initial hydrates present in the freshly drilled ice may have been decomposed and transformed into secondary bubbles, or gas cavities (Lipenkov, *Pers. Comm.*). We expect the same fractionation as during the clathrate formation process, hence bubbles would be depleted in $CO_2$. Thus, $CO_2$ concentrations from EDC may be lower. The portion of the Vostok ice core covering MIS 6 is also clathrate ice, but it was drilled in 1998 and measured immediately (Petit et al., 1999), and less clathrates may have transformed into secondary bubbles. Thus $CO_2$ concentrations from Vostok during MIS 6 may be higher and potentially reflect the true atmospheric concentration more closely. In our study we concentrate on the relative millennial changes of $CO_2$ around the mean glacial concentration, which are the same in all the $CO_2$ records available so far, Thus, our conclusion in this paper are independent of which absolute mean $CO_2$ level is correct. As the new data in this study are currently the best quality data in terms of repeatability, we use our new data as the reference record and correct for any inter-core offsets. We, however, state explicitly in the text that the absolute mean $CO_2$ level during MIS6 is not known better than 5 ppm.

This offset does appear to evolve over time, changing during late MIS 6. Additionally, uncertainties in the alignment of the Vostok and EDC age scales over MIS 6 make it unclear if the variations in the two data series are indeed contemporaneous.

This is written to Section 3.1 The new high-resolution and high precision $CO_2$ record during MIS 6.

**Data verification:** Replicates account for differences between two ice samples at the same depth, making a better estimate of standard deviation of the final measurement but not necessarily of system precision itself. For example, Lüthi et al. (2010) show that there exists true small scale variability in $CO_2$ concentrations in the ice below the Bubble Clathrate Transition Zone, which could be accounted for by using replicates, especially for small sample sizes. Due to the diffusion effect, this small variation of atmospheric $CO_2$ is smoothed to some degree. In our study, large sample sizes (40g) of the ball mill system were used to reconstruct atmospheric $CO_2$, so a low-noise signal from the ice core is extracted (the smaller measurements used in other systems would be noisier in theory). The standard deviation of the measurement is estimated from the 5 injections, but system precision was calculated from blank measurements, which were performed after every 10 measurements accounting for the possible sources of $CO_2$ contamination with our analytical procedure.

To verify our new dataset, we made a composite data set using by aligning previous sets of measurements made over the MIS 6 period on the EDC ice core to our dataset. First, we compared to two existing $CO_2$ data sets and two new $CO_2$ data sets from EDC (Figure 1 and Table 1). There are two published $CO_2$ datasets for EDC during MIS6—the first measured by the ball mill system at IGE (Lourantou et al., 2010) and the second by the sublimation system at CEP (Schneider et al., 2013). We also compared unpublished atmospheric $CO_2$ measurements from EDC by a novel centrifugal ice microtome (CIM) system, a needle cracker and a ring mill system (Shin, 2019). All records are on the AICC2012 air age scale (Bazin et al., 2013). All data sets is corrected for the gravitational fractionation effect using the new $\delta^{15}N$ data in our study.

[Figure]

**Figure 1:** Atmospheric $CO_2$ from EDC and Vostok ice cores, compared to the δD of water at EDC (temperature proxy) during 190—135 kyr BP. Blue dots: Atmospheric $CO_2$ from EDC by ball mill system (this study). Yellow dots: Atmospheric $CO_2$ from EDC by ball mill system (Lourantou et al., 2010). Purple dots: Atmospheric $CO_2$ from EDC by ring mill system. Red equilateral triangles: Atmospheric $CO_2$ from EDC by needle cracker. Black inverted triangles: Atmospheric $CO_2$ from EDC by CIM. Green rhombuses: Atmospheric $CO_2$ from EDC by sublimation. Grey dots: Atmospheric $CO_2$ from the Vostok ice core (Petit et al., 1999). Grey line: δD of water at EDC (Jouzel et al., 2007).

Because of the limited amount of samples available, the data reconstructed by both ball mill and ring mill methods are single measurements from the depth interval. $CO_2$ records by CIM, needle cracker and the sublimation methods were reconstructed from 2–5 replicates from individual depth intervals. The error bars of data without replicate indicate that the standard deviation of five consecutive injections of the gas extracted from each sample into the gas chromatography (Lourantou et al., 2010; Petit et al., 1999). The error bars of data with replicate indicate the standard deviation of the mean of replicates from the same depth interval (Schneider et al., 2013). Figure 1 shows $CO_2$ concentrations measured by the ball mill system, the ring system, the sublimation, the CIM and the needle cracker. These $CO_2$ concentrations by the ball mill system (Lourantou et al., 2010), the ring system, the sublimation (Schneider et al., 2013), the CIM and the needle cracker are systematically higher than $CO_2$ concentrations measured by the ball mill system in our study (Table 1 and Figure 1). Atmospheric $CO_2$ during the MIS 6 period shows an offset between $CO_2$ data in this study and other $CO_2$ sets, which might be related with different analytical methods.

Where the additional datasets have enough resolution, the millennial-scale variations shown in our MIS 6 dataset are reproduced. Nevertheless, the measurements in the different datasets cannot be immediately aggregated because of offsets between their absolute $CO_2$ values. Offset residuals show that these offsets do not present any significant temporal evolution over MIS 6, but rather appear to be constant. In order to estimate these offsets while accurately accounting for both measurement uncertainty and uncertainty in the offsets themselves, we rely on a Monte Carlo procedure, which is run for 1000 iterations. At each iteration, the data from all datasets is resampled

within its measurement uncertainty. Then, a Savitsky-Golay filter with an approximate cutoff period of 150 years (using a 7-point sliding window and cubic fit, sampled at 250-year resolution) is applied to the new EDC data from this study. The offsets between each additional dataset and our data are calculated. At the end of the stochastic procedure, mean and standard deviations of each offset are calculated, and used to adjust each dataset

5    to create the composite.

In order to test the sensitivity of the stack to the interpolation methods, Monte Carlo procedures were also run using linear interpolation, cubic spline filtering, and enting spline filtering in place of the Savitsky-Golay filter. The mean calculated offsets did not vary by more than 0.2 ppm depending on the method, well within the uncertainty ranges calculated for the offsets themselves.

[Figure]

**Figure 2:** Atmospheric $CO_2$ from EDC and Vostok ice cores, compared to the $\delta D$ of water at EDC (temperature proxy) during 190—135 kyr BP. Blue dots: Atmospheric $CO_2$ from EDC by ball mill system (this study). Yellow dots: Atmospheric $CO_2$ from EDC by ball mill system (Lourantou et al., 2010). Purple dots: Atmospheric $CO_2$ from EDC by ring mill system. Red equilateral triangles: Atmospheric $CO_2$ from EDC by needle cracker. Black inverted triangles: Atmospheric $CO_2$ from EDC by CIM. Green rhombuses: Atmospheric $CO_2$ from EDC by sublimation. Grey dots: Atmospheric $CO_2$ from the Vostok ice core (Petit et al., 1999). Grey line: $\delta D$ of water at EDC (Jouzel et al., 2007).

10   There are two main sources of uncertainty in the composite dataset--the measurement uncertainty of the data, and the uncertainty of the offset itself. The offset uncertainty is not independent for each point, but should rather have very high covariance, which we cannot account for exactly since there are no exact replicates, and we are limited to estimating a mean offset for each data set. Therefore, these two sources of uncertainty are presented separately, and not aggregated.

15   We also use this procedure to estimate an offset between our data and the data measured on the Vostok ice core. However, this offset does appear to evolve over time, changing during late MIS 6. Additionally, uncertainties in

the alignment of the Vostok and EDC age scales over MIS 6 make it unclear if the variations in the two data series are indeed contemporaneous. We therefore do not include the Vostok data in the composite.

The composite dataset confirms the millenial-scale variations shown in the data from this study (Figure 2). Although none of the individual additional datasets is of high enough resolution to show millenial-scale variations with accuracy, when aligned to our data the new data follow the millenial-scale variations with very few outliers.

Finally, the uncertainty with respect to the absolute $CO_2$ value should be noted. The offsets between the multiple datasets are in large part likely due to differences in extraction efficiency between the measurement methods. The sublimation and ring mill systems have high extraction efficiency on clathrates, and should therefore present more unbiased baseline $CO_2$ values. However, since these datasets are as of now incomplete, we have aligned all datasets to the baseline absolute value of our ball mill dataset, and the absolute $CO_2$ values are reported within an uncertainty of ~5 ppm. We emphasize that the conclusions in this paper are only made with respect to relative values, and absolute values are only considered within their uncertainties.

**Table 1:** Existing $CO_2$ data sets from EDC and Vostok ice core and new $CO_2$ data from EDC during MIS 6.

| Ice core | Method (Reference) | $CO_2$ difference with $CO_2$ from EDC by ball mill in this study (ppm) | Contamination correction | Number of replicates | Number of sample |
|---|---|---|---|---|---|
| EDC | Sublimation at CEP Schneider et al. (2013) | 4.7± 1.7 (1σ) | O | 2–5 | 14 |
| | Ball mill at IGE Lourantou et al. (2010) | 2.4±2.1 (1σ) | X | 1 | 11 |
| | Ring mill at IGE (In this study) | 8.2±1.1 (1σ) | O | 1 | 11 |
| | Needle cracker at CEP (In this study) | 7.8± 1.1 (1σ) | O | 2–4 | 35 |
| | CIM at CEP (In this study) | 5.4± 1.0 (1σ) | O | 2–4 | 26 |
| Vostok | Ball mill at CEP Petit et al. (1999) | 4.6± 3.0 (1σ) | X | 1 | 49 |

This is written to new section 3.2 Data verification.

2) The authors state that they "make use of contrasting boundary conditions during the last two glacial periods to gain insight into the co-occurring carbon cycle changes." They then note that those boundary conditions are only "slightly different", and in the end, never explain what those differences are or why they might be expected to matter. This undermines the rationale for proceeding with this study and should be much better explained in a revised manuscript.

P3L1-P3L11 revised to: Comparing $CO_2$ changes on millennial time scales during the past two glacial periods, MIS 3 (MIS 3, 60–27 kyr BP) and early MIS 6 (early MIS 6, 185–160 kyr BP) can provide us with a better understanding of the carbon mechanisms at work, due to the similarities but also differences of climate conditions and events during the last two glacial periods (see Figure S1 in SI (Supplement Information)). Proxy evidence indicates that the states of several important components of the climate-carbon cycle were not entirely analogous in MIS 3 and MIS 6. Sea ice cover in the South Atlantic was more extensive during MIS 6, and sea surface temperature in the South Atlantic is thought to have been lower (Gottschalk et al., 2020). The bipolar see-saw phenomenon also has been observed to be active during the early MIS 6 period (Cheng et al., 2016; Jouzel et al., 2007; Margari et al., 2010). For example, the bipolar see-saw events during MIS 6 are longer than MIS 3. Events of iceberg discharge into the NA, which are thought to have driven millennial-scale changes in the meridional overturning circulation during MIS 3 (de Abreu et al., 2003; McManus et al., 1999) appear to be much more frequent during MIS 3 than during MIS 6. During the early MIS 6, iceberg discharge was muted (de Abreu et al., 2003; McManus et al., 1999). During the time period around 175 kyr BP, summer insolation levels in the Northern Hemisphere approached interglacial values (Berger, 1978). Due to the stronger Northern Hemisphere insolation, the intertropical convergence zone (ITCZ) is thought to have shifted northward, intensifying monsoon systems in low latitude regions, such as in Asia, the Appenine Peninsula and the Levant (Ayalon et al., 2002; Bard et al., 2002; Cheng et al., 2016). This may have led to a weaker overturning circulation due to the reduction of the density of the North Atlantic surface water, making the AMOC cell shallower during MIS 6 than during MIS 3 (Margari et al., 2010). Thus, this intensified hydrological cycle may help explain a shallower AMOC cell with a limited iceberg discharge in NA and the prolonged bipolar see-saw events during the early MIS 6 (Gottschalk et al., 2020; Margari et al., 2010).

3)   It appears that a potentially significant conclusion of the manuscript derives from the observations associated with a single small millennial event. This hardly seems justified and should be bolstered either by theoretical arguments or indications of similar behavior in existing data from another time interval.

Many similar events are present in MIS3 (see Figure 7 in the main text), bolstering our conclusion.

Revised to: We measured 150 samples of the EDC ice core to reconstruct atmospheric $CO_2$ during the MIS 6 period (189–135 kyr BP), with a high time resolution and an improved analytical precision. In this study, we investigate how different climate background conditions during the last two glacial periods impact atmospheric $CO_2$. Millennial-scale atmospheric $CO_2$ changes are found during both last two glacial periods, with amplitudes ranging between 15 to 25 ppm, mimicking similar trends in Antarctic $\delta D$ variations (Ahn and Brook, 2014; Bereiter et al., 2012). On the other hand, during short NA stadials which last less than 1,500 yrs, atmospheric $CO_2$ variations are negligible and decoupled with $\delta D$ in EDC. This finding suggest that during the last two glacial periods, the amplitude of $CO_2$ is highly determined by the NA stadial duration (r=0.83, n=20).

The division and labeling of sub-events is neither referenced nor adequately described, much less explained. Such division is understandable and can be helpful, but only if clearly delineated and consistently applied. Are the

divisions related to marine oxygen isotopes, and should they be, or to something else? What is the justification for 6c, 6d, and 6e, when there is no 6a or 6b?

Carbon Dioxide Maxima (CDM) are named according to the MIS timescale developed by Railsback et al. (2015). Figure 1 in the main text revised.

[Figure]

**Figure 1:** Proxy data during 250 kyr BP. A: Ice-rafted debris (IRD) input in the Iberian margin core MD95–2040 (de Abreu et al., 2003). B: 21 June insolation for 65°N (Berger, 1978). C: $\delta^{18}O_{calcite}$ from Sanbao cave, corresponding with the strength of the East Asian monsoon (Cheng et al., 2016). D: Dust flux in EDC (Lambert et al., 2012). E: Atmospheric $CH_4$ in EDC (green dots) (Loulergue et al., 2008) and Atmospheric $CH_4$ in EDC in this study (light yellow dots). F: Atmospheric $CO_2$ from EDC in this study (light blue dots) and composite $CO_2$ from Antarctic ice cores (dark blue dots) (Bereiter et al., 2015). G: $\delta D$ composition in EDC, Antarctica (Jouzel et al., 2007). Vertical grey bars indicate the timing of Heinrich events. The sub-stage numbers are written at the bottom (Railsback et al., 2015).

Smaller points to be considered and addressed:

Page 2 line 15 – "opposite behaviour"

10 Revised

Page 2 line 17 – Do the authors really infer that CO2 changes are "in response" to temperature?

Removed

Page 7 line 8 - The half cycle between minimum and maximum values is not the definition of an inflection point, nor is it any point at all.

Revised to: "The midpoint of the stadial transitions in both $\delta^{18}O$ of planktonic foraminifera and tree pollen in MD01–2444 were used to identify the NH stadial stadial transitions."

Page 14-15 – Data do appear to be limited, although Helmke, 2003, Kandiano, 2003, Obrochta 2014, Mokkeddem 2016, and Barker 2015 come to mind.

These datasets, while valuable for discussing MIS6, are too scattered to observe exact variations on millennial time scales and have larger age uncertainties with respect to EDC, making it complicated to use them to comment on the relationship between climate and prominent $CO_2$ variation during the early MIS 6 on millennial time scales.

Page 15, line 16 – "available"

Revised

Figure 3 – What are the "Six variations on millennial time scales: : :"?

Revised to: The durations of the six NA stadials during MIS 6 defined by Margari et al. (2010). Tree pollen percentage (top) and $\delta^{18}O$ of planktonic foraminifera (bottom) in the MD01-2444 (Margari et al., 2010). Red lines indicate the midpoint between the maximum and the preceding minimum of both $\delta^{18}O$ of planktonic foraminifera and tree pollen in MD01–2444. Blue bars indicate the range between the maximum and the preceding minimum.

Figure 5 – Golay should be capitalized in the legend.

Revised

**References**

Ahn, J. and Brook, E. J.: Siple Dome ice reveals two modes of millennial $CO_2$ change during the last ice age, Nat. Commun., 5, 3723, 2014.

Ayalon, A., Bar-Matthews, M., and Kaufman, A.: Climatic conditions during marine oxygen isotope stage 6 in the eastern Mediterranean region from the isotopic composition of speleothems of Soreq Cave, Israel, Geology, 30, 303-306, 2002.

Bard, E., Antonioli, F., and Silenzi, S.: Sea-level during the penultimate interglacial period based on a submerged stalagmite from Argentarola Cave (Italy), Earth Planet. Sci. Lett., 196, 135-146, 2002.

Bereiter, B., Lüthi, D., Siegrist, M., Schüpbach, S., Stocker, T. F., and Fischer, H.: Mode change of millennial $CO_2$ variability during the last glacial cycle associated with a bipolar marine carbon seesaw, Proc. Natl. Acad. Sci., 109, 9755-9760, 2012.

Berger, A. L.: Long-Term Variations of Caloric Insolation Resulting from the Earth's Orbital Elements 1, Quat. Res., 9, 139-167, 1978.

Cheng, H., Edwards, R. L., Sinha, A., Spötl, C., Yi, L., Chen, S., Kelly, M., Kathayat, G., Wang, X., and Li, X.: The Asian monsoon over the past 640,000 years and ice age terminations, Nature, 534, 640, 2016.

de Abreu, L., Shackleton, N. J., Schönfeld, J., Hall, M., and Chapman, M.: Millennial-scale oceanic climate variability off the Western Iberian margin during the last two glacial periods, Mar. Geol., 196, 1-20, 2003.

Gottschalk, J., Skinner, L. C., Jaccard, S. L., Menviel, L., Nehrbass-Ahles, C., and Waelbroeck, C.: Southern Ocean link between changes in atmospheric $CO_2$ levels and northern-hemisphere climate anomalies during the last two glacial periods, Quaternary Sci. Rev., 230, 106067, 2020.

Jouzel, J., Masson-Delmotte, V., Cattani, O., Dreyfus, G., Falourd, S., Hoffmann, G., Minster, B., Nouet, J., Barnola, J.-M., and Chappellaz, J.: Orbital and millennial Antarctic climate variability over the past 800,000 years, Science, 317, 793-796, 2007.

Margari, V., Skinner, L., Tzedakis, P., Ganopolski, A., Vautravers, M., and Shackleton, N.: The nature of millennial-scale climate variability during the past two glacial periods, Nat. Geosci., 3, 127, 2010.

McManus, J. F., Oppo, D. W., and Cullen, J. L.: A 0.5-million-year record of millennial-scale climate variability in the North Atlantic, science, 283, 971-975, 1999.

Railsback, L. B., Gibbard, P. L., Head, M. J., Voarintsoa, N. R. G., and Toucanne, S.: An optimized scheme of lettered marine isotope substages for the last 1.0 million years, and the climatostratigraphic nature of isotope stages and substages, Quaternary Science Reviews, 111, 94-106, 2015.

Lüthi, D., Bereiter, B., Stauffer, B., Winkler, R., Schwander, J., Kindler, P., Leuenberger, M., Kipfstuhl, S., Capron, E., and Landais, A.: $CO_2$ and $O_2/N_2$ variations in and just below the bubble–clathrate transformation zone of Antarctic ice cores, Earth and planetary science letters, 297, 226-233, 2010.

Railsback, L. B., Gibbard, P. L., Head, M. J., Voarintsoa, N. R. G., and Toucanne, S.: An optimized scheme of lettered marine isotope substages for the last 1.0 million years, and the climatostratigraphic nature of isotope stages and substages, Quaternary Science Reviews, 111, 94-106, 2015.

Schaefer, H., Lourantou, A., Chappellaz, J., Lüthi, D., Bereiter, B., and Barnola, J.-M.: On the suitability of partially clathrated ice for analysis of concentration and $\delta^{13}C$ of palaeo-atmospheric $CO_2$, Earth Planet. Sci. Lett., 307, 334-340, 2011.

---

## Author Comment (AC2) · 1 Jun 2020

Dear reviewers,

We are grateful for the detailed suggestions, and we believe that these suggestions will considerably improve our study. Below, we address the comments in blue and the revised texts in the manuscript in green. All the best, on behalf of all co-authors,

Jinhwa Shin

Please also note the supplement to this comment:

https://www.clim-past-discuss.net/cp-2019-142/cp-2019-142-AC2-supplement.pdf

[Figure]

**Supplement:**

**Dear reviewers,**

**We thank all four reviewers for their careful review of our paper, and their positive evaluation of the work. We appreciate their useful comments and believe their input has improved the paper. Below, we address the comments in blue and the revised texts in the manuscript in green.**

**All the best, on behalf of all co-authors,**
**Jinhwa Shin**

**Anonymous Referee #3**

This manuscript presents new and important atmospheric CO2 concentration data from the penultimate glacial period, also known as MIS6. The data concern so-called millennial-scale climate change, which has been well documented from Greenland ice cores. Because the Greenland ice cores do not extend back into MIS6, the natural archive in which to study millennial-scale climate for this period is Antarctic ice. The data appear to be of high quality and the discussion is appropriately oriented to the question of the temporal lag of peak CO2 behind millennial-scale warm intervals. The lag is found to be larger in the colder intervals than in the warmer intervals, much as was previously found for the more recent period of MIS 3 to MIS 5.

The one major thing I find lacking in this paper is replication of CO2 data points from the same depth in the ice core. Replication of gas measurements in ice cores is fundamental in order to have confidence in the accuracy of the data. Furthermore, the authors should calculate a pooled standard deviation from the means of replicates cut from the same depth in the ice core. This is widely viewed in the field as the most reliable indicator of the overall precision of the measurement, including potential issues arising from the ice itself (such as in-situ CO2 production).

Replicates account for differences between two ice samples at the same depth, making a better estimate of standard deviation of the final measurement but not necessarily of system precision itself. For example, Lüthi et al. (2010) show that there exists true small scale variability in $CO_2$ concentrations in the ice below the Bubble Clathrate Transition Zone, which could be accounted for by using replicates, especially for small sample sizes. Due to the diffusion effect, this small variation of atmospheric $CO_2$ is smoothed to some degree. In our study, large sample sizes (40g) of the ball mill system were used to reconstruct atmospheric $CO_2$, so a low-noise signal from the ice core is extracted (the smaller measurements used in other systems would be noisier in theory). The standard deviation of the measurement is estimated from the 5 injections, but system precision was calculated from blank measurements, which were performed after every 10 measurements accounting for the possible sources of $CO_2$ contamination with our analytical procedure.

To verify our new dataset, we made a composite data set using by aligning previous sets of measurements made over the MIS 6 period on the EDC ice core to our dataset. First, we compared to two existing $CO_2$ data sets and two new $CO_2$ data sets from EDC (Figure 1 and Table 1). There are two published $CO_2$ datasets for EDC during MIS6—the first measured by the ball mill system at IGE (Lourantou et al., 2010) and the second by the sublimation system at CEP (Schneider et al., 2013). We also compared unpublished atmospheric $CO_2$

measurements from EDC by a novel centrifugal ice microtome (CIM) system, a needle cracker and a ring mill system (Shin, 2019). All records are on the AICC2012 air age scale (Bazin et al., 2013). All data sets is corrected for the gravitational fractionation effect using the new $\delta^{15}N$ data in our study.

[Figure]

**Figure 1:** Atmospheric $CO_2$ from EDC and Vostok ice cores, compared to the $\delta D$ of water at EDC (temperature proxy) during 190—135 kyr BP. Blue dots: Atmospheric $CO_2$ from EDC by ball mill system (this study). Yellow dots: Atmospheric $CO_2$ from EDC by ball mill system (Lourantou et al., 2010). Purple dots: Atmospheric $CO_2$ from EDC by ring mill system. Red equilateral triangles: Atmospheric $CO_2$ from EDC by needle cracker. Black inverted triangles: Atmospheric $CO_2$ from EDC by CIM. Green rhombuses: Atmospheric $CO_2$ from EDC by sublimation. Grey dots: Atmospheric $CO_2$ from the Vostok ice core (Petit et al., 1999). Grey line: $\delta D$ of water at EDC (Jouzel et al., 2007).

Because of the limited amount of samples available, the data reconstructed by both ball mill and ring mill methods are single measurements from the depth interval. $CO_2$ records by CIM, needle cracker and the sublimation methods were reconstructed from 2–5 replicates from individual depth intervals. The error bars of data without replicate indicate that the standard deviation of five consecutive injections of the gas extracted from each sample into the gas chromatography (Lourantou et al., 2010; Petit et al., 1999). The error bars of data with replicate indicate the standard deviation of the mean of replicates from the same depth interval (Schneider et al., 2013). Figure 1 shows $CO_2$ concentrations measured by the ball mill system, the ring system, the sublimation, the CIM and the needle cracker. These $CO_2$ concentrations by the ball mill system (Lourantou et al., 2010), the ring system, the sublimation (Schneider et al., 2013), the CIM and the needle cracker are systematically higher than $CO_2$ concentrations measured by the ball mill system in our study (Table 1 and Figure 1). Atmospheric $CO_2$ during the MIS 6 period shows an offset between $CO_2$ data in this study and other $CO_2$ sets, which might be related with different analytical methods.

When the air is extracted from an ice core sample where bubble and clathrates co-exist, different dry extraction methods with different extraction efficiencies on bubbly and clathrate ice may lead to biased $CO_2$ concentrations

(Lüthi et al., 2010; Schaefer et al., 2011). During clathrate formation, the gas is partitioned into clathrates due to the different gas diffusivities and solubilities (Salamatin et al., 2001). $CO_2$ has consistently been observed to be depleted in bubbles and enriched in clathrates (Schaefer et al., 2011). Degassing from clathrates during extraction takes much longer than air release from bubbles; thus, if air from the clathrate ice is not extracted entirely, $CO_2$ measurement will be lower than the true value. The ball mill shows extraction efficiencies of ~62% for bubbles and ~52% for clathrates on average (Schaefer et al., 2011). If the ball mill is used to reconstruct $CO_2$ in Bubble–Clathrate Transformation Zone (BTCZ), $CO_2$ concentrations can be biased.

**Table 1:** Existing $CO_2$ data sets from EDC and Vostok ice core and new $CO_2$ data from EDC during MIS 6.

| Ice core | Method (Reference) | $CO_2$ difference with $CO_2$ from EDC by ball mill in this study (ppm) | Contamination correction | Number of replicates | Number of sample |
|---|---|---|---|---|---|
| EDC | Sublimation at CEP Schneider et al. (2013) | 4.7± 1.7 (1σ) | O | 2–5 | 14 |
| | Ball mill at IGE Lourantou et al. (2010) | 2.4±2.1 (1σ) | X | 1 | 11 |
| | Ring mill at IGE (In this study) | 8.2±1.1 (1σ) | O | 1 | 11 |
| | Needle cracker at CEP (In this study) | 7.8± 1.1 (1σ) | O | 2–4 | 35 |
| | CIM at CEP (In this study) | 5.4± 1.0 (1σ) | O | 2–4 | 26 |
| Vostok | Ball mill at CEP Petit et al. (1999) | 4.6± 3.0 (1σ) | X | 1 | 49 |

$CO_2$ concentrations from EDC were reconstructed from 150 depth intervals that cover 2036.7 to 1787.5 m along the EDC ice core, which consist of clathrate ice. There exists true small scale variability in $CO_2$ concentrations in the ice below the Clathrate Zone (Lüthi et al., 2010). Due to the diffusion effect, this small variation of atmospheric $CO_2$ is smoothed. Thus, $CO_2$ concentrations in these depth intervals might represent the initial mean atmospheric concentration. However, the EDC ice core for MIS 6 was drilled in 1999 and, the ice core has been stored for ~20 years in cold rooms at -22.5 ± 2.5°C before the gas is analysed. More than 50% of the initial hydrates present in the freshly drilled ice may have been decomposed and transformed into secondary bubbles, or gas cavities (Lipenkov, *Pers. Comm.*). We expect the same fractionation as during the clathrate formation process, hence bubble would be depleted in $CO_2$. Thus, $CO_2$ concentrations from EDC may be lower. In addition, different analytical methods can cause $CO_2$ offsets.

In our study, we concentrate on the relative millennial changes of $CO_2$, which are confirmed by all of the EDC $CO_2$ records available so far. Thus, our conclusion in this paper are independent which absolute mean $CO_2$ level is correct. As the new data in this study are currently the best quality data in terms of repeatability, we use our

new data as reference record and correct for any inter-core offsets (see Figure 2 and Figure 3). We, however, state explicitly in the text that the absolute mean $CO_2$ level during MIS6 is not known better than 5 ppm.

In order to estimate these offsets while accurately accounting for both measurement uncertainty and uncertainty in the offsets themselves, we rely on a Monte Carlo procedure, which is run for 1000 iterations. At each iteration, the data from all datasets is resampled within its measurement uncertainty. Then, a Savitsky-Golay filter with an approximate cutoff period of 150 years (using a 7-point sliding window and cubic fit, sampled at 250-year resolution) is applied to the new EDC data from this study. The offsets between each additional dataset and our data are calculated.

In order to test the sensitivity of the stack to the interpolation methods, Monte Carlo procedures were also run using linear interpolation, cubic spline filtering, and enting spline filtering in place of the Savitsky-Golay filter. The mean calculated offsets did not vary by more than 0.2 ppm depending on the method, well within the uncertainty ranges calculated for the offsets themselves. At the end of the stochastic procedure, mean and standard deviations of each offset are calculated, and used to adjust each dataset to create the composite.

[Figure]

**Figure 2:** Atmospheric $CO_2$ from EDC and Vostok ice cores, compared to the $\delta D$ of water at EDC (temperature proxy) during 190—135 kyr BP. Blue dots: Atmospheric $CO_2$ from EDC by ball mill system (this study). Yellow dots: Atmospheric $CO_2$ from EDC by ball mill system (Lourantou et al., 2010). Purple dots: Atmospheric $CO_2$ from EDC by ring mill system. Red equilateral triangles: Atmospheric $CO_2$ from EDC by needle cracker. Black inverted triangles: Atmospheric $CO_2$ from EDC by CIM. Green rhombuses: Atmospheric $CO_2$ from EDC by sublimation. Grey dots: Atmospheric $CO_2$ from the Vostok ice core (Petit et al., 1999). Grey line: $\delta D$ of water at EDC (Jouzel et al., 2007).

There are two main sources of uncertainty in the composite dataset, the measurement uncertainty of the data and the uncertainty of the offset itself. The offset uncertainty is not independent for each point--rather, since the offsets

appear to be approximately constant, the offset uncertainty should apply to all points together (or at least present very high covariance). Therefore, these two sources of uncertainty are presented separately, and not aggregated.

We also use this procedure to estimate an offset between our data and the data measured on the Vostok ice core. However, this offset does appear to evolve over time, changing during late MIS 6. Additionally, uncertainties in the alignment of the Vostok and EDC age scales over MIS 6 make it unclear if the variations in the two data series are indeed contemporaneous. We therefore do not include the Vostok data in the composite.

[Figure]

**Figure 3:** A composite $CO_2$ from EDC and Vostok ice cores, compared to the $\delta D$ of water at EDC (temperature proxy) during 190—135 kyr BP.

The composite dataset confirms the millennial-scale variations shown in the data from this study (Figure 2 and Figure 3). Although none of the individual additional datasets is of high enough resolution to show millennial-scale variations with accuracy, when aligned to our data the new data follow the millennial-scale variations with very few outliers.

Finally, the uncertainty with respect to the absolute $CO_2$ value should be noted. The offsets between the multiple datasets are in large part likely due to differences in extraction efficiency between the measurement methods. The sublimation and ring mill systems have high extraction efficiency on clathrates, and should therefore present more unbiased baseline $CO_2$ values. However, since these datasets are as of now incomplete, we have aligned all datasets to the baseline absolute value of our ball mill dataset, and the absolute $CO_2$ values are reported within an uncertainty of ~5 ppm. We emphasize that the conclusions in this paper are only made with respect to relative values, and absolute values are only considered within their uncertainties.

As the new data set measured in this study provides the best record in terms of repeatability of the $CO_2$ measurements for the time interval of MIS 6, we use it as reference data set to homogenize all the individual $CO_2$ reconstructions from different cores. To this end we used a low-pass filtered version of new data from this study and calculated the residuals of each individual other $CO_2$ data set to this spline. To correct that data set, we used a constant offset that minimizes the root mean square error relative to this spline. Note that while this methods

finds an optimum homogenization of the data sets given their scatter and potential cross-dating issues, it does not make a statement of the correct absolute level of the homogenized data set, as all data sets are equally likely to be correct in their absolute level. As we are only interested in the relative variations over MIS 6 in our study, this has no impact on our conclusions.

P2L24─P3L19 written to the revised manuscript in Section 3.1 The new high-resolution and high precision $CO_2$ record during MIS 6.

P1L23─P2L23 and P3L20─P6L4 written to the revised manuscript in new section 3.2 Data verification.

10 It is now well known that bacteria living in the ice can and do produce CO2. The only question is, how much? So it is absolutely essential to replicate CO2 analyses on pieces of ice cut from the same depth (and therefore presumably the same age, and having been exposed to the same atmospheric gas concentrations).

$CO_2$ records can be contaminated by the in-situ production of $CO_2$ caused by carbonate-acid reactions and oxidation of organic molecules, which are mostly observed in Greenland ice cores. This is because of higher

15 values of impurities such as $Ca^{2+}$, hydrogen peroxide $H_2O_2$ and formaldehyde HCHO in Greenland ice cores. These impurities can cause carbonate-acid reactions and the oxidation of organic carbon, leading to large scattering of atmospheric $CO_2$ data.

Thus to obtain less in situ $CO_2$ production in ice, a low carbonate concentration and $H_2O_2$ in an ice core are important. Luckily, Antarctic ice cores have relatively low concentrations of $H_2O_2$ and carbonates and have low

20 temperature compared to Greenlandic ice cores, which reduces the risk of $CO_2$ contamination (Tschumi and Stauffer, 2000). It is estimated that the in-situ production of $CO_2$ for Antarctic ice cores is smaller than 1.5 ppm (Bereiter et al., 2009). Thus, in-situ production of $CO_2$ cannot be ruled out but the effect should not greatly impact our main observations. In contrast, the observed offsets (see comments above) can be explained by the combination of clathratization/relaxation processes and incomplete extraction efficiencies of the various methods

25 used. Accordingly, we refrain from discussing a potential in situ production issue in our manuscript.

Therefore the authors must return to the laboratory and measure essentially another 150 pieces of ice, before this manuscript can be published in CP. The authors must also quote their value they have found for the pooled standard deviation.

I also did not notice any mention of the number of samples that were rejected (but perhaps I just missed it). The

30 authors must mention this number clearly in the main text (not in the Supplement).

 2 data points were identified for which experimental error could not be ruled out, so we did not include these 2 points in this study. Except for these two points, data was not rejected.

Another problem with the manuscript as it stands is the large amount of speculation in the discussion. This doesn't add to the value of the paper and can be mostly cut out, or clearly labelled as speculation in the text.

Due to the lack of existing proxy data with high temporal resolution and high precision and modelling studies, explanations of carbon cycle mechanisms during MIS 6 are limited. However, hypotheses of these mechanisms have been presented by previous studies, and the continued discussion of these hypotheses and how our new observations may redirect the discussion, even if the very limited amount of data means that this discussion is speculative in nature, is important. We hope that this discussion will be helpful for future studies, and have made sure, as suggested by the reviewer, to clearly label any speculative discussion in the text.

**References**

Bazin, L., Landais, A., Lemieux-Dudon, B., Kele, H. T. M., Veres, D., Parrenin, F., Martinerie, P., Ritz, C., Capron, E., and Lipenkov, V.: An optimized multi-proxy, multi-site Antarctic ice and gas orbital chronology (AICC2012): 120-800 ka, Clim. Past, 9, 1715-1731, 2013.

Bereiter, B., Schwander, J., Lüthi, D., and Stocker, T. F.: Change in $CO_2$ concentration and $O_2/N_2$ ratio in ice cores due to molecular diffusion, Geophys. Res. Lett., 36, 2009.

Lourantou, A., Chappellaz, J., Barnola, J.-M., Masson-Delmotte, V., and Raynaud, D.: Changes in atmospheric $CO_2$ and its carbon isotopic ratio during the penultimate deglaciation, Quaternary Sci. Rev., 29, 1983-1992, 2010.

Lüthi, D., Bereiter, B., Stauffer, B., Winkler, R., Schwander, J., Kindler, P., Leuenberger, M., Kipfstuhl, S., Capron, E., and Landais, A.: $CO_2$ and $O_2/N_2$ variations in and just below the bubble–clathrate transformation zone of Antarctic ice cores, Earth and planetary science letters, 297, 226-233, 2010.

Petit, J.-R., Jouzel, J., Raynaud, D., Barkov, N. I., Barnola, J.-M., Basile, I., Bender, M., Chappellaz, J., Davis, M., and Delaygue, G.: Climate and atmospheric history of the past 420,000 years from the Vostok ice core, Antarctica, Nature, 399, 429, 1999.

Schneider, R., Schmitt, J., Köhler, P., Joos, F., and Fischer, H.: A reconstruction of atmospheric carbon dioxide and its stable carbon isotopic composition from the penultimate glacial maximum to the last glacial inception, Climate of the Past, 9, 2507-2523, 2013.

Shin, J.: Millennial-scale atmospheric $CO_2$ variations during the Marine Isotope Stage 6, 2019. Grenoble Alpes, 2019.

Lüthi, D., Bereiter, B., Stauffer, B., Winkler, R., Schwander, J., Kindler, P., Leuenberger, M., Kipfstuhl, S., Capron, E., and Landais, A.: $CO_2$ and $O_2/N_2$ variations in and just below the bubble–clathrate transformation zone of Antarctic ice cores, Earth and planetary science letters, 297, 226-233, 2010.

Schaefer, H., Lourantou, A., Chappellaz, J., Lüthi, D., Bereiter, B., and Barnola, J.-M.: On the suitability of partially clathrated ice for analysis of concentration and $\delta^{13}C$ of palaeo-atmospheric $CO_2$, Earth Planet. Sci. Lett., 307, 334-340, 2011.

---

## Author Comment (AC3) · 1 Jun 2020

Dear reviewers,

We are grateful for the detailed suggestions, and we believe that these suggestions will considerably improve our study. Below, we address the comments in blue and the revised texts in the manuscript in green. All the best, on behalf of all co-authors,

Jinhwa Shin

Please also note the supplement to this comment:

[Figure]

https://www.clim-past-discuss.net/cp-2019-142/cp-2019-142-AC3-supplement.pdf

[Figure]

**Supplement:**

Dear reviewers,

We are grateful for the detailed suggestions, and we believe that these suggestions will considerably improve our study. Below, we address the comments in blue and the revised texts in the manuscript in green.

All the best, on behalf of all co-authors,

Jinhwa Shin

10 **Anonymous Referee #2**

Shin et al. present new records of ice core CO2, CH4, and d15N from the EPICA dome C core during MIS 6. They show that the CO2 maxima tend to lag Antarctic d18O maxima, by an amount that increases thought the glacial period. The magnitude of CO2 increase scales with the duration of North Atlantic stadial period, suggesting

15 a key role of AMOC variations in millennial-scale CO2 variability.

The data and analysis are obviously of interest to the broader paleoclimate community, and this paper should be published with only minor corrections. I believe that the manuscript can be clarified in some places. The main conclusion seems to be that MIS6 behaves very similarly to the last glacial period (as expected). Owing to this similarity, the discussion is somewhat long and involves a lot of speculation – much of which has already been

20 said in earlier work (for example Bereiter et al. 2012).

Throughout the paper the authors present speculative climatic mechanisms, or uncorroborated results from individual model simulations as established fact. One example: "Due to the reduction of Summer Monsoon intensity in East Asia, salinity at the surface of the Pacific Ocean is increased. Thus, AABW and North Pacific Deep Water (NPDW) transport is enhanced (Menviel et al., 2014). Enhanced NPDW transport ventilates deep

25 Pacific carbon via the Southern Ocean which may lead to atmospheric CO2 increases." While this is not a bad description of Menviel 2014, I think this would be better presented with some caution because while possibly correct, this is in no way a consensus view.

This paragraph (P11L12-P11L22) removed.

30 Throughout the paper, the authors compare MIS 3 and MIS6. In several places the authors write that MIS3 and MIS6 had different "background conditions". I am not sure what is meant by that. In what way are they really different? Both periods represent a range of orbital conditions, sea ice volumes, ITCZ positions, Heinrich events etc. So there are many places where they are very similar. I would advise the authors remove this idea that these two glacial periods are somehow very different – I don't think they have made the case that they are (and their

35 data surely suggest that the carbon cycle responds in a very similar manner).

We do not claim that the last two glacial periods are "very different" as the reviewer states, but rather slightly different as also stated by Margari et al., 2010 and Gottschalk et al. (2020). This is an important distinction, which we attempt to clarify in the revision below. Our analysis focuses on whether these slight differences can impact

the variability of $CO_2$ on millennial time scales. This is of course already known to be the case for periods presenting more marked differences in background climatic conditions.

P2L31-P3L11 Revised to: Comparing $CO_2$ changes on millennial time scales during the past two glacial periods, MIS 3 (MIS 3, 60–27 kyr BP) and early MIS 6 (early MIS 6, 185–160 kyr BP) can provide us with a better understanding of the carbon mechanisms at work, due to the similarities but also differences in climate conditions and events during the last two glacial periods (see Figure S1 in SI (Supplement Information)). Proxy evidence indicates that several important components of the climate-carbon cycle were not entirely analogous between MIS 3 and MIS 6. Sea ice cover in the South Atlantic was more extensive during MIS 6, and sea surface temperature in the South Atlantic is thought to have been lower (Gottschalk et al., 2020). The bipolar see-saw phenomenon also has been observed to be active during the early MIS 6 period (Cheng et al., 2016; Jouzel et al., 2007; Margari et al., 2010). However, the bipolar see-saw events during MIS 6 are longer than MIS 3. Events of iceberg discharge into the NA, which are thought to have driven millennial-scale changes in the meridional overturning circulation during MIS 3 (de Abreu et al., 2003; McManus et al., 1999) appear to be much more frequent during MIS 3 than during MIS 6. During the early MIS 6, iceberg discharge was muted (de Abreu et al., 2003; McManus et al., 1999),.

During the time period around 175 kyr BP, summer insolation levels in the Northern Hemisphere approached interglacial values (Berger, 1978). Due to the stronger Northern Hemisphere insolation, the intertropical convergence zone (ITCZ) is thought to have shifted northward, intensifying monsoon systems in low latitude regions, such as in Asia, the Appenine Peninsula and the Levant (Ayalon et al., 2002; Bard et al., 2002; Cheng et al., 2016). This may have led to a weaker overturning circulation due to the reduction of the density of the North Atlantic surface water, making the AMOC cell shallower during MIS 6 than during MIS 3 (Margari et al., 2010). Thus, this intensified hydrological cycle may help explain a shallower AMOC cell with a limited iceberg discharge in NA and the prolonged bipolar see-saw events during the early MIS 6 (Gottschalk et al., 2020; Margari et al., 2010).

Due to the historical convention the last ice age is actually MIS2-4, rather than just MIS3. So a more meaningful comparison would be MIS2-4 to MIS6. Also, the authors also include MIS5 in their analysis (Fig. 8). I think the paper would be a lot simpler if the authors just claim to be studying millennial-scale CO2 variability, rather than focus on Marine Isotope Stage distinctions that may not be relevant.

The last glacial period covers MIS 2 to MIS 4. MIS 2 and 4 are full glacial periods but MIS 3 is an interstadial period, i.e. a less cold period during a glacial period.
MIS 6 covers the penultimate glacial period, and can be divided into 3 parts according to the magnitude of climate variability and climate characteristics observed in proxy data (Margari et al., 2014): early (185.2−157.7 kyr BP), transition (157.7−151 kyr BP) and late MIS 6 (151−135 kyr BP). Each part shows similarities to a specific period of the last Ice Age. Climate change on millennial time scales during the late MIS 6 (the penultimate glacial maximum) is subdued, similar to MIS 2 (the last glacial maximum). Climate variations on millennial time scales during the earlier MIS 6 (185-157 kyr BP) are more prominent, similar to those during MIS 3. Accordingly, similarities and differences of climate variations during MIS 3 and the earlier MIS 6 were chosen to understand similarities/differences in atmospheric $CO_2$ variations on millennial time scale during the past two glacial periods.

MIS 5, the interglacial period, was mentioned as a reference for our analysis of lags of $CO_2$ variations with respect to Northern Hemisphere warming. The paper by Bereiter et al. (2012) shows two modes of atmospheric $CO_2$ variations on millennial time scales with respect to abrupt warming in NA during MIS 3 and MIS 5. These modes might be caused by different configurations of oceanic circulation during MIS 5 and 3. We similarly observed two modes of lags of $CO_2$ variations with respect to abrupt warming in NA during MIS 6.

In all figures I would appreciate a more clear demarcation of the sub-sections. I am not very familiar with the MIS6a-6e definitions. Do they follow precession/Benthic sequences like in MIS5, and who has defined these? We used the MIS timescale developed by Railsback et al. (2015).

Could you please add the MIS5 and MIS6 (and MIS7?) sub-stage numbering into figures 1, 4 and 5. Also, for consistency you should mark the H-events of stage 6 in Fig 1. Where does the event numbering 6.e1 etc. come from? I have seen alternative numberings elsewhere in the literature.

[Figure]

**Figure 1:** Proxy data during 250 kyr BP. A: Ice-rafted debris (IRD) input in the Iberian margin core MD95−2040 (de Abreu et al., 2003). B: 21 June insolation for 65°N (Berger, 1978). C: $\delta^{18}O_{calcite}$ from Sanbao cave, corresponding with the strength of the East Asian monsoon (Cheng et al., 2016). D: Dust flux in EDC (Lambert et al., 2012). E: Atmospheric $CH_4$ in EDC (green dots) (Loulergue et al., 2008) and Atmospheric $CH_4$ in EDC in this study (light yellow dots). F: Atmospheric $CO_2$ from EDC in this study (light blue dots) and composite $CO_2$ from Antarctic ice cores (dark blue dots) (Bereiter et al., 2015). G: $\delta D$ composition in EDC, Antarctica (Jouzel et al., 2007). Vertical grey bars indicate the timing of Heinrich events. The sub-stage numbers are written at the bottom (Railsback et al., 2015).

Carbon Dioxide Maxima (CDM) are named according to the MIS timescale developed by Railsback et al. (2015) (Figure 5). This sentence written to the Section, 3.5 Atmospheric CO2 variability on millennial time scale.

Could you please add the synthetic Greenland reconstruction from Barker et al. 2011 to Figure 3, to see how it compares?

Accepted

[Figure]

**Figure 3 (in the text).** The durations of the six NA stadials during MIS 6 defined by Margari et al. (2010). A: $\delta D$ composition of the EDC ice core (Jouzel et al., 2007). B: Greenland synthetic $\delta^{18}O$ composition of ice (Barker et al., 2011). C: Tree pollen percentage in the MD01-2444 (Margari et al., 2010) D: $\delta^{18}O$ of planktonic foraminifera in the MD01-2444 (Margari et al., 2010). Proxy data shown here are given on the AICC2012 age scale. Red lines indicate the midpoints of the stadial transition of both $\delta^{18}O$ of planktonic foraminifera and tree pollen in MD01–2444. Light green bars indicate the uncertainty of the duration of each stadial estimated as half of the temporal difference between maxima and minima of $\delta^{18}O$ of planktonic foraminifera. Red dots indicate minima and maxima of $\delta D$ composition of the EDC ice core selected in this study.

The authors do not address the CO2 offset between the records enough. It is up to 10 ppm with Vostok, which is quite large. They explain this as due to the blank correction, which is only around 1.7 ppm and therefore insufficient. Such offsets are seen more often in comparing CO2 from different cores, and may actually be in the ice. Can you explain the EDC CO2 offset between this work and Lourantou?

The ball mill system has a different extraction efficiency depending on the presence of bubbles and/or clathrates in the ice sample, which may cause an accuracy for reconstructing absolute mean $CO_2$ level. When the air is extracted from an ice core sample where bubble and clathrates co-exist, different dry extraction methods with different extraction efficiencies on bubbly and clathrate ice may lead to biased $CO_2$ concentrations (Lüthi et al., 2010; Schaefer et al., 2011b). During clathrate formation, the gas is partitioned into clathrates due to the different gas diffusivities and solubilities (Salamatin et al., 2001). $CO_2$ has consistently been observed to be depleted in bubbles and enriched in clathrates (Schaefer et al., 2011a). Degassing from clathrates during extraction takes much longer than air release from bubbles; thus, if air from the clathrate ice is not extracted entirely, $CO_2$ measurement will be lower than the true value.

The ball mill shows extraction efficiencies of ~62% for bubbles and ~52% for clathrates on average (Schaefer et al., 2011a). If the ball mill is used to reconstruct $CO_2$ in Bubble–Clathrate Transformation Zone (BTCZ), $CO_2$ concentrations can be biased.

$CO_2$ concentrations from EDC were reconstructed from 150 depth intervals that cover 2036.7 to 1787.5 m along the EDC ice core, which consist of clathrate ice. There exists true small scale variability in $CO_2$ concentrations in the ice below the Clathrate Zone (Lüthi et al., 2010). Due to the diffusion effect, this small variation of atmospheric $CO_2$ is smoothed. Thus, $CO_2$ concentrations in these depth intervals might represent the initial mean atmospheric concentration. However, the EDC ice core for MIS 6 was drilled in 1999 and, the ice core has been stored for ~20 years in cold rooms at $-22.5 \pm 2.5°C$ before the gas is analysed. More than 50% of the initial hydrates present in the freshly drilled ice may have been decomposed and transformed into secondary bubbles, or gas cavities (Lipenkov, *Pers. Comm.*). We expect the same fractionation as during the clathrate formation process, hence bubbles would be depleted in $CO_2$. Thus, $CO_2$ concentrations from EDC may be lower. The portion of the Vostok ice core covering MIS 6 is also clathrate ice, but it was drilled in 1998 and measured immediately (Petit et al., 1999), and less clathrates may have transformed into secondary bubbles. Thus $CO_2$ concentrations from Vostok during MIS 6 may be higher and potentially reflect the true atmospheric concentration more closely. In our study we concentrate on the relative millennial changes of $CO_2$ around the mean glacial concentration, which are the same in all the $CO_2$ records available so far, Thus, our conclusion in this paper are independent of which absolute mean $CO_2$ level is correct. As the new data in this study are currently the best quality data in terms of repeatability, we use our new data as the reference record and correct for any inter-core offsets. We, however, state explicitly in the text that the absolute mean $CO_2$ level during MIS6 is not known better than 5 ppm.

The estimated offset between the existing $CO_2$ dataset from EDC by Lourantou et al. (2010) and our new dataset is ~2.4±2.1 ppm. The $CO_2$ data from EDC by Lourantou et al. (2010) were also reconstructed using the ball mill system. However, this dataset was not corrected for the $CO_2$ contamination caused by the analytical procedure. We estimated the level of $CO_2$ contamination to be between 1 and 2 ppm for our study. Considering that the previous dataset was not corrected for, the offset between the two data sets is small when compared to their uncertainties.

This is written to Section 3.1 The new high-resolution and high precision $CO_2$ record during MIS 6

Specific line-by-line comments:

P1L16: I don't think you can argue that the background conditions are different. That hasn't been established.
The detailed information about background conditions during the last two glacial periods is re-written for greater clarity on Page 3. We already mentioned before, please see the response to first question on Page 2 in this document.

P2L11: Broecker does not talk about the bipolar seesaw, but a seesaw in deepwater formation. Other references to consider are Blunier & Brook (2001); Pedro et al. (2018).
Blunier & Brook (2001); Pedro et al. (2018) added, Broecker, 1998 removed

P3L6: Normally a stronger monsoon is not associated with a weaker AMOC. How does this work?

This paragraph re-written.  Please see the response to the first question on Page 2 in this document.

P6L9: Add or replace with Etheridge et al (1992); this idea is much older.

Added

P5L5: The "assumption" that the bipolar seesaw was present is a pretty obvious one, and I don't think it needs to be questioned. My personal choice would have been to use Antarctic isotopes to define the stadials and interstadials (see e.g. Kawamura et al., 2017), rather than NA sediments that have much poorer age control.

[Figure]

**Figure 3 (in the text):** The durations of the six NA stadials during MIS 6 defined by Margari et al. (2010). A: δD composition of the EDC ice core (Jouzel et al., 2007). B: Greenland synthetic $\delta^{18}O$ composition of ice  (Barker et al., 2011). C: Tree pollen percentage in the MD01-2444 (Margari et al., 2010) D: $\delta^{18}O$ of planktonic foraminifera in the MD01-2444 (Margari et al., 2010). Proxy data shown here is on the AICC2012 age scale. Red lines indicate the midpoint between the midpoints of the stadial transition of both $\delta^{18}O$ of planktonic foraminifera and tree pollen in MD01–2444. Light green bars indicate the uncertainty of the duration of each stadial estimated as half of the temporal difference between maxima and minima of $\delta^{18}O$ of planktonic foraminifera. Red dots indicate minima and maxima of δD composition of the EDC ice core selected in this study.

In our study, the durations of the six NA stadials were originally defined as the interval between the midpoints of the stadial transition of both $\delta^{18}O$ of planktonic foraminifera and tree pollen in MD01–2444 (C and D in figure 3) which was suggested by Margari et al. (2010). With this data we observed that the magnitude of atmospheric $CO_2$ change is generally correlated with the NA stadial duration (r=0.7, n=6) during the early MIS 6 period.

As the reviewer mentioned, not all of the stadial durations during MIS 6 are entirely clear using this method. As suggested by the reviewer, a synthetic Greenland $\delta^{18}O_{ice}$ record (Barker et al., 2011) and Antarctic (δD) variations in Antarctic ice core are plotted in Figure 3 as references, on the AICC2012 age scale. The interval between the maximum and the preceding minimum of δD in the EDC record can also be used to estimate the duration of the stadial transitions (Gottschalk et al., 2020; Margari et al., 2010). In most cases, the synthetic Greenland $\delta^{18}O_{ice}$

record and the interval between the maximum and the preceding minimum of δD in the EDC record confirm the definition of NA stadials selected by $\delta^{18}O$ of planktonic foraminifera in MD01–2444 and tree pollen in MD01–2444. However, the duration of the NA stadial in MIS 6d.2 is not clearly confirmed by Greenland $\delta^{18}O_{ice}$ and δD in the EDC (Figure 3 (in the text)).

5    We recalculated the durations of the six NA stadials using the interval between the stadial transitions as recorded in the EDC δD record (Gottschalk et al., 2020; Kawamura et al., 2017; Margari et al., 2010). Minima and maxima were selected by finding zero values in the second Savitsky–Golay filtered derivative of the data (the same method we used to pick minima and maxima of atmospheric $CO_2$; P9 in SI and Figure 1 in this text).

[Figure]

**Figure 1:** Temperature records from EDC during MIS 6. The black curve in both panels shows the Savitsky-Golay filtered δD series, and the blue curve shows the original data. (A) Red vertical lines mark inflection points. (B) Blue vertical lines show the minima and maxima, the blue shading illustrates the estimated uncertainties of their timing. The event numbers are written at the top.

The red dots and error bars on δD in the EDC record in Figure 3 of the main text show the estimated minima and maxima of temperature corresponding to stadial transitions using this method, along with their uncertainties. However, using this tool, durations of 6e.1 and 6e.2 are apparently overestimated due to ambiguity concerning of maximum in 6e.1 and minimum in 6e.2. Neither our method nor that of Margari et al. (2010) can be considered

15    absolutely correct. To account for the differences between the two methods, we took the stadial duration to be the mean of the duration estimated by both $\delta^{18}O$ of planktonic foraminifera and tree pollen in MD01–2444 and dD definitions. The correlation coefficient between the magnitude of atmospheric $CO_2$ change and the NA stadial duration remains high (r=0.93, n=6) during the early MIS 6 period.

20    This new calculation added to Section 2.6. Section 2.6 Definition of NA stadial duration re-written: Due to the absence of a Greenland temperature record for MIS 6, the durations of the six NA stadials were defined using $\delta^{18}O$ of planktonic foraminifera and tree pollen in MD01–2444, which reflect temperature variability in the NH (Margari et al., 2010). The midpoint of the stadial transitions in both $\delta^{18}O$ of planktonic foraminifera and tree pollen in MD01–2444 were used to identify the NH stadial stadial transitions. The time interval between two

25    stadial transition points were defined as the NA stadial duration. In this approach, small variations of the two

records may bias the calculation of the duration of short stadials in the NH. However, the average age difference between the durations identified using the two methods is only 205 years, which is less than the sampling resolution of MD01–2444 during MIS 6. The stadials identified for MIS 6 are shown in Figure 3. The uncertainty of the duration of each stadial was estimated as half of the temporal difference between maxima and minima of

5    $\delta^{18}O$ of planktonic foraminifera. However, not all of the the stadial durations during MIS 6 are entirely clear using this method. Figure 3 shows synthetic Greenland $\delta^{18}O_{ice}$ record (Barker et al., 2011) and Antarctic ($\delta D$) variations in Antarctic ice core on the AICC2012 age scale. The interval between the maximum and the preceding minimum of $\delta D$ in the EDC record can also be used to estimate the duration of the stadial transitions (Gottschalk et al., 2020; Margari et al., 2010). In most cases, the synthetic Greenland $\delta^{18}O_{ice}$ record and the interval between the maximum

10    and the preceding minimum of $\delta D$ in the EDC record confirm the definition of NA stadials selected by $\delta^{18}O$ of planktonic foraminifera in MD01–2444 and tree pollen in MD01–2444. However, the duration of the NA stadial in MIS 6d.2 is not clearly confirmed by Greenland $\delta^{18}O_{ice}$ and $\delta D$ in the EDC.

We recalculated the durations of the six NA stadials identified during the MIS 6 period were previously defined by Margari et al. (2010). Between the maximum and the preceding minimum of $\delta D$ in the EDC record is defined

15    as the stadial durations (Gottschalk et al., 2020; Kawamura et al., 2017; Margari et al., 2010). Minima and maxima were selected by finding zero values in the second Savitsky–Golay filtered derivative of the data (the same method we used to pick minima and maxima of atmospheric $CO_2$; Figure S11 in SI). The red dots and error bars on $\delta D$ in the EDC record in Figure 3 show the estimated minima and maxima of temperature corresponding to stadial transitions using this method, along with their uncertainties. However, using this tool, durations of 6e.1 and 6e.2

20    are apparently overestimated due to ambiguity concerning of maximum in 6e.1 and minimum in 6e.2. Neither our method nor that of Margari et al. (2010) can be considered absolutely correct. To account for the differences between the two methods, we took the stadial duration to be the mean of the duration estimated by both $\delta^{18}O$ of planktonic foraminifera and tree pollen in MD01–2444 and dD definitions (Table 2).

P7L31: the offsets persist in periods of stable CO2, suggesting there is more than chronological error going on.

25    Please discuss offsets between the cores.

Accepted, Re-written. We mentioned about $CO_2$ offset on Page 4 in this document. Please see Page 4 in this document.

P10L19: again, the link between monsoon and AMOC does not make sense to me

This paragraph re-written: These two CDM events occurred during MIS 6d, when iceberg discharge was muted

30    and the intertropical convergence zone (ITCZ) is thought to have shifted northward, intensifying monsoon systems in low latitude Northern Hemisphere regions, such as in Asia, the Appenine Peninsula and the Levant (Ayalon et al., 2002; Bard et al., 2002; Cheng et al., 2016). This shift may have led to a weaker overturning circulation due to the reduction of the density of the North Atlantic surface water, making the AMOC cell shallower with limited iceberg discharge in NA during MIS 6 as compared to MIS 3 (Margari et al., 2010). The two apparent $CO_2$ lag

35    timescales with respect to abrupt warming in NH during MIS 6 might be related to this difference.

P12L7: upwelling or ventilation /de-stratification?

Upwelling or ventilation

P12L15-16: Anderson does not cover MIS6, plus those records lack the resolution to investigate short stadials.

Two sentences removed: "During the short stadial in MIS 6 (AIM 6e.1) and the short stadials in MIS 3, the duration and strength of AMOC disruption are similar (Margari et al., 2010). This is supported by the marine proxy data for upwelling in the Southern Ocean which do not show strong variations during short stadials for both MIS periods (Anderson et al., 2009)."

5    P13L22: yet the $CO_2$ variations of MIS5 are larger than those in MIS3?

The sentence at P13L22 is about $CO_2$ outgassing from the Ocean.

P14L22: remove "unprecedented". Some ice core $CO_2$ records have decadal precision.

Removed

P21: Could you add the H-events of Stage 6 also (or perhaps an IRD record that spans the full period)? Could you

10    mark the MIS6a-6e substage numbering? (I am not familiar with this nomeclature).

[Figure]

**Figure 1:** Proxy data during 250 kyr BP. A: Ice-rafted debris (IRD) input in the Iberian margin core MD95–2040 (de Abreu et al., 2003). B: 21 June insolation for 65°N (Berger, 1978). C: $\delta^{18}O_{calcite}$ from Sanbao cave, corresponding with the strength of the East Asian monsoon (Cheng et al., 2016). D: Dust flux in EDC (Lambert et al., 2012). E: Atmospheric $CH_4$ in EDC (green dots) (Loulergue et al., 2008) and atmospheric $CH_4$ in EDC in this study (light yellow dots). F: Atmospheric $CO_2$ from EDC in this study (light blue dots) and composite $CO_2$ from Antarctic ice cores (dark blue dots) (Bereiter et al., 2015). G: $\delta D$ composition in EDC, Antarctica (Jouzel et al., 2007). Vertical grey bars indicate the timing of Heinrich events. The sub-stage numbers are written at the bottom (Railsback et al., 2015).

Although the IRD dataset added in Figure 1 does not cover the whole 250 kyr period shown in figure 1, it is in our opinion the most appropriate to show North Atlantic events during MIS 6, and is thus used for our analysis. We prefer to not include any additional IRD datasets in the figure to avoid confusion for the reader.

15

P26: could you add the DO onsets you infer from CH4 as vertical bands?

It would be better to add vertical lines for CDM because there are 6 variations of atmospheric $CO_2$ but we could find only three abrupt $CH_4$ increases indicating the onset of DO events with the $CH_4$ data set. Vertical lines for CDMs added in Figure 6.

P27: Why did you not add the Stage 5 events here?

In this study we focused on atmospheric $CO_2$ variations on millennial time scales during the past two glacial periods, thus, the stage 5 was not included in Figure 7. In section 3.4 and 4.2, the stage 5 is mentioned, but only to discuss the factors that can influence the lag of atmospheric $CO_2$ with respect to abrupt warming in NH.

**References**

[revised manuscript text omitted]

---

## Author Comment (AC4) · 1 Jun 2020

Dear reviewers,

We are grateful for the detailed suggestions, and we believe that these suggestions will considerably improve our study. Below, we address the comments in blue and the revised texts in the manuscript in green. All the best, on behalf of all co-authors,

Jinhwa Shin

Please also note the supplement to this comment:

[Figure]

https://www.clim-past-discuss.net/cp-2019-142/cp-2019-142-AC4-supplement.pdf

[Figure]

**Supplement:**

Dear reviewers,

We thank all four reviewers for their careful review of our paper, and their positive evaluation of the work. We appreciate their useful comments and believe their input has improved the paper. Below, we address the comments in blue and the revised texts in the manuscript in green.

All the best, on behalf of all co-authors,

Jinhwa Shin

**Anonymous Referee #1**

Review of manuscript cp-2019-142:

**General Comments:**

The manuscript by Shin and others presents new, high-resolution measurements of CO2, CH4, and δ15N in EDC ice core samples spanning the glacial period, MIS 6. The new data resolve millennial-scale variations in CO2 and CH4. The authors independently identified MIS 6 stadial durations in tree pollen % and planktonic δ18O in the Iberian Margin marine sediment core MD01-2444. The authors also revised the MIS 6 gas age chronology of the EDC ice core (previously AICC 2012) using new estimates of Δdepth from the δ15N data. The revised EDC age scale, along with the timing of climate variations observed in the sediment core, provides the authors with a temporal framework for understanding millennial-scale CO2 variations during the penultimate glacial period.

The authors specifically analyze the timing of the CO2 changes relative to changes in CH4, considered here a proxy for NH warming, identifying leads/ lags between the two records. They also discuss differences between the CO2 features in MIS 6 and analogous features that occurred in MIS 3. The authors also observe differences in the magnitudes of CO2 maxima during MIS 6. They identify a relationship between the amplitude of CO2 change and the duration of the preceding stadial event, offering the hypothesis that the amplitude of CO2 variations depends on the duration of AMOC perturbations. They also identify a shift in the lag of CO2 maxima from MIS 6e to MIS 6d and suggest that this may be due to a change in the organization of AMOC. This manuscript is well written, organized, and clearly presented, the science is in my opinion sound, and the new datasets represent important contributions that will be of interest to others in the field. The work is appropriate for the journal Climate of the Past, and I recommend this paper for publication after minor revisions. Below I list specific comments that, if addressed, will aid in the clarity of the paper and hopefully strengthen the analyses therein. I also list technical corrections below.

**Specific Comments:**

**INTRODUCTION**

– P3L9 – Can you provide a reference for the longer duration of stadials in early MIS 6?

Accepted. A reference (Margari et al., 2010) added in the text.

– P3L15 – There are more pre-existing $CO_2$ measurements from late MIS 6 besides those from Vostok (Lourantou et al., 2010; Schneider et al., 2013).

References added in P3L15.

**METHODS**

– P3L31 – Did the measured $CO_2$ concentration depend on the amount of air injected?

The $CO_2$ concentrations of 5 measurements were constant. The amount of air injected did not impact the $CO_2$ concentration.

(Presumably, the pressure in the sample loop depleted across the 5 individual injections. Was there a linearity effect?)

The $CO_2$ concentrations do not depend on the sample size, and for the 5 consecutive injections on the same sample we obtained a linear relationship between pressure and partial pressure of $CO_2$, i.e., concentrations did not change

– P4L18-19 – Was the amount of contamination in each chamber consistent from day to day?

Each chamber shows different contamination levels. We could measure 3-4 samples a day and blank tests were conducted every 10 measurements of natural ice. The amount of contamination in each chamber varied within ~0.5-2 ppm, averaging 1 ppm during the measurement, and no drifts were observed during the 2 months.

Did it depend on the length/ amount of crushing?

To avoid any effect of the duration/ amount of crushing and $CO_2$ contamination caused by crushing process on the measured CO2 concentration we kept the same length/ amount of crushing process both for real samples and standards over ice.

– Did you run replicate $CO_2$ measurements on ice samples from the same depths?

In this study, large sample sizes (40g) were used, and because limited ice samples were available, we could not make replicates. To ensure precise data, we controlled the precision of the system to around 1 ppm using gas standards over gas-free synthetic ice measurements and assume their variability to be the same as for true ice.

In my opinion, this would be a better estimate of the true system precision.

Replicates additionally account for any differences between two ice samples, making a better estimate of standard deviation of the final measurement but not necessarily of system precision itself. For example, Lüthi et al. (2010) show that there exists true small scale variability in $CO_2$ concentrations in the ice below the Bubble Clathrate Transition Zone. Due to the diffusion effect, this small variation of atmospheric $CO_2$ is smoothed to some degree. In our study, large sample sizes (40g) of the ball mill system were used to reconstruct atmospheric $CO_2$, so a low-noise signal from the ice core is extracted (the smaller measurements used in other systems would be noisier in theory). The standard deviation of the measurement is estimated from the 5 injections, but system precision, which is added to the uncertainty of the measurements, was calculated from the blank measurement, accounting for the possible sources of $CO_2$ contamination with our analytical procedure.

This is added to new section 3.2 Data verification

– P4L25 – Can you state briefly how the new, corrected CO2 record compares to the preexisting CO2 data?

The new $CO_2$ data from EDC are corrected for gravitational fractionation and contamination caused by the analytical process. The previous $CO_2$ measurements from Vostok ice core by ball mill system (Petit et al., 1999) from EDC measured by ball mill (Lourantou et al., 2010) were corrected for gravitational fractionation effect but not corrected the $CO_2$ contamination effect (Figure 4 in the manuscript). Additionally, the Vostok data use a less precise age scale.

We have included a discussion of the $CO_2$ offset between the new data set and existing data sets in detail in section 3.1.

– P4L32 – Can you state the precision of the CH4 measurements?

This is now included. The precision of the system was estimated at ~11 ppb on average.

– P5L3 – What do you think are possible reasons for the systematic offset? Please describe briefly.

A systematic offset of 6 ppb between IGE and CEP was observed (Loulergue et al., 2008). The offsets are due to differences in corrections for contamination caused by the analytical procedure, a systematic offset of 6 ppb between IGE and CEP was observed (Loulergue et al., 2008).

Revised to: The previous CH4 dataset (Loulergue et al., 2008) from EDC was produced at both IGE and Climate and Environmental Physics (CEP), Physics Institute, University of Bern, Switzerland. $CH_4$ measured at CEP is systematically higher than $CH_4$ measured at IGE by 6 ppb (Loulergue et al., 2008). The offsets are due to differences in corrections for contamination caused by the analytical procedure between the datasets. In order to produce a coherent dataset, 6 ppb were added to the data obtained at IGE (Loulergue et al., 2008).

– P6L20 – Figure S4 in the SI does not have a label to distinguish blue from red.

Revised

– P6L22 – I do not follow how Figure 2 supports the claim that the previous method was "relatively unbiased but not entirely exact."

Sorry, this was the wrong figure number. That was explained in Figure S3. Figure number revised to Figure S3.

– P6L13 – In Figure 3 it appears that the midpoints in the transitions are somewhat ambiguously defined. Sometimes they fall between a local max and min for d18O, sometimes for pollen %. The markers are chosen as midpoints between local maxima/minima, but sometimes it is unclear where those max/ min data points are. 6d.2, for example, could easily be shorter (i.e., it looks like the end marker at 174.2 ka could be defined at an older age). 6c.2 is a particularly ambiguously defined stadial – I do not see which maximum and minimum pair defines the older marker. Could you define the stadial durations more objectively? The ambiguity and subjectivity in picking the stadial transitions lead me to believe that they were defined while also considering the ice core data. That's not necessarily a bad thing, but perhaps you should just be forthright and show the gas data in Figure 3 along with the sediment core data.

In our study, the durations of the six NA stadials were originally defined as the interval between the midpoints of the stadial transition of both $\delta^{18}O$ of planktonic foraminifera and tree pollen in MD01–2444 (C and D in figure 3) which was suggested by Margari et al. (2010). With this data we observed that the magnitude of atmospheric $CO_2$ change is generally correlated with the NA stadial duration (r=0.7, n=6) during the early MIS 6 period.

As the reviewer mentioned, not all of the stadial durations during MIS 6 are entirely clear using this method. As suggested by the reviewer, a synthetic Greenland $\delta^{18}O_{ice}$ record (Barker et al., 2011) and $\delta D$ variations in Antarctic ice core are plotted in Figure 3 as references(AICC2012 age scale). The interval between the maximum and the preceding minimum of $\delta D$ in the EDC record can also be used to estimate the duration of the stadial transitions (Gottschalk et al., 2020; Margari et al., 2010). In most cases, the synthetic Greenland $\delta^{18}O_{ice}$ record and the interval between the maximum and the preceding minimum of $\delta D$ in the EDC record confirm the definition of NA stadials selected by $\delta^{18}O$ of planktonic foraminifera in MD01–2444 and tree pollen in MD01–2444. However, the duration of the NA stadial in MIS 6d.2 is not clearly confirmed by Greenland $\delta^{18}O_{ice}$ and $\delta D$ in the EDC (Figure 3).

[Figure]

**Figure 3 (in the manuscript):** The durations of the six NA stadials during MIS 6 defined by Margari et al. (2010). A: $\delta D$ composition of the EDC ice core (Jouzel et al., 2007). B: Greenland synthetic $\delta^{18}O$ composition of ice (Barker et al., 2011). C: Tree pollen percentage in the MD01-2444 (Margari et al., 2010) D: $\delta^{18}O$ of planktonic foraminifera in the MD01-2444 (Margari et al., 2010). Proxy data shown here are given on the AICC2012 age scale. Red lines indicate the midpoint between the midpoints of the stadial transition of both $\delta^{18}O$ of planktonic foraminifera and tree pollen in MD01–2444. Light green bars indicate the uncertainty of the duration of each stadial estimated as half of the temporal difference between maxima and minima of $\delta^{18}O$ of planktonic foraminifera. Red dots indicate minima and maxima of $\delta D$ composition of the EDC ice core selected in this study. The event numbers are written at the top.

We recalculated the durations of the six NA stadials using the interval between the stadial transitions as recorded in the EDC $\delta D$ record (Gottschalk et al., 2020; Kawamura et al., 2017; Margari et al., 2010). Minima and maxima were selected by finding zero values in the second Savitsky–Golay filtered derivative of the data (the same method we used to pick minima and maxima of atmospheric $CO_2$; P9 in SI and Figure 1 in this text).

The red dots and error bars on $\delta D$ in the EDC record in Figure 3 of the main text show the estimated minima and maxima of temperature corresponding to stadial transitions using this method, along with their uncertainties.

However, using this tool, durations of 6e.1 and 6e.2 are apparently overestimated due to ambiguity concerning the maximum in 6e.1 and minimum in 6e.2. Neither our method nor that of Margari et al. (2010) can be considered absolutely correct. To account for the differences between the two methods, we took the stadial duration to be the mean of the duration estimated by both $\delta^{18}O$ of planktonic foraminifera and tree pollen in MD01–2444 and dD definitions. The correlation coefficient between the magnitude of atmospheric $CO_2$ change and the NA stadial duration remains high (r=0.93, n=6) during the early MIS 6 period.

This new calculation was added in section 2.6

[Figure]

**Figure** 1**:** Temperature records from EDC during MIS 6. The black curve in both panels shows the Savitsky-Golay filtered δD series, and the blue curve shows the original data. (A) Red vertical lines mark inflection points. (B) Blue vertical lines show the minima and maxima, the blue shading illustrates the estimated uncertainties of their timing. The event numbers are written at the top.

**2.6 Definition of NA stadial duration is revised to:**

Due to the absence of a Greenland temperature record for MIS 6, the durations of the six NA stadials were defined using $\delta^{18}O$ of planktonic foraminifera and tree pollen in MD01–2444, which reflect temperature variability in the NH (Margari et al., 2010). The midpoint of the stadial transitions in both $\delta^{18}O$ of planktonic foraminifera and tree pollen in MD01–2444 were used to identify the NH stadial stadial transitions. The time interval between two stadial transition points were defined as the NA stadial duration. In this approach, small variations of the two records may bias the calculation of the duration of short stadials in the NH. However, the average age difference between the durations identified using the two methods is only 205 years, which is less than the sampling resolution of MD01–2444 during MIS 6. The stadials identified for MIS 6 are shown in Figure 3. The uncertainty of the duration of each stadial was estimated as half of the temporal difference between maxima and minima of $\delta^{18}O$ of planktonic foraminifera. However, not all of the the stadial durations during MIS 6 are entirely clear using this method. Figure 3 shows synthetic Greenland $\delta^{18}O_{ice}$ record (Barker et al., 2011) and Antarctic (δD) variations in Antarctic ice core on the AICC2012 age scale. The interval between the maximum and the preceding minimum of δD in the EDC record can also be used to estimate the duration of the stadial transitions (Gottschalk et al., 2020; Margari et al., 2010). In most cases, the synthetic Greenland $\delta^{18}O_{ice}$ record and the interval between the maximum

and the preceding minimum of δD in the EDC record confirm the definition of NA stadials selected by $\delta^{18}O$ of planktonic foraminifera in MD01–2444 and tree pollen in MD01–2444. However, the duration of the NA stadial in MIS 6d.2 is not clearly confirmed by Greenland $\delta^{18}O_{ice}$ and δD in the EDC. We recalculated the durations of the six NA stadials identified during the MIS 6 period were previously defined by Margari et al. (2010). Between the maximum and the preceding minimum of δD in the EDC record is defined as the stadial durations (Gottschalk et al., 2020; Kawamura et al., 2017; Margari et al., 2010). Minima and maxima were selected by finding zero values in the second Savitsky–Golay filtered derivative of the data (the same method we used to pick minima and maxima of atmospheric $CO_2$; Figure S11 in SI). The red dots and error bars on δD in the EDC record in Figure 3 show the estimated minima and maxima of temperature corresponding to stadial transitions using this method, along with their uncertainties. However, using this tool, durations of 6e.1 and 6e.2 are apparently overestimated due to ambiguity concerning of maximum in 6e.1 and minimum in 6e.2. Neither our method nor that of Margari et al. (2010) can be considered absolutely correct. To account for the differences between the two methods, we took the stadial duration to be the mean of the duration estimated by both $\delta^{18}O$ of planktonic foraminifera and tree pollen in MD01–2444 and dD definitions (Table 2).

**RESULTS**

– P8L3 – You should mention the known phenomenon of CO2 offsets between different ice cores (e.g. WAIS versus Law Dome). The co-author Christoph could certainly comment on this.

When the air is extracted from an ice core sample where bubble and clathrates co-exist, different dry extraction methods with different extraction efficiencies on bubbly and clathrate ice may lead to biased $CO_2$ concentrations (Lüthi et al., 2010; Schaefer et al., 2011). During clathrate formation, the gas is partitioned into clathrates due to the different gas diffusivities and solubilities (Salamatin et al., 2001). $CO_2$ has consistently been observed to be depleted in bubbles and enriched in clathrates (Schaefer et al., 2011). Degassing from clathrates during extraction takes much longer than air release from bubbles; thus, if air from the clathrate ice is not extracted entirely, $CO_2$ measurement will be lower than the true value.

The ball mill shows extraction efficiencies of ~62% for bubbles and ~52% for clathrates on average (Schaefer et al., 2011). If the ball mill is used to reconstruct $CO_2$ in Bubble–Clathrate Transformation Zone (BTCZ), $CO_2$ concentrations can be biased.

$CO_2$ concentrations from EDC were reconstructed from 150 depth intervals that cover 2036.7 to 1787.5 m along the EDC ice core, which consist of clathrate ice. There exists true small scale variability in $CO_2$ concentrations in the ice below the Clathrate Zone (Lüthi et al., 2010). Due to the diffusion effect, this small variation of atmospheric $CO_2$ is smoothed. Thus, $CO_2$ concentrations in these depth intervals might represent the initial mean atmospheric concentration. However, the EDC ice core for MIS 6 was drilled in 1999 and, the ice core has been stored for ~20 years in cold rooms at -22.5 ± 2.5°C before the gas is analysed. More than 50% of the initial hydrates present in the freshly drilled ice may have been decomposed and transformed into secondary bubbles, or gas cavities (Lipenkov, *Pers. Comm.*). We expect the same fractionation as during the clathrate formation process, hence bubbles would be depleted in $CO_2$. Thus, $CO_2$ concentrations from EDC may be lower. The portion of the Vostok ice core covering MIS 6 is also clathrate ice, but it was drilled in 1998 and measured immediately (Petit et al., 1999), and less clathrates may have transformed into secondary bubbles. Thus $CO_2$ concentrations from Vostok

during MIS 6 may be higher and potentially reflect the true atmospheric concentration more closely. In our study we concentrate on the relative millennial changes of $CO_2$ around the mean glacial concentration, which are the same in all the $CO_2$ records available so far, Thus, our conclusion in this paper are independent of which absolute mean $CO_2$ level is correct. As the new data in this study are currently the best quality data in terms of repeatability, we use our new data as the reference record and correct for any inter-core offsets. We, however, state explicitly in the text that the absolute mean $CO_2$ level during MIS6 is not known better than 5 ppm.

This is written to Section 3.1 The new high-resolution and high precision $CO_2$ record during MIS 6

– P25 Fig5 – It is unclear how the blue CDM events were defined. Do they relate somehow to the stadial duration markers you defined previously? If not, please clarify how you identified them (or provide proper reference to SI). Revised. It is explained in Table S1 and Figure S10. These references were added in Section 3.3.

– P26Fig6 – Shading or vertical lines would help to delineate the CDM's in Figure 6. Right now the text floats at the bottom and is unclear exactly what the labels refer to.

Revised

[Figure]

– One result that strikes me as interesting, and not discussed in the paper, is that the lowest CO2 and Antarctic temperature values occur in the early/ middle part of MIS 6, not the latest part (as in MIS 2). CH4, on the other hand, reaches the lowest values during late MIS 6, right before the termination, as does peak glaciation as inferred from the benthic d18O. This is unlike MIS 2, which is characterized by low CO2, low Antarctic temperature, low CH4, and peak glaciation occurring simultaneously. Can you speculate why CO2 is higher in late MIS 6 relative to earlier in MIS 6, despite full glacial extent?

A saturation index indicating variations in respired carbon content in the deep sub-Antarctic Atlantic (MD07-3077) and atmospheric $CO_2$ have been shown to be closely anti-correlated (Gottschalk et al., 2020). This observation indicates that the regulation of global atmospheric $CO_2$ variations on millennial time scales is highly influenced by the marine carbon cycle in the Southern Ocean (Fischer et al., 2010) during MIS 6.

As shown in this figure, atmospheric $CO_2$ from EDC is highly co-related with dust flux in EDC (Lambert et al., 2012), δD in EDC (Jouzel et al., 2007) and summer sea surface temperature in the deep sub-Antarctic Atlantic (MD07-3077) (Gottschalk et al., 2020). Iron Fertilization and temperature in the Southern Ocean can affect $CO_2$ variations on millennial time scales. However, the main difference of climate between late MIS 6 and early MIS 6 is temperature in the Southern Ocean. Colder conditions are observed in the Southern Ocean in early MIS 6 than in late MIS 6. Colder conditions in early MIS 6 would allow for more carbon uptake in the southern Ocean. Thus, the $CO_2$ level during the early MIS 6 might be slightly lower than the late MIS 6 due to colder ocean conditions during the early MIS 6. In contrast, $CH_4$ is reflecting primarily climate/hydrological conditions on land in the tropics and to a much smaller extent in high northern latitudes. Thus, a decoupling of the two parameters suggests different glacial climate evolution in high southern latitudes and the tropics.

(P8L1−P8L10 is written in the Section 3.2. the Section name is revised to "Relationship between the temperature in the Southern Ocean and atmospheric $CO_2$")

[Figure]

**Figure 2:** Climate proxies during MIS 6. Vertical blue dotted lines indicate the six CDM events that we identify during the early MIS 6. A: Dust flux in EDC (Lambert et al., 2012). B: EDC water isotopic record (Jouzel et al., 2007). C: Sea summer surface temperature in the deep sub-Antarctic Atlantic (MD07-3077) (Gottschalk et al., 2020). D: Saturation Index in the deep sub-Antarctic Atlantic (MD07-3077) (Gottschalk et al., 2020). E: Atmospheric $CO_2$ from EDC (this study). The red line indicates Savitsky Golay filtering curve made with a ~150 yr cut-off period (red dotted line).

− P28Fig8 - The authors compare the timing of CO2 maxima relative to the onset of NH warming. The CO2 measurements come from different ice cores with different age scales (to my knowledge at least, Byrd is not synchronized to the AICC 2012 as EDML, EDC, and TALDICE are). What is the bias or uncertainty in the analysis due to age offsets? Why not exclusively use the EPICA cores on a unified age scale for this analysis?

To calculate leads and lags between $CO_2$ and the abrupt warming in NH, we calculated the time lag for each CDM following abrupt warming events in the NH. In this study, given the fact that when temperature increases rapidly in Greenland, $CH_4$ increases rapidly within 50 yrs (Baumgartner et al., 2014; Rosen et al., 2014), we used $CH_4$ as a time marker of rapid warming in the NH.

CH$_4$ and CO$_2$ signals are both reconstructed from the air bubbles in the same ice, and as such there is no chronological uncertainty with respect to individual timings. The Byrd core was synchronized to the EDML core in the gas phase by Bereiter et al. (2012), and thus can be synchronized to the AICC2012 chronology as well. Without synchronization, there can be significant differences in event duration between two cores. However, with the synchronization between Byrd and EDML, these inconsistencies should be minimized. The measurements for each period are chosen to maximize resolution and minimize uncertainty related to gas trapping. –The estimation of the exact timing of CDM from the EDC ice core might be less accurate compared to that from the TALDICE ice core, for example, due to the narrower gas age distribution of TALDICE (Bereiter et al., 2012). The remaining uncertainty is related to analytical uncertainties and to the temporal resolution of the two records.

**DISCUSSION**

– P11L26&31 – When you say that the terrestrial biosphere can "compensate" for the slow response of the deep ocean, do you mean in terms of its timing or in terms of the direction of CO2 change?

The direction of CO$_2$ change.

This paragraph was revised to: As mentioned above, atmospheric CO$_2$ on millennial timescales can be controlled by CO$_2$ exchange between the ocean and the atmosphere, as well as changes of terrestrial carbon stocks. Coupled climate carbon cycle models reported that the variations of atmospheric CO$_2$ concentration on millennial timescales are mainly dominated by deep ocean inventory, requiring a few millennia to react to climate change (Schmittner and Galbraith, 2008). On the other hand, the response of the terrestrial biosphere is fast (centennial timescale) (Bouttes et al., 2012; Menviel et al., 2014; Schmittner and Galbraith, 2008). Although different models differ significantly in the CO$_2$ response to AMOC changes, the initial CO$_2$ evolution of the terrestrial biosphere and deep ocean to AMOC perturbations tends to be opposite in model simulations (Gottschalk et al., 2019). Thus, due to the opposite direction of CO$_2$ change of ocean and terrestrial reservoirs, atmospheric CO$_2$ variations might be muted if the stadial in the NA is short (Bouttes et al., 2012; Menviel et al., 2014; Schmittner and Galbraith, 2008). There is, on the other hand, evidence that not all of the processes of CO2 exchange follows these general trends. For example, atmospheric CO2 might be changed on centennial timescales by carbon exchange between the deep and surface ocean (Rae et al., 2018) or atmospheric CO2 might be influenced slowly by soil decomposition (Köhler et al., 2005).

Please clarify. "Compensate" may not be the best word to use in case it is confused with carbonate compensation.
Revised. Compensate was mentioned twice in the text at P11L27 and P11L32. The first one was changed to "muted" and the second one was changed to "be offset by"

– P13 – After the discussion of AMOC and deep ocean ventilation, I realized there was no discussion entertaining productivity fluctuations as a possible mechanism for millennial-scale CO2 variability (Ziegler et al., 2013; Gottschalk et al., 2016; Anderson et al., 2014; Martinez-Garcia et al., 2014).

The dust flux in EDC clearly shows millennial variations during MIS 6. The anti-correlation between atmospheric CO$_2$ and dust fluxes in EDC during the MIS 6 implies millennial-scale CO$_2$ variations might be influenced by iron fertilization in the Southern Ocean during the MIS 6 (Ziegler et al., 2013; Gottschalk et al., 2016; Anderson et al., 2014; Martinez-Garcia et al., 2014). In today's Southern Ocean, biological productivity is limited, reflected in a relatively low chlorophyll content. This indicates that the phytoplankton in the Southern Ocean have limited access to essential micronutrients such as iron. Aeolian dust input into the Southern Ocean can modulate iron

deposition. If the amount of aeolian dust input in the Southern Ocean increases, the productivity of phytoplankton in the Southern Ocean increases and carbon fixation in the Southern Ocean biosphere is thus enhanced. Organic detritus sinks into the deep ocean reservoir (Marinov et al., 2008), and atmospheric $CO_2$ can thus be drawn down by what is known as the biological carbon pump (Martin, 1990).

This is also written in the revised manuscript in new section 3.3 Relationship between the dust flux in EDC and atmospheric $CO_2$

– P13L13-18 – Need more references in this paragraph.

Bereiter et al. (2012) added

– P14 – After reading this section it strikes me that there is a large amount of discussion about AMOC changes without actually showing any AMOC data. The discussion is very "AMOC-centric." Indeed, we believe that AMOC changes are probably key to explaining the MIS 3 CO2 changes, but to assume the same mechanism operates in MIS 6 without data to suggest so, and then to make assertions about the AMOC based on the CO2 trends at least requires some qualification in my mind. It is okay to speculate, but please say explicitly that you are doing so and that it is based on extrapolation of the relationships observed in MIS 3.

Due to the lack of existing proxy data with high temporal resolution and high precision and modelling studies, explanations of carbon cycle mechanisms during MIS 6 are limited. However, hypotheses of these mechanisms have been presented by previous studies, and the continued discussion of these hypotheses and how our new observations may redirect the discussion, even if the very limited amount of data means that this discussion is speculative in nature, is important. We hope that this discussion will be helpful for future studies, and have made sure, as suggested by the reviewer, to clearly label any speculative discussion in the text.

Some paragraphs in Section 4.1 and 4.2 were removed and re-written, please see below.

P11L12- P12L16 in the manuscript revised to: As mentioned above, atmospheric $CO_2$ on millennial timescales can be controlled by $CO_2$ exchange between the ocean and the atmosphere, as well as changes of terrestrial carbon stocks. Coupled climate carbon cycle models reported that the variations of atmospheric $CO_2$ concentration on millennial timescales are mainly dominated by deep ocean inventory, requiring a few millennia to react to climate change (Schmittner and Galbraith, 2008). On the other hand, the response of the terrestrial biosphere is fast (centennial timescale) (Bouttes et al., 2012; Menviel et al., 2014; Schmittner and Galbraith, 2008). Although different models differ significantly in the $CO_2$ response to AMOC changes, the initial $CO_2$ evolution of the terrestrial biosphere and deep ocean to AMOC perturbations are opposite in model simulations (Gottschalk et al., 2019). Thus, due to the opposite direction of $CO_2$ change of ocean and terrestrial reservoirs, atmospheric $CO_2$ variations might be muted if the NH duration is short (Bouttes et al., 2012; Menviel et al., 2014; Schmittner and Galbraith, 2008). There is, on the other hand, evidence that not all of the processes of $CO_2$ exchange follow these general trends--for example, atmospheric $CO_2$ might be changed on centennial timescales by carbon exchange between the deep and surface ocean (Rae et al., 2018) or atmospheric $CO_2$ might be influenced slowly by soil

decomposition (Köhler et al., 2005); and it is important to note that modeling studies are limited by the available proxy data during MIS 6.

Another possible reason for the difference between $CO_2$ changes during short and long stadials may be related to a stronger reduction of the NADW during long stadials (Henry et al., 2016; Margari et al., 2010), which would cause a stronger upwelling of deep water in the Southern Ocean (Menviel et al., 2008; Schmittner et al., 2007). These events may reduce stratification in the Southern Ocean due to an increase in salinity of the surface waters and a relative freshening of the deep water (Schmittner et al., 2007). As a result, atmospheric $CO_2$ can be increased due to upwelling and outgassing of $CO_2$ in the Southern Ocean (Schmittner et al., 2007). The co-occurring upwelling in the SO during AIMs for the last termination has been examined (Anderson et al., 2009) but, due to the lack of proxy data with precise age scale for upwelling in the Southern Ocean, this hypothesis cannot be confirmed during MIS 6.

P13L5-P13L12 in the manuscript revised to: Two different lags of $CO_2$ variations with respect to NH warming are present in the MIS 6 period (Figure 8). CDM 6e.2 is nearly synchronous with the abrupt warming in the NH (no significant lag of 200±360 yrs), while the lags for CDM 6d.2 (1,300±450 yrs) and CDM 6d.1 (1,500±280 yrs) are much longer. Two modes of $CO_2$ variations are also observed during the last glacial period. As the last glaciation progressed from MIS 5 to MIS 3 (Figure 8), the lag of $CO_2$ maxima with respect to NH millennial-scale warming significantly increased. This observation may be explained by the different AMOC settings in MIS 5 and MIS 3 (Bereiter et al., 2012). We speculate that, as observed during the last glacial period, the configuration of oceanic circulation during MIS 6d might be also the cause of the change in the time lags between NH abrupt warming events and $CO_2$ variations during the early MIS 6.

P13L30- P14L12 in the manuscript revised to: In spite of the inconclusive modeling studies, limited proxy evidence does not exclude the possibility that the configuration of AMOC and its changes over MIS 6 may explain the presence of two different CDM lags. We find this hypothesis to be worth at least a speculative discussion. According to the $\delta^{13}C_{benthic}$ record in the MD01-2444 core (Margari et al., 2010), the value of $\delta^{13}C_{benthic}$ during during 180–168 kyr BP was lower than during MIS3, which indicates that the North Atlantic overturning cell during MIS 6 was likely even shallower than that during MIS 3 (Margari et al., 2010). This implies southern-sourced water masses were more expanded to the north, and the density difference between the northern-sourced water masses and southern-sourced water masses increased. This shallower oceanic circulation during MIS 6 (Margari et al., 2010) may have caused the millennial–scale delays with respect to abrupt NH warming events during MIS 6. It seems pertinent to investigate whether the slightly different ocean settings during MIS 3 and 6 (Margari et al., 2010) can also explain the longer lag between the abrupt warming in NH and CDMs during MIS 6d (1,400±375 yrs on average) when compared to the lags of CDMs (770±180 years on average) during MIS 3. However, our study has lower temporal resolution compared to the $CO_2$ data set during MIS 3. In addition, because of the low accumulation at EDC, the estimation of the exact timing of CDM from the EDC ice core might be less accurate compared to that from the TALDICE ice core, for example, due to the narrower gas age distribution of TALDICE (Bereiter et al., 2012). The remaining uncertainty is related to analytical uncertainties and to the temporal resolution of the two records. To further investigate the exact relationship between CDM and abrupt warming in the NH, additional $CO_2$ measurements from a higher accumulation site could be helpful.

**CONCLUSIONS**

– P14L22 – "Unprecedented" strikes me as too strong of a word.

Revised to "with high temporal resolution and improved analytical precision"

– I think the conclusion section should contain less about the AMOC. The primary contributions of the paper (in my mind) are the new data, the revisions to the EDC gas age scale, and perhaps the observations of leads/ lags relative to abrupt CH4 changes. The differences in the organization of AMOC between and within MIS 6 and MIS 3, as well as the relationships between stadial length and AMOC perturbation should be left out here. They are interesting hypotheses, but they are not supported by data. See also my note above about rewording the discussion to be more explicitly speculative.

Two sentences removed from the text: "probably because the duration of upwelling in the Southern Ocean was not sufficient to impact atmospheric $CO_2$, in line with Ahn and Brook (2014)" "The change in lag time might be related to a change in the organization of the AMOC from MIS 6e to MIS 6d."

**Technical Corrections:**

– Section 2.4 is titled "Ice age revision: : :" but the gas chronology, not the ice chronology, is what is actually revised. It might be confusing, so consider titling this section "Gas age revision: : :"

Revised

– In Figure 8 the authors show various CO2 maxima plotted against the lead/lag with respect to the onset of Northern Hemisphere warming. It would be helpful to clarify, for example, "CDM 12" corresponds to DO 12, etc.

A sentence added to the caption: During the last glacial period, the AIM number corresponds to the DO number for corresponding DO and AIM events.

– P2L10 – Capitalize "Hemisphere" in "Northern and Southern hemisphere, respectively."

Revised

– P2L15 – "opposite"

Revised

– P2L17 – I suggest leaving out "In response to the millennial temperature perturbations,"

Removed

– P2L32-33 – No need to repeat "MIS 3" and "MIS 6" in parentheses. Just state the age ranges.

This sentence summarizes the research purpose in this study. We prefer re-introducing the target period specifically here. In addition, the age of both MIS 3 and MIS 6 were not mentioned before that sentence. Thus, in our opinion, it is appropriate to mention both stage name and age range in this sentence.

– P2L32 – Why just "early MIS 6?" The data also span some of late MIS 6, younger than 160 kyr.

New data covers the entire MIS 6 but we focused on the interpretation of prominent $CO_2$ variations, which occur in early MIS 6.

– P3L10 – I think the sentence about a shallower AMOC cell can be combined with the preceding discussion about weaker AMOC.

Accepted, rewritten.

– P12L8-9 – You already said this in the previous sentence (NADW can be slowed down after freshwater forcing). I think it can be omitted.

The sentence "When large amounts of low-density fresh water are released into the NA, NADW formation can be slowed down." removed

– P25Fig5 – There is a typo in the legend. "Uncertainties of calculated from savitsky golay filtering." I am not certain exactly what it is supposed to say.

Revised to "Uncertainties of Savitsky Golay filtering."

– SI P7FigS7 – The caption says "Two boxes: : :" but there are five.

5  Revised

Anderson, R. F., Barker, S., Fleisher, M., Gersonde, R., Goldstein, S. L., Kuhn, G.,Mortyn, P. G., Pahnke, K., and Sachs, J. P.: Biological response to millennial variability of dust and nutrient supply in the Subantarctic South Atlantic Ocean, Philosophical Transactions of the Royal Society a-Mathematical Physical and Engineering

10  Sciences,372, 17, 10.1098/rsta.2013.0054, 2014.

Gottschalk, J., Skinner, L. C., Lippold, J., Vogel, H., Frank, N., Jaccard, S. L., andWaelbroeck,C.: Biological and physical controls in the Southern Ocean on past millennialscale atmospheric CO2 changes, Nat. Commun., 7, 11, 10.1038/ncomms11539, 2016.

Lourantou, A., Chappellaz, J., Barnola, J. M., Masson-Delmotte, V., and Raynaud,D.: Changes in atmospheric CO2 and its carbon isotopic ratio during the penultimate deglaciation, Quat. Sci. Rev., 29, 1983-1992, 10.1016/j.quascirev.2010.05.002, 2010.

20  Martinez-Garcia, A., Sigman, D. M., Ren, H. J., Anderson, R. F., Straub, M., Hodell, D. A., Jaccard, S. L., Eglinton, T. I., and Haug, G. H.: Iron Fertilization of the Subantarctic Ocean During the Last Ice Age, Science, 343, 1347-1350, 10.1126/science.1246848, 2014.

Schneider, R., Schmitt, J., Kohler, P., Joos, F., and Fischer, H.: A reconstruction of atmospheric carbon dioxide

25  and its stable carbon isotopic composition from the penultimate glacial maximum to the last glacial inception, Climate of the Past, 9, 2507-2523, 10.5194/cp-9-2507-2013, 2013.

Ziegler, M., Diz, P., Hall, I. R., and Zahn, R.: Millennial-scale changes in atmospheric CO2 levels linked to the Southern Ocean carbon isotope gradient and dust flux, Nature Geoscience, 6, 457-461, 10.1038/ngeo1782, 2013.

30

**References**

[revised manuscript text omitted]

---

## Author Response (AR1)

Dear editor and reviewers,

This compiled document, as requested, contains point-by-point responses to each of the reviewers' comments and a marked-up version of our revised manuscript. These responses are mostly the same as those submitted in the previous step of the review except for a few small revisions. Within the point-by-point responses, our detailed responses to the comments are shown in blue, and the resulting changes to the manuscript are shown in green.

We would like to draw attention to two more significant changes made in response to the reviewers' concerns. First, we have developed a compiled $CO_2$ record for the EDC ice core over MIS 6, using both published and previously unpublished data from multiple measurement systems. In addition, we have adjusted the nomenclature of the Carbon Dioxide Maxima in order to match the numbering of Margari et al. (2010) and Gottschalk et al. (2020), two key studies about MIS 6.

On behalf of all co-authors,

Jinhwa Shin

Gottschalk, J., Skinner, L. C., Jaccard, S. L., Menviel, L., Nehrbass-Ahles, C., and Waelbroeck, C.: Southern Ocean link between changes in atmospheric CO2 levels and northern-hemisphere climate anomalies during the last two glacial periods, Quaternary Sci. Rev., 230, 106067, 2020.

Margari, V., Skinner, L., Tzedakis, P., Ganopolski, A., Vautravers, M., and Shackleton, N.: The nature of millennial-scale climate variability during the past two glacial periods, Nat. Geosci., 3, 127, 2010.

**Anonymous Referee #1**

Review of manuscript cp-2019-142:

**General Comments:**

The manuscript by Shin and others presents new, high-resolution measurements of $CO_2$, $CH_4$, and $\delta15N$ in EDC ice core samples spanning the glacial period, MIS 6. The new data resolve millennial-scale variations in $CO_2$ and $CH_4$. The authors independently identified MIS 6 stadial durations in tree pollen % and planktonic $\delta18O$ in the Iberian Margin marine sediment core MD01-2444. The authors also revised the MIS 6 gas age chronology of the EDC ice core (previously AICC 2012) using new estimates of $\Delta$depth from the $\delta15N$ data. The revised EDC age scale, along with the timing of climate variations observed in the sediment core, provides the authors with a temporal framework for understanding millennial-scale $CO_2$ variations during the penultimate glacial period.

The authors specifically analyze the timing of the $CO_2$ changes relative to changes in $CH_4$, considered here a proxy for NH warming, identifying leads/ lags between the two records. They also discuss differences between the $CO_2$ features in MIS 6 and analogous features that occurred in MIS 3. The authors also observe differences in the magnitudes of $CO_2$ maxima during MIS 6. They identify a relationship between the amplitude of $CO_2$ change and the duration of the preceding stadial event, offering the hypothesis that the amplitude of $CO_2$ variations depends on the duration of AMOC perturbations. They also identify a shift in the lag of $CO_2$ maxima from MIS 6e to MIS 6d and suggest that this may be due to a change in the organization of AMOC. This manuscript is well written, organized, and clearly presented, the science is in my opinion sound, and the new datasets represent important contributions that will be of interest to others in the field. The work is appropriate for the journal Climate of the Past, and I recommend this paper for publication after minor revisions. Below I list specific comments that, if addressed, will aid in the clarity of the paper and hopefully strengthen the analyses therein. I also list technical corrections below.

**Specific Comments:**

**INTRODUCTION**

– P3L9 – Can you provide a reference for the longer duration of stadials in early MIS 6?

Accepted. A reference (Margari et al., 2010) added in the text.

– P3L15 – There are more pre-existing $CO_2$ measurements from late MIS 6 besides those from Vostok (Lourantou et al., 2010; Schneider et al., 2013).

References added in P(page)3L(line)15.

**METHODS**

– P3L31 – Did the measured $CO_2$ concentration depend on the amount of air injected?

The $CO_2$ concentrations of 5 measurements were constant. The amount of air injected did not impact the $CO_2$ concentration.

(Presumably, the pressure in the sample loop depleted across the 5 individual injections. Was there a linearity effect?)

5    The $CO_2$ concentrations do not depend on the sample size, and for the 5 consecutive injections on the same sample we obtained a linear relationship between pressure and partial pressure of $CO_2$, i.e., concentrations did not change

– P4L18-19 – Was the amount of contamination in each chamber consistent from day to day?

Each chamber shows different contamination levels. We could measure 3-4 samples a day and blank tests were conducted every 10 measurements of natural ice. The amount of contamination in each chamber varied within

10    ~0.5-2 ppm, averaging 1 ppm during the measurement, and no drifts were observed during the 2 months.

Did it depend on the length/ amount of crushing?

To avoid any effect of the duration/ amount of crushing and $CO_2$ contamination caused by crushing process on the measured CO2 concentration we kept the same length/ amount of crushing process both for real samples and standards over ice.

15    – Did you run replicate CO2 measurements on ice samples from the same depths?

In this study, large sample sizes (40g) were used, and because limited ice samples were available, we could not make replicates. To ensure precise data, we controlled the precision of the system to around 1 ppm using gas standards over gas-free synthetic ice measurements and assume their variability to be the same as for true ice.

In my opinion, this would be a better estimate of the true system precision.

20    Replicates additionally account for any differences between two ice samples, making a better estimate of standard deviation of the final measurement but not necessarily of system precision itself. For example, Lüthi et al. (2010) show that there exists true small scale variability in $CO_2$ concentrations in the ice below the Bubble Clathrate Transition Zone. Due to the diffusion effect, this small variation of atmospheric $CO_2$ is smoothed to some degree. In our study, large sample sizes (40g) of the ball mill system were used to reconstruct atmospheric $CO_2$, so a low-

25    noise signal from the ice core is extracted (the smaller measurements used in other systems would be noisier in theory). The standard deviation of the measurement is estimated from the 5 injections, but system precision, which is added to the uncertainty of the measurements, was calculated from the blank measurement, accounting for the possible sources of $CO_2$ contamination with our analytical procedure.

– P4L25 – Can you state briefly how the new, corrected CO2 record compares to the preexisting CO2 data?

30    The new $CO_2$ data from EDC are corrected for gravitational fractionation and contamination caused by the analytical process. The previous $CO_2$ measurements from Vostok ice core by ball mill system (Petit et al., 1999) from EDC measured by ball mill (Lourantou et al., 2010) were corrected for gravitational fractionation effect but not corrected the $CO_2$ contamination effect (Figure 4 in the revised manuscript). Additionally, the Vostok data use a less precise age scale.

We have included a discussion of the $CO_2$ offset between the new data set and existing data sets in detail in section 3.1.

– P4L32 – Can you state the precision of the CH4 measurements?

This is now included. The precision of the system was estimated at ~11 ppb on average.

– P5L3 – What do you think are possible reasons for the systematic offset? Please describe briefly.

A systematic offset of 6 ppb between IGE and CEP was observed (Loulergue et al., 2008). The offsets are due to differences in corrections for contamination caused by the analytical procedure, a systematic offset of 6 ppb between IGE and CEP was observed (Loulergue et al., 2008).

Please see P4L15-P20L22.

– P6L20 – Figure S4 in the SI does not have a label to distinguish blue from red.

Revised

– P6L22 – I do not follow how Figure 2 supports the claim that the previous method was "relatively unbiased but not entirely exact."

Sorry, this was the wrong figure number. Figure number revised to Figure S3.

– P6L13 – In Figure 3 it appears that the midpoints in the transitions are somewhat ambiguously defined. Sometimes they fall between a local max and min for d18O, sometimes for pollen %. The markers are chosen as midpoints between local maxima/minima, but sometimes it is unclear where those max/ min data points are. 6iii, for example, could easily be shorter (i.e., it looks like the end marker at 174.2 ka could be defined at an older age). 6v is a particularly ambiguously defined stadial – I do not see which maximum and minimum pair defines the older marker. Could you define the stadial durations more objectively? The ambiguity and subjectivity in picking the stadial transitions lead me to believe that they were defined while also considering the ice core data. That's not necessarily a bad thing, but perhaps you should just be forthright and show the gas data in Figure 3 along with the sediment core data.

In our study, the durations of the six NA stadials were originally defined as the interval between the midpoints of the stadial transition of both $\delta^{18}O$ of planktonic foraminifera and tree pollen in MD01–2444 (C and D in figure 3) which was suggested by Margari et al. (2010). With this data we observed that the magnitude of atmospheric $CO_2$ change is generally correlated with the NA stadial duration (r=0.7, n=6) during the early MIS 6 period.

As the reviewer mentioned, not all of the stadial durations during MIS 6 are entirely clear using this method. As suggested by the reviewer, a synthetic Greenland $\delta^{18}O_{ice}$ record (Barker et al., 2011) and $\delta D$ variations in Antarctic ice core are plotted in Figure 3 as references(AICC2012 age scale). The interval between the maximum and the preceding minimum of $\delta D$ in the EDC record can also be used to estimate the duration of the stadial transitions (Gottschalk et al., 2020; Margari et al., 2010). In most cases, the synthetic Greenland $\delta^{18}O_{ice}$ record and the interval between the maximum and the preceding minimum of $\delta D$ in the EDC record confirm the definition of NA stadials selected by $\delta^{18}O$ of planktonic foraminifera in MD01–2444 and tree pollen in MD01–2444. However, the duration of the NA stadial in MIS 6iii is not clearly confirmed by Greenland $\delta^{18}O_{ice}$ and $\delta D$ in the EDC (Figure 3).

We recalculated the durations of the six NA stadials using the interval between the stadial transitions as recorded in the EDC δD record (Gottschalk et al., 2020; Kawamura et al., 2017; Margari et al., 2010). Minima and maxima were selected by finding zero values in the second Savitsky–Golay filtered derivative of the data (the same method we used to pick minima and maxima of atmospheric $CO_2$; P9 in SI and Figure 1 in this text).

The red dots and error bars on δD in the EDC record in Figure 3 of the main text show the estimated minima and maxima of temperature corresponding to stadial transitions using this method, along with their uncertainties. However, using this tool, durations of 6ii and 6i are apparently overestimated due to ambiguity concerning the maximum in 6ii and minimum in 6i. Neither our method nor that of Margari et al. (2010) can be considered absolutely correct. To account for the differences between the two methods, we took the stadial duration to be the mean of the duration estimated by both $\delta^{18}O$ of planktonic foraminifera and tree pollen in MD01–2444 and dD definitions. The correlation coefficient between the magnitude of atmospheric $CO_2$ change and the NA stadial duration remains high (r=0.93, n=6) during the early MIS 6 period.

This new calculation was added in section 2.6. Please see section 2.6.

**RESULTS**

– P8L3 – You should mention the known phenomenon of CO2 offsets between different ice cores (e.g. WAIS versus Law Dome). The co-author Christoph could certainly comment on this.

When the air is extracted from an ice core sample where bubble and clathrates co-exist, different dry extraction methods with different extraction efficiencies on bubbly and clathrate ice may lead to biased $CO_2$ concentrations (Lüthi et al., 2010; Schaefer et al., 2011). During clathrate formation, the gas is partitioned into clathrates due to the different gas diffusivities and solubilities (Salamatin et al., 2001). $CO_2$ has consistently been observed to be depleted in bubbles and enriched in clathrates (Schaefer et al., 2011). Degassing from clathrates during extraction takes much longer than air release from bubbles; thus, if air from the clathrate ice is not extracted entirely, $CO_2$ measurement will be lower than the true value.

The ball mill shows extraction efficiencies of ~62% for bubbles and ~52% for clathrates on average (Schaefer et al., 2011). If the ball mill is used to reconstruct $CO_2$ in Bubble–Clathrate Transformation Zone (BTCZ), $CO_2$ concentrations can be biased.

$CO_2$ concentrations from EDC were reconstructed from 150 depth intervals that cover 2036.7 to 1787.5 m along the EDC ice core, which consist of clathrate ice. There exists true small scale variability in $CO_2$ concentrations in the ice below the Clathrate Zone (Lüthi et al., 2010). Due to the diffusion effect, this small variation of atmospheric $CO_2$ is smoothed. Thus, $CO_2$ concentrations in these depth intervals might represent the initial mean atmospheric concentration. However, the EDC ice core for MIS 6 was drilled in 1999 and, the ice core has been stored for ~20 years in cold rooms at -22.5 ± 2.5°C before the gas is analysed. More than 50% of the initial hydrates present in the freshly drilled ice may have been decomposed and transformed into secondary bubbles, or gas cavities (Lipenkov, *Pers. Comm.*). We expect the same fractionation as during the clathrate formation process, hence bubbles would be depleted in $CO_2$. Thus, $CO_2$ concentrations from EDC may be lower. The portion of the Vostok ice core covering MIS 6 is also clathrate ice, but it was drilled in 1998 and measured immediately (Petit et al., 1999), and less clathrates may have transformed into secondary bubbles. Thus $CO_2$ concentrations from Vostok

during MIS 6 may be higher and potentially reflect the true atmospheric concentration more closely. In our study we concentrate on the relative millennial changes of $CO_2$ around the mean glacial concentration, which are the same in all the $CO_2$ records available so far, Thus, our conclusion in this paper are independent of which absolute mean $CO_2$ level is correct. As the new data in this study are currently the best quality data in terms of repeatability, we use our new data as the reference record and correct for any inter-core offsets. We, however, state explicitly in the text that the absolute mean $CO_2$ level during MIS6 is not known better than 5 ppm.

The new section, '3.1 Data compilation' in the revised manuscript is dedicated to the $CO_2$ offset between the EDC and Vostok ice cores.

– P25 Fig5 – It is unclear how the blue CDM events were defined. Do they relate somehow to the stadial duration markers you defined previously? If not, please clarify how you identified them (or provide proper reference to SI).

We now explain the method used to define CDM events in detail in the SI (Table S2 and Figures S5 in SI). These references were added in Section 3.3. Please see the section 'Definition of minima and maxima of atmospheric CO2 and temperature' in SI.

– P26Fig6 – Shading or vertical lines would help to delineate the CDM's in Figure 6. Right now the text floats at the bottom and is unclear exactly what the labels refer to.

Added in Figure 6

– One result that strikes me as interesting, and not discussed in the paper, is that the lowest CO2 and Antarctic temperature values occur in the early/ middle part of MIS 6, not the latest part (as in MIS 2). CH4, on the other hand, reaches the lowest values during late MIS 6, right before the termination, as does peak glaciation as inferred from the benthic d18O. This is unlike MIS 2, which is characterized by low CO2, low Antarctic temperature, low CH4, and peak glaciation occurring simultaneously. Can you speculate why CO2 is higher in late MIS 6 relative to earlier in MIS 6, despite full glacial extent?

A saturation index indicating variations in respired carbon content in the deep sub-Antarctic Atlantic (MD07-3077) and atmospheric $CO_2$ have been shown to be closely anti-correlated (Gottschalk et al., 2020). This observation indicates that the regulation of global atmospheric $CO_2$ variations on millennial time scales is highly influenced by the marine carbon cycle in the Southern Ocean (Fischer et al., 2010) during MIS 6.

As shown in this figure, atmospheric $CO_2$ from EDC is highly co-related with dust flux in EDC (Lambert et al., 2012), $\delta D$ in EDC (Jouzel et al., 2007) and summer sea surface temperature in the deep sub-Antarctic Atlantic (MD07-3077) (Gottschalk et al., 2020). Iron Fertilization and temperature in the Southern Ocean can affect $CO_2$ variations on millennial time scales. However, the main difference of climate between late MIS 6 and early MIS 6 is temperature in the Southern Ocean. Colder conditions are observed in the Southern Ocean in early MIS 6 than in late MIS 6. Colder conditions in early MIS 6 would allow for more carbon uptake in the southern Ocean. Thus, the $CO_2$ level during the early MIS 6 might be slightly lower than the late MIS 6 due to colder ocean conditions during the early MIS 6. In contrast, $CH_4$ is reflecting primarily climate/hydrological conditions on land in the tropics and to a much smaller extent in high northern latitudes. Thus, a decoupling of the two parameters suggests different glacial climate evolution in high southern latitudes and the tropics.

[Figure]

**Figure R1:** Climate proxies during MIS 6. Vertical blue dotted lines indicate the six CDM events that we identify during the early MIS 6. A: Dust flux in EDC (Lambert et al., 2012). B: EDC water isotopic record (Jouzel et al., 2007). C: Sea summer surface temperature in the deep sub-Antarctic Atlantic (MD07-3077) (Gottschalk et al., 2020). D: Saturation Index in the deep sub-Antarctic Atlantic (MD07-3077) (Gottschalk et al., 2020). E: Atmospheric $CO_2$ from EDC (this study). The red line indicates Savitsky Golay filtering curve made with a ~150 yr cut-off period (red dotted line).

– P28Fig8 - The authors compare the timing of CO2 maxima relative to the onset of NH warming. The CO2 measurements come from different ice cores with different age scales (to my knowledge at least, Byrd is not synchronized to the AICC 2012 as EDML, EDC, and TALDICE are). What is the bias or uncertainty in the analysis due to age offsets? Why not exclusively use the EPICA cores on a unified age scale for this analysis?

To calculate leads and lags between $CO_2$ and the abrupt warming in NH, we calculated the time lag for each CDM following abrupt warming events in the NH. In this study, given the fact that when temperature increases rapidly in Greenland, $CH_4$ increases rapidly within 50 yrs (Baumgartner et al., 2014; Rosen et al., 2014), we used $CH_4$ as a time marker of rapid warming in the NH.

$CH_4$ and $CO_2$ signals are both reconstructed from the air bubbles in the same ice, and as such there is no chronological uncertainty with respect to individual timings. The Byrd core was synchronized to the EDML core in the gas phase by Bereiter et al. (2012), and thus can be synchronized to the AICC2012 chronology as well. Without synchronization, there can be significant differences in event duration between two cores. However, with the synchronization between Byrd and EDML, these inconsistencies should be minimized. The measurements for each period are chosen to maximize resolution and minimize uncertainty related to gas trapping. –The estimation of the exact timing of CDM from the EDC ice core might be less accurate compared to that from the TALDICE ice core, for example, due to the narrower gas age distribution of TALDICE (Bereiter et al., 2012). The remaining uncertainty is related to analytical uncertainties and to the temporal resolution of the two records.

**DISCUSSION**

– P11L26&31 – When you say that the terrestrial biosphere can "compensate" for the slow response of the deep ocean, do you mean in terms of its timing or in terms of the direction of CO2 change?

The direction of $CO_2$ change.

This paragraph re-written, please see P11L23-P11L29.

Please clarify. "Compensate" may not be the best word to use in case it is confused with carbonate compensation.

Revised. Changed to "muted". Please see P10L37.

– P13 – After the discussion of AMOC and deep ocean ventilation, I realized there was no discussion entertaining productivity fluctuations as a possible mechanism for millennial-scale CO2 variability (Ziegler et al., 2013; Gottschalk et al., 2016; Anderson et al., 2014; Martinez-Garcia et al., 2014).

The dust flux in EDC clearly shows millennial variations during MIS 6. The anti-correlation between atmospheric $CO_2$ and dust fluxes in EDC during the MIS 6 implies millennial-scale $CO_2$ variations might be influenced by iron fertilization in the Southern Ocean during the MIS 6 (Ziegler et al., 2013; Gottschalk et al., 2016; Anderson et al., 2014; Martinez-Garcia et al., 2014). In today's Southern Ocean, biological productivity is limited, reflected in a relatively low chlorophyll content. This indicates that the phytoplankton in the Southern Ocean have limited access to essential micronutrients such as iron. Aeolian dust input into the Southern Ocean can modulate iron deposition. If the amount of aeolian dust input in the Southern Ocean increases, the productivity of phytoplankton in the Southern Ocean increases and carbon fixation in the Southern Ocean biosphere is thus enhanced. Organic detritus sinks into the deep ocean reservoir (Marinov et al., 2008), and atmospheric $CO_2$ can thus be drawn down by what is known as the biological carbon pump (Martin, 1990).

– P13L13-18 – Need more references in this paragraph.

Bereiter et al. (2012) added

– P14 – After reading this section it strikes me that there is a large amount of discussion about AMOC changes without actually showing any AMOC data. The discussion is very "AMOC-centric." Indeed, we believe that AMOC changes are probably key to explaining the MIS 3 CO2 changes, but to assume the same mechanism operates in MIS 6 without data to suggest so, and then to make assertions about the AMOC based on the CO2 trends at least requires some qualification in my mind. It is okay to speculate, but please say explicitly that you are doing so and that it is based on extrapolation of the relationships observed in MIS 3.

Due to the lack of existing proxy data with high temporal resolution and high precision and modelling studies, explanations of carbon cycle mechanisms during MIS 6 are limited. However, hypotheses of these mechanisms have been presented by previous studies, and the continued discussion of these hypotheses and how our new observations may redirect the discussion, even if the very limited amount of data means that this discussion is speculative in nature, is important. We hope that this discussion will be helpful for future studies, and have made sure, as suggested by the reviewer, to clearly label any speculative discussion in the text.

Some paragraphs in Section 4.1 and 4.2 were removed and re-written. Please see Section 4.1 and 4.2.

**CONCLUSIONS**

– P14L22 – "Unprecedented" strikes me as too strong of a word.

Revised to "a high temporal resolution"

– I think the conclusion section should contain less about the AMOC. The primary contributions of the paper (in my mind) are the new data, the revisions to the EDC gas age scale, and perhaps the observations of leads/ lags relative to abrupt CH4 changes. The differences in the organization of AMOC between and within MIS 6 and MIS 3, as well as the relationships between stadial length and AMOC perturbation should be left out here. They are interesting hypotheses, but they are not supported by data. See also my note above about rewording the discussion to be more explicitly speculative.

Two sentences removed from the text: "probably because the duration of upwelling in the Southern Ocean was not sufficient to impact atmospheric $CO_2$, in line with Ahn and Brook (2014)" "The change in lag time might be related to a change in the organization of the AMOC from MIS 6e to MIS 6d."

**Technical Corrections:**

– Section 2.4 is titled "Ice age revision: : :" but the gas chronology, not the ice chronology, is what is actually revised. It might be confusing, so consider titling this section "Gas age revision: : :"

Revised

– In Figure 8 the authors show various CO2 maxima plotted against the lead/lag with respect to the onset of Northern Hemisphere warming. It would be helpful to clarify, for example, "CDM 12" corresponds to DO 12, etc.

A sentence added to the caption: During the last glacial period, the AIM number corresponds to the DO number for corresponding DO and AIM events.

– P2L10 – Capitalize "Hemisphere" in "Northern and Southern hemisphere, respectively."

Revised

– P2L15 – "opposite"

Revised

– P2L17 – I suggest leaving out "In response to the millennial temperature perturbations,"

Removed

– P2L32-33 – No need to repeat "MIS 3" and "MIS 6" in parentheses. Just state the age ranges.

This sentence summarizes the research purpose in this study. We prefer re-introducing the target period specifically here. In addition, the age of both MIS 3 and MIS 6 were not mentioned before that sentence. Thus, in our opinion, it is appropriate to mention both stage name and age range in this sentence.

– P2L32 – Why just "early MIS 6?" The data also span some of late MIS 6, younger than 160 kyr.

New data covers the entire MIS 6 but we focused on the interpretation of prominent $CO_2$ variations, which occur in early MIS 6.

– P3L10 – I think the sentence about a shallower AMOC cell can be combined with the preceding discussion about weaker AMOC.

Accepted, rewritten.

– P12L8-9 – You already said this in the previous sentence (NADW can be slowed down after freshwater forcing). I think it can be omitted.

The sentence "When large amounts of low-density fresh water are released into the NA, NADW formation can be slowed down." removed

– P25Fig5 – There is a typo in the legend. "Uncertainties of calculated from savitsky golay filtering." I am not certain exactly what it is supposed to say.

Revised to "Uncertainties of Savitsky Golay filtering."

– SI P7FigS7 – The caption says "Two boxes: : :" but there are five.

Revised

**References**

[revised manuscript text omitted]

**Anonymous Referee #2**

Shin et al. present new records of ice core CO2, CH4, and d15N from the EPICA dome C core during MIS 6. They show that the CO2 maxima tend to lag Antarctic d18O maxima, by an amount that increases thought the glacial period. The magnitude of CO2 increase scales with the duration of North Atlantic stadial period, suggesting a key role of AMOC variations in millennial-scale CO2 variability.

The data and analysis are obviously of interest to the broader paleoclimate community, and this paper should be published with only minor corrections. I believe that the manuscript can be clarified in some places. The main conclusion seems to be that MIS6 behaves very similarly to the last glacial period (as expected). Owing to this similarity, the discussion is somewhat long and involves a lot of speculation – much of which has already been said in earlier work (for example Bereiter et al. 2012).

Throughout the paper the authors present speculative climatic mechanisms, or uncorroborated results from individual model simulations as established fact. One example: "Due to the reduction of Summer Monsoon intensity in East Asia, salinity at the surface of the Pacific Ocean is increased. Thus, AABW and North Pacific Deep Water (NPDW) transport is enhanced (Menviel et al., 2014). Enhanced NPDW transport ventilates deep Pacific carbon via the Southern Ocean which may lead to atmospheric CO2 increases." While this is not a bad description of Menviel 2014, I think this would be better presented with some caution because while possibly correct, this is in no way a consensus view.

This paragraph (P11L12-P11L22) was removed.

 Throughout the paper, the authors compare MIS 3 and MIS6. In several places the authors write that MIS3 and MIS6 had different "background conditions". I am not sure what is meant by that. In what way are they really different? Both periods represent a range of orbital conditions, sea ice volumes, ITCZ positions, Heinrich events etc. So there are many places where they are very similar. I would advise the authors remove this idea that these two glacial periods are somehow very different – I don't think they have made the case that they are (and their data surely suggest that the carbon cycle responds in a very similar manner).

 We do not claim that the last two glacial periods are "very different" as the reviewer states, but rather slightly different as also stated by Margari et al., 2010 and Gottschalk et al. (2020). This is an important distinction, which we attempt to clarify in the revision below. Our analysis focuses on whether these slight differences can impact the variability of $CO_2$ on millennial time scales. This is of course already known to be the case for periods presenting more marked differences in background climatic conditions.

P2L31-P3L11 Revised to: Comparing $CO_2$ changes on millennial time scales during the past two glacial periods, MIS 3 (60–27 kyr BP) and early MIS 6 (185–160 kyr BP) can provide us with a better understanding of the carbon cycle, due to the similarities but also differences of climate conditions and events during the last two glacial periods (Figure 1). Proxy evidence indicates that the states of several important components of the climate-carbon cycle were not the same between MIS 3 and MIS 6. Sea ice cover in the South Atlantic was more extensive in MIS 6, and sea surface temperature in the South Atlantic is thought to have been lower (Gottschalk et al., 2020). The bipolar see-saw phenomenon also has been observed during the early MIS 6 period (Cheng et al., 2016; Jouzel et al., 2007; Margari et al., 2010). However, the bipolar see-saw events during MIS 6 are longer than those found during MIS 3. Events of massive iceberg discharge into the NA, which are thought to have driven millennial-scale changes in the meridional overturning circulation during MIS 3 (de Abreu et al., 2003; McManus

et al., 1999) appear to be much more frequent during MIS 3 than during MIS 6. During the early MIS 6, iceberg discharge was muted and during the time period around 175 kyr BP, summer insolation levels in the NH approached interglacial values (Berger, 1978). Due to the stronger NH summer insolation, the Intertropical Convergence Zone (ITCZ) had shifted to the north, which intensified monsoon systems in low latitude regions, such as in Asia, the Appenine Peninsula and the Levant (Ayalon et al., 2002; Bard et al., 2002; Cheng et al., 2016). This may have led to a weaker overturning circulation due to the reduction of the density of the North Atlantic surface water, making the AMOC cell shallower during MIS 6 than during MIS 3 (Gottschalk et al., 2020; Margari et al., 2010).

Due to the historical convention the last ice age is actually MIS2-4, rather than just MIS3. So a more meaningful comparison would be MIS2-4 to MIS6. Also, the authors also include MIS5 in their analysis (Fig. 8). I think the paper would be a lot simpler if the authors just claim to be studying millennial-scale CO2 variability, rather than focus on Marine Isotope Stage distinctions that may not be relevant.

The last glacial period covers MIS 2 to MIS 4. MIS 2 and 4 are full glacial periods but MIS 3 is an interstadial period, i.e. a less cold period during a glacial period.

MIS 6 covers the penultimate glacial period, and can be divided into 3 parts according to the magnitude of climate variability and climate characteristics observed in proxy data (Margari et al., 2014): early (185.2─157.7 kyr BP), transition (157.7─151 kyr BP) and late MIS 6 (151─135 kyr BP). Each part shows similarities to a specific period of the last Ice Age. Climate change on millennial time scales during the late MIS 6 (the penultimate glacial maximum) is subdued, similar to MIS 2 (the last glacial maximum). Climate variations on millennial time scales during the earlier MIS 6 (185-157 kyr BP) are more prominent, similar to those during MIS 3. Accordingly, similarities and differences of climate variations during MIS 3 and the earlier MIS 6 were chosen to understand similarities/differences in atmospheric $CO_2$ variations on millennial time scale during the past two glacial periods.

MIS 5, the interglacial period, was mentioned as a reference for our analysis of lags of $CO_2$ variations with respect to Northern Hemisphere warming. The paper by Bereiter et al. (2012) shows two modes of atmospheric $CO_2$ variations on millennial time scales with respect to abrupt warming in NA during MIS 3 and MIS 5. These modes might be caused by different configurations of oceanic circulation during MIS 5 and 3. We similarly observed two modes of lags of $CO_2$ variations with respect to abrupt warming in NA during MIS 6.

In all figures I would appreciate a more clear demarcation of the sub-sections. I am not very familiar with the MIS6a-6e definitions. Do they follow precession/Benthic sequences like in MIS5, and who has defined these?

We now use the substage numbering developed by Margari et al. (2010) and Gottschalk et al. (2020), who identify six isotopic maxima (6i-6vi from oldest to youngest) that correspond with our Carbon Dioxide Maxima. We feel that adopting this numbering maintains consistency across studies about MIS 6.

Could you please add the MIS5 and MIS6 (and MIS7?) sub-stage numbering into figures 1, 4 and 5. Also, for consistency you should mark the H-events of stage 6 in Fig 1. Where does the event numbering 6.e1 etc. come from? I have seen alternative numberings elsewhere in the literature.

We named new CDM according to the numbering by Margari et al., (2010) and Gottschalk et al. (2020). The new numbering can be found in Figure 3 and Figure 5.

Could you please add the synthetic Greenland reconstruction from Barker et al. 2011 to Figure 3, to see how it compares?

Accepted. Please see Figure 3 in the revised manuscript.

The authors do not address the CO2 offset between the records enough. It is up to 10 ppm with Vostok, which is quite large. They explain this as due to the blank correction, which is only around 1.7 ppm and therefore insufficient. Such offsets are seen more often in comparing CO2 from different cores, and may actually be in the ice. Can you explain the EDC CO2 offset between this work and Lourantou?

The ball mill system has a different extraction efficiency depending on the presence of bubbles and/or clathrates in the ice sample, which may cause an accuracy for reconstructing absolute mean $CO_2$ level. When the air is extracted from an ice core sample where bubble and clathrates co-exist, different dry extraction methods with different extraction efficiencies on bubbly and clathrate ice may lead to biased $CO_2$ concentrations (Lüthi et al., 2010; Schaefer et al., 2011b). During clathrate formation, the gas is partitioned into clathrates due to the different gas diffusivities and solubilities (Salamatin et al., 2001). $CO_2$ has consistently been observed to be depleted in bubbles and enriched in clathrates (Schaefer et al., 2011a). Degassing from clathrates during extraction takes much longer than air release from bubbles; thus, if air from the clathrate ice is not extracted entirely, $CO_2$ measurement will be lower than the true value.

The ball mill shows extraction efficiencies of ~62% for bubbles and ~52% for clathrates on average (Schaefer et al., 2011a). If the ball mill is used to reconstruct $CO_2$ in Bubble–Clathrate Transformation Zone (BTCZ), $CO_2$ concentrations can be biased.

$CO_2$ concentrations from EDC were reconstructed from 150 depth intervals that cover 2036.7 to 1787.5 m along the EDC ice core, which consist of clathrate ice. There exists true small scale variability in $CO_2$ concentrations in the ice below the Clathrate Zone (Lüthi et al., 2010). Due to the diffusion effect, this small variation of atmospheric $CO_2$ is smoothed. Thus, $CO_2$ concentrations in these depth intervals might represent the initial mean atmospheric concentration. However, the EDC ice core for MIS 6 was drilled in 1999 and, the ice core has been stored for ~20 years in cold rooms at -22.5 ± 2.5°C before the gas is analysed. More than 50% of the initial hydrates present in the freshly drilled ice may have been decomposed and transformed into secondary bubbles, or gas cavities (Lipenkov, *Pers. Comm.*). We expect the same fractionation as during the clathrate formation process, hence bubbles would be depleted in $CO_2$. Thus, $CO_2$ concentrations from EDC may be lower. The portion of the Vostok ice core covering MIS 6 is also clathrate ice, but it was drilled in 1998 and measured immediately (Petit et al., 1999), and less clathrates may have transformed into secondary bubbles. Thus $CO_2$ concentrations from Vostok during MIS 6 may be higher and potentially reflect the true atmospheric concentration more closely. In our study we concentrate on the relative millennial changes of $CO_2$ around the mean glacial concentration, which are the same in all the $CO_2$ records available so far, Thus, our conclusion in this paper are independent of which absolute mean $CO_2$ level is correct. As the new data in this study are currently the best quality data in terms of repeatability, we use our new data as the reference record and correct for any inter-core offsets (see Figure S7 and the revised text in the manuscript). We, however, state explicitly in the text that the absolute mean $CO_2$ level during MIS6 is not known better than 5 ppm.

The estimated offset between the existing $CO_2$ dataset from EDC by Lourantou et al. (2010) and our new dataset is ~2.4±2.1 ppm. The $CO_2$ data from EDC by Lourantou et al. (2010) were also reconstructed using the ball mill system. However, this dataset was not corrected for the $CO_2$ contamination caused by the analytical procedure. We estimated the level of $CO_2$ contamination to be between 1 and 2 ppm for our study. Considering that the

previous dataset was not corrected for, the offset between the two data sets is small when compared to their uncertainties.

The CO$_2$ offset between our dataset and that of Lourantou et al. (2010) is addressed in detail in the SI (P11, lines 22-26). The offset with respect to Vostok is treated in section 3.2, Page 7, lines 29-41.

5 Specific line-by-line comments:

P1L16: I don't think you can argue that the background conditions are different. That hasn't been established.

The detailed information about background conditions during the last two glacial periods is re-written for greater clarity on Page 2, lines 22-35. Additionally, we no longer refer to 'background conditions' but to specific components of the climate system that varied between the two periods.

10 P2L11: Broecker does not talk about the bipolar seesaw, but a seesaw in deepwater formation. Other references to consider are Blunier & Brook (2001); Pedro et al. (2018).

Blunier & Brook (2001); Pedro et al. (2018) added, Broecker, 1998 removed

P3L6: Normally a stronger monsoon is not associated with a weaker AMOC. How does this work?

This paragraph has been re-written. Please see the response to the first question.

15 P6L9: Add or replace with Etheridge et al (1992); this idea is much older.

Added

P5L5: The "assumption" that the bipolar seesaw was present is a pretty obvious one, and I don't think it needs to be questioned. My personal choice would have been to use Antarctic isotopes to define the stadials and interstadials (see e.g. Kawamura et al., 2017), rather than NA sediments that have much poorer age control.

[Figure]

**Figure 3:** The durations of the six NA stadials during MIS 6. A: δD of the EDC ice core (Jouzel et al., 2007). B: synthetic Greenland δ$^{18}$O$_{ice}$ record (Barker et al., 2011). C: Tree pollen percentage in the MD01-2444 core (Margari et al., 2010) D: δ$^{18}$O of planktonic foraminifera in the MD01-2444 core (Margari et al., 2010). Proxy data shown here are given on the AICC2012 age scale. Red lines indicate the midpoints of the stadial transition
25 of both δ$^{18}$O of planktonic foraminifera and tree pollen in MD01–2444. Light green bars indicate the uncertainty of the duration of each stadial transition estimated as half the temporal difference between maxima and minima

of $\delta^{18}O$ of planktonic foraminifera before and after the transition. Red dots indicate minima and maxima of $\delta D$ in the EDC ice core as selected in this study. The event numbers are indicated at the top.

In our study, the durations of the six NA stadials were originally defined as the interval between the midpoints of the stadial transition of both $\delta^{18}O$ of planktonic foraminifera and tree pollen in MD01–2444 (C and D in figure 3) which was suggested by Margari et al. (2010). With this data we observed that the magnitude of atmospheric $CO_2$ change is generally correlated with the NA stadial duration (r=0.7, n=6) during the early MIS 6 period.

As the reviewer mentioned, not all of the stadial durations during MIS 6 are entirely clear using this method. As suggested by the reviewer, a synthetic Greenland $\delta^{18}O_{ice}$ record (Barker et al., 2011) and Antarctic ($\delta D$) variations in Antarctic ice core are plotted in Figure 3 as references, on the AICC2012 age scale. The interval between the maximum and the preceding minimum of $\delta D$ in the EDC record can also be used to estimate the duration of the stadial transitions (Gottschalk et al., 2020; Margari et al., 2010). In most cases, the synthetic Greenland $\delta^{18}O_{ice}$ record and the interval between the maximum and the preceding minimum of $\delta D$ in the EDC record confirm the definition of NA stadials selected by $\delta^{18}O$ of planktonic foraminifera in MD01–2444 and tree pollen in MD01–2444. However, the duration of the NA stadial in MIS 6iii is not clearly confirmed by Greenland $\delta^{18}O_{ice}$ and $\delta D$ in the EDC (Figure 3 (in the text)).

We recalculated the durations of the six NA stadials using the interval between the stadial transitions as recorded in the EDC $\delta D$ record (Gottschalk et al., 2020; Kawamura et al., 2017; Margari et al., 2010). Minima and maxima were selected by finding zero values in the second Savitsky–Golay filtered derivative of the data (the same method we used to pick minima and maxima of atmospheric $CO_2$; P6 in SI and Figure S6).

The red dots and error bars on $\delta D$ in the EDC record in Figure 3 of the main text show the estimated minima and maxima of temperature corresponding to stadial transitions using this method, along with their uncertainties. However, using this tool, durations of 6ii and 6i are apparently overestimated due to ambiguity concerning the maximum in 6ii and the minimum of 6i. Neither our method nor that of Margari et al. (2010) can be considered absolutely correct. To account for the differences between the two methods, we took the stadial duration to be the mean of the duration estimated by both $\delta^{18}O$ of planktonic foraminifera and tree pollen in MD01–2444 and dD definitions. The correlation coefficient between the magnitude of atmospheric $CO_2$ change and the NA stadial duration remains high (r=0.93, n=6) during the early MIS 6 period.

This new calculation is described in detail Section 2.6 and P6 of the SI.

P7L31: the offsets persist in periods of stable CO2, suggesting there is more than chronological error going on. Please discuss offsets between the cores.

Accepted, Re-written. This is now treated in detail in Section 3.1, Section 3.2, and the supplement.

P10L19: again, the link between monsoon and AMOC does not make sense to me

This paragraph re-written (now beginning on the final line of page 9): Interestingly, these two CDM events occurred during MIS 6d (Figure 1), when iceberg discharge was muted and the ITCZ is thought to have shifted northward, intensifying monsoon systems in low latitude Northern Hemisphere regions, such as in Asia, the Appennine Peninsula and the Levant (Ayalon et al., 2002; Bard et al., 2002; Cheng et al., 2016). This may have led to a weaker overturning circulation due to the reduction of the density of the NA surface water, making the AMOC cell shallower with a smaller threshold in NA during MIS 6 than during MIS 3 (Margari et al., 2010).

Therefore, the two different CO2 lag timescales with respect to abrupt warming in NH during MIS 6 might be explained by this difference in background climate conditions.

P12L7: upwelling or ventilation /de-stratification?

Upwelling or ventilation

P12L15-16: Anderson does not cover MIS6, plus those records lack the resolution to investigate short stadials.

Two sentences removed: "During the short stadial in MIS 6 (AIM 6id) and the short stadials in MIS 3, the duration and strength of AMOC disruption are similar (Margari et al., 2010). This is supported by the marine proxy data for upwelling in the Southern Ocean which do not show strong variations during short stadials for both MIS periods (Anderson et al., 2009)."

P13L22: yet the CO2 variations of MIS5 are larger than those in MIS3?

The sentence at P13L22 is about $CO_2$ outgassing from the Ocean.

P14L22: remove "unprecedented". Some ice core CO2 records have decadal precision.

Removed

P21: Could you add the H-events of Stage 6 also (or perhaps an IRD record that spans the full period)? Could you mark the MIS6a-6e substage numbering? (I am not familiar with this nomeclature).

Revised. Please see Figure 1 in the revised manuscript.

Although the IRD dataset added in Figure 1 does not cover the whole 250 kyr period shown in figure 1, it is in our opinion the most appropriate to show North Atlantic events during MIS 6, and is thus used for our analysis. We prefer to not include any additional IRD datasets in the figure to avoid confusion for the reader.

P26: could you add the DO onsets you infer from CH4 as vertical bands?

It would be better to add vertical lines for CDM because there are 6 variations of atmospheric $CO_2$ but we could find only three abrupt $CH_4$ increases indicating the onset of DO events with the $CH_4$ data set. Vertical lines for CDMs added in Figure 6.

P27: Why did you not add the Stage 5 events here?

In this study we focused on atmospheric $CO_2$ variations on millennial time scales during the past two glacial periods, thus, the stage 5 was not included in Figure 7. In section 3.4 and 4.2, the stage 5 is mentioned, but only to discuss the factors that can influence the lag of atmospheric $CO_2$ with respect to abrupt warming in NH.

**References**

[revised manuscript text omitted]

**Anonymous Referee #3**

This manuscript presents new and important atmospheric CO2 concentration data from the penultimate glacial period, also known as MIS6. The data concern so-called millennial-scale climate change, which has been well documented from Greenland ice cores. Because the Greenland ice cores do not extend back into MIS6, the natural archive in which to study millennial-scale climate for this period is Antarctic ice. The data appear to be of high quality and the discussion is appropriately oriented to the question of the temporal lag of peak CO2 behind millennial-scale warm intervals. The lag is found to be larger in the colder intervals than in the warmer intervals, much as was previously found for the more recent period of MIS 3 to MIS 5.

The one major thing I find lacking in this paper is replication of CO2 data points from the same depth in the ice core. Replication of gas measurements in ice cores is fundamental in order to have confidence in the accuracy of the data. Furthermore, the authors should calculate a pooled standard deviation from the means of replicates cut from the same depth in the ice core. This is widely viewed in the field as the most reliable indicator of the overall precision of the measurement, including potential issues arising from the ice itself (such as in-situ CO2 production).

Replicates account for differences between two ice samples at the same depth, making a better estimate of standard deviation of the final measurement but not necessarily of system precision itself. For example, Lüthi et al. (2010) show that there exists true small scale variability in $CO_2$ concentrations in the ice below the Bubble Clathrate Transition Zone, which could be accounted for by using replicates, especially for small sample sizes. Due to the diffusion effect, this small variation of atmospheric $CO_2$ is smoothed to some degree. In our study, large sample sizes (40g) of the ball mill system were used to reconstruct atmospheric $CO_2$, so a low-noise signal from the ice core is extracted (the smaller measurements used in other systems would be noisier in theory). The standard deviation of the measurement is estimated from the 5 injections, but system precision was calculated from blank measurements, which were performed after every 10 measurements accounting for the possible sources of $CO_2$ contamination with our analytical procedure.

To verify our new dataset, we made a composite data set using by aligning previous sets of measurements made over the MIS 6 period on the EDC ice core to our dataset. First, we compared to two existing $CO_2$ data sets and two new $CO_2$ data sets from EDC (Figure 1 and Table 1). There are two published $CO_2$ datasets for EDC during MIS6—the first measured by the ball mill system at IGE (Lourantou et al., 2010) and the second by the sublimation system at CEP (Schneider et al., 2013). We also compared unpublished atmospheric $CO_2$ measurements from EDC by a novel centrifugal ice microtome (CIM) system, a needle cracker and a ring mill system (Shin, 2019). All records are on the AICC2012 air age scale (Bazin et al., 2013). All data sets is corrected for the gravitational fractionation effect using the new $\delta^{15}N$ data in our study.

[Figure]

**Figure 1:** Atmospheric $CO_2$ from EDC and Vostok ice cores, compared to the δD of water at EDC (temperature proxy) during 190—135 kyr BP. Blue dots: Atmospheric $CO_2$ from EDC by ball mill system (this study). Yellow dots: Atmospheric $CO_2$ from EDC by ball mill system (Lourantou et al., 2010). Purple dots: Atmospheric $CO_2$ from EDC by ring mill system. Red equilateral triangles: Atmospheric $CO_2$ from EDC by needle cracker. Black inverted triangles: Atmospheric $CO_2$ from EDC by CIM. Green rhombuses: Atmospheric $CO_2$ from EDC by sublimation. Grey dots: Atmospheric $CO_2$ from the Vostok ice core (Petit et al., 1999). Grey line: δD of water at EDC (Jouzel et al., 2007).

Because of the limited amount of samples available, the data reconstructed by both ball mill and ring mill methods are single measurements from the depth interval. $CO_2$ records by CIM, needle cracker and the sublimation methods were reconstructed from 2–5 replicates from individual depth intervals. The error bars of data without replicate indicate that the standard deviation of five consecutive injections of the gas extracted from each sample into the gas chromatography (Lourantou et al., 2010; Petit et al., 1999). The error bars of data with replicate indicate the standard deviation of the mean of replicates from the same depth interval (Schneider et al., 2013). Figure 1 shows $CO_2$ concentrations measured by the ball mill system, the ring system, the sublimation, the CIM and the needle cracker. These $CO_2$ concentrations by the ball mill system (Lourantou et al., 2010), the ring system, the sublimation (Schneider et al., 2013), the CIM and the needle cracker are systematically higher than $CO_2$ concentrations measured by the ball mill system in our study (Table 1 and Figure 1). Atmospheric $CO_2$ during the MIS 6 period shows an offset between $CO_2$ data in this study and other $CO_2$ sets, which might be related with different analytical methods.

When the air is extracted from an ice core sample where bubble and clathrates co-exist, different dry extraction methods with different extraction efficiencies on bubbly and clathrate ice may lead to biased $CO_2$ concentrations (Lüthi et al., 2010; Schaefer et al., 2011). During clathrate formation, the gas is partitioned into clathrates due to the different gas diffusivities and solubilities (Salamatin et al., 2001). $CO_2$ has consistently been observed to be depleted in bubbles and enriched in clathrates (Schaefer et al., 2011). Degassing from clathrates during extraction takes much longer than air release from bubbles; thus, if air from the clathrate ice is not extracted entirely, $CO_2$

measurement will be lower than the true value. The ball mill shows extraction efficiencies of ~62% for bubbles and ~52% for clathrates on average (Schaefer et al., 2011). If the ball mill is used to reconstruct $CO_2$ in Bubble–Clathrate Transformation Zone (BTCZ), $CO_2$ concentrations can be biased.

**Table 1:** Existing $CO_2$ data sets from EDC and Vostok ice core and new $CO_2$ data from EDC during MIS 6.

| Ice core | Method (Reference) | $CO_2$ difference with $CO_2$ from EDC by ball mill in this study (ppm) | Contamination correction | Number of replicates | Number of sample |
|---|---|---|---|---|---|
| EDC | Sublimation at CEP Schneider et al. (2013) | 4.7± 1.7 (1σ) | O | 2–5 | 14 |
| | Ball mill at IGE Lourantou et al. (2010) | 2.4±2.1 (1σ) | X | 1 | 11 |
| | Ring mill at IGE (In this study) | 8.2±1.1 (1σ) | O | 1 | 11 |
| | Needle cracker at CEP (In this study) | 7.8± 1.1 (1σ) | O | 2–4 | 35 |
| | CIM at CEP (In this study) | 5.4± 1.0 (1σ) | O | 2–4 | 26 |
| Vostok | Ball mill at CEP Petit et al. (1999) | 4.6± 3.0 (1σ) | X | 1 | 49 |

$CO_2$ concentrations from EDC were reconstructed from 150 depth intervals that cover 2036.7 to 1787.5 m along the EDC ice core, which consist of clathrate ice. There exists true small scale variability in $CO_2$ concentrations in the ice below the Clathrate Zone (Lüthi et al., 2010). Due to the diffusion effect, this small variation of atmospheric $CO_2$ is smoothed. Thus, $CO_2$ concentrations in these depth intervals might represent the initial mean atmospheric

10    concentration. However, the EDC ice core for MIS 6 was drilled in 1999 and, the ice core has been stored for ~20 years in cold rooms at -22.5 ± 2.5°C before the gas is analysed. More than 50% of the initial hydrates present in the freshly drilled ice may have been decomposed and transformed into secondary bubbles, or gas cavities (Lipenkov, *Pers. Comm.*). We expect the same fractionation as during the clathrate formation process, hence bubble would be depleted in $CO_2$. Thus, $CO_2$ concentrations from EDC may be lower. In addition, different

15    analytical methods can cause $CO_2$ offsets.

In our study, we concentrate on the relative millennial changes of $CO_2$, which are confirmed by all of the EDC $CO_2$ records available so far. Thus, our conclusion in this paper are independent which absolute mean $CO_2$ level is correct. As the new data in this study are currently the best quality data in terms of repeatability, we use our new data as reference record and correct for any inter-core offsets (see Figure 2 and Figure 3).

20    In order to estimate these offsets while accurately accounting for both measurement uncertainty and uncertainty in the offsets themselves, we rely on a Monte Carlo procedure, which is run for 1000 iterations. At each iteration, the data from all datasets is resampled within its measurement uncertainty. Then, a Savitsky-Golay filter with an

approximate cutoff period of 150 years (using a 7-point sliding window and cubic fit, sampled at 250-year resolution) is applied to the new EDC data from this study. The offsets between each additional dataset and our data are calculated.

In order to test the sensitivity of the stack to the interpolation methods, Monte Carlo procedures were also run using linear interpolation, cubic spline filtering, and enting spline filtering in place of the Savitsky-Golay filter. The mean calculated offsets did not vary by more than 0.2 ppm depending on the method, well within the uncertainty ranges calculated for the offsets themselves. At the end of the stochastic procedure, mean and standard deviations of each offset are calculated, and used to adjust each dataset to create the composite.

[Figure]

**Figure 2:** Atmospheric $CO_2$ from EDC and Vostok ice cores, compared to the $\delta D$ of water at EDC (temperature proxy) during 190—135 kyr BP. Blue dots: Atmospheric $CO_2$ from EDC by ball mill system (this study). Yellow dots: Atmospheric $CO_2$ from EDC by ball mill system (Lourantou et al., 2010). Purple dots: Atmospheric $CO_2$ from EDC by ring mill system. Red equilateral triangles: Atmospheric $CO_2$ from EDC by needle cracker. Black inverted triangles: Atmospheric $CO_2$ from EDC by CIM. Green rhombuses: Atmospheric $CO_2$ from EDC by sublimation. Grey dots: Atmospheric $CO_2$ from the Vostok ice core (Petit et al., 1999). Grey line: $\delta D$ of water at EDC (Jouzel et al., 2007).

There are two main sources of uncertainty in the composite dataset, the measurement uncertainty of the data and the uncertainty of the offset itself. The offset uncertainty is not independent for each point--rather, since the offsets appear to be approximately constant, the offset uncertainty should apply to all points together (or at least present very high covariance). Therefore, these two sources of uncertainty are presented separately, and not aggregated.

We also use this procedure to estimate an offset between our data and the data measured on the Vostok ice core. However, this offset does appear to evolve over time, changing during late MIS 6. Additionally, uncertainties in

the alignment of the Vostok and EDC age scales over MIS 6 make it unclear if the variations in the two data series are indeed contemporaneous. We therefore do not include the Vostok data in the composite.

[Figure]

**Figure 3:** A composite $CO_2$ from EDC and Vostok ice cores, compared to the δD of water at EDC (temperature proxy) during 190―135 kyr BP.

5  The composite dataset confirms the millennial-scale variations shown in the data from this study (Figure 2 and Figure 3). Although none of the individual additional datasets is of high enough resolution to show millennial-scale variations with accuracy, when aligned to our data the new data follow the millennial-scale variations with very few outliers.

Finally, the uncertainty with respect to the absolute $CO_2$ value should be noted. The offsets between the multiple datasets are in large part likely due to differences in extraction efficiency between the measurement methods. The sublimation and ring mill systems have high extraction efficiency on clathrates, and should therefore present more unbiased baseline $CO_2$ values. However, since these datasets are as of now incomplete, we have aligned all datasets to the baseline absolute value of our ball mill dataset, and the absolute $CO_2$ values are reported within an uncertainty of ~5 ppm. We emphasize that the conclusions in this paper are only made with respect to relative values, and absolute values are only considered within their uncertainties.

As the new data set measured in this study provides the best record in terms of repeatability of the $CO_2$ measurements for the time interval of MIS 6, we use it as reference data set to homogenize all the individual $CO_2$ reconstructions from different cores. To this end we used a low-pass filtered version of new data from this study and calculated the residuals of each individual other $CO_2$ data set to this spline. To correct that data set, we used a constant offset that minimizes the root mean square error relative to this spline. Note that while this methods finds an optimum homogenization of the data sets given their scatter and potential cross-dating issues, it does not make a statement of the correct absolute level of the homogenized data set, as all data sets are equally likely to be correct in their absolute level. As we are only interested in the relative variations over MIS 6 in our study, this has no impact on our conclusions.

The information about the composite dataset is now given in section 3.1: Data compilation in the revised manuscript and 'A composite data set during MIS 6' in the SI. The composite dataset now replaces the ball mill dataset in all calculations and figures. We note that the composite dataset still supports our original conclusions about millennial scale variability.

It is now well known that bacteria living in the ice can and do produce CO2. The only question is, how much? So it is absolutely essential to replicate CO2 analyses on pieces of ice cut from the same depth (and therefore presumably the same age, and having been exposed to the same atmospheric gas concentrations).

$CO_2$ records can be contaminated by the in-situ production of $CO_2$ caused by carbonate-acid reactions and oxidation of organic molecules, which are mostly observed in Greenland ice cores. This is because of higher values of impurities such as $Ca^{2+}$, hydrogen peroxide $H_2O_2$ and formaldehyde HCHO in Greenland ice cores. These impurities can cause carbonate-acid reactions and the oxidation of organic carbon, leading to large scattering of atmospheric $CO_2$ data.

Thus to obtain less in situ $CO_2$ production in ice, a low carbonate concentration and $H_2O_2$ in an ice core are important. Luckily, Antarctic ice cores have relatively low concentrations of $H_2O_2$ and carbonates and have low temperature compared to Greenlandic ice cores, which reduces the risk of $CO_2$ contamination (Tschumi and Stauffer, 2000). It is estimated that the in-situ production of $CO_2$ for Antarctic ice cores is smaller than 1.5 ppm (Bereiter et al., 2009). Thus, in-situ production of $CO_2$ cannot be ruled out but the effect should not greatly impact our main observations. In contrast, the observed offsets (see comments above) can be explained by the combination of clathratization/relaxation processes and incomplete extraction efficiencies of the various methods used. Accordingly, we refrain from discussing a potential in situ production issue in our manuscript.

Therefore the authors must return to the laboratory and measure essentially another 150 pieces of ice, before this manuscript can be published in CP. The authors must also quote their value they have found for the pooled standard deviation.

I also did not notice any mention of the number of samples that were rejected (but perhaps I just missed it). The authors must mention this number clearly in the main text (not in the Supplement).

2 data points were identified for which experimental error could not be ruled out, so we did not include these 2 points in this study. Except for these two points, data was not rejected.

Another problem with the manuscript as it stands is the large amount of speculation in the discussion. This doesn't add to the value of the paper and can be mostly cut out, or clearly labelled as speculation in the text.

Due to the lack of existing proxy data with high temporal resolution and high precision and modelling studies, explanations of carbon cycle mechanisms during MIS 6 are limited. However, hypotheses of these mechanisms have been presented by previous studies, and the continued discussion of these hypotheses and how our new observations may redirect the discussion, even if the very limited amount of data means that this discussion is speculative in nature, is important. We hope that this discussion will be helpful for future studies, and have made sure, as suggested by the reviewer, to clearly label any speculative discussion in the text.

**$CO_2$ offsets:** The ball mill system has a different extraction efficiency depending on the presence of bubbles and/or clathrates in the ice sample, which may cause an accuracy for reconstructing absolute mean $CO_2$ level. When the air is extracted from an ice core sample where bubble and clathrates co-exist, different dry extraction methods with different extraction efficiencies on bubbly and clathrate ice may lead to biased $CO_2$ concentrations (Lüthi et al., 2010; Schaefer et al., 2011). During clathrate formation, the gas is partitioned into clathrates due to the different gas diffusivities and solubilities (Salamatin et al., 2001). $CO_2$ has consistently been observed to be depleted in bubbles and enriched in clathrates (Schaefer et al., 2011). Degassing from clathrates during extraction takes much longer than air release from bubbles; thus, if air from the clathrate ice is not extracted entirely, $CO_2$ measurement will be lower than the true value.

The ball mill shows extraction efficiencies of ~62% for bubbles and ~52% for clathrates on average (Schaefer et al., 2011). If the ball mill is used to reconstruct $CO_2$ in Bubble–Clathrate Transformation Zone (BTCZ), $CO_2$ concentrations can be biased. $CO_2$ concentrations from EDC were reconstructed from 150 depth intervals that cover 2036.7 to 1787.5 m along the EDC ice core, which consist of clathrate ice. There exists true small scale variability in $CO_2$ concentrations in the ice below the Clathrate Zone (Lüthi et al., 2010). Due to the diffusion effect, this small variation of atmospheric $CO_2$ is smoothed. Thus, $CO_2$ concentrations in these depth intervals might represent the initial mean atmospheric concentration. However, the EDC ice core for MIS 6 was drilled in 1999 and, the ice core has been stored for ~20 years in cold rooms at $-22.5 \pm 2.5°C$ before the gas is analysed. More than 50% of the initial hydrates present in the freshly drilled ice may have been decomposed and transformed into secondary bubbles, or gas cavities (Lipenkov, *Pers. Comm.*). We expect the same fractionation as during the

clathrate formation process, hence bubbles would be depleted in $CO_2$. Thus, $CO_2$ concentrations from EDC may be lower. The portion of the Vostok ice core covering MIS 6 is also clathrate ice, but it was drilled in 1998 and measured immediately (Petit et al., 1999), and less clathrates may have transformed into secondary bubbles. Thus $CO_2$ concentrations from Vostok during MIS 6 may be higher and potentially reflect the true atmospheric concentration more closely. In our study we concentrate on the relative millennial changes of $CO_2$ around the mean glacial concentration, which are the same in all the $CO_2$ records available so far, Thus, our conclusion in this paper are independent of which absolute mean $CO_2$ level is correct. As the new data in this study are currently the best quality data in terms of repeatability, we use our new data as the reference record and correct for any inter-core offsets. We, however, state explicitly in the text that the absolute mean $CO_2$ level during MIS6 is not known better than 5 ppm.

This offset does appear to evolve over time, changing during late MIS 6. Additionally, uncertainties in the alignment of the Vostok and EDC age scales over MIS 6 make it unclear if the variations in the two data series are indeed contemporaneous.

This is written to in the revised manuscript in Section 3.1: Data compilation and in the SI.

**Data verification:** Replicates account for differences between two ice samples at the same depth, making a better estimate of standard deviation of the final measurement but not necessarily of system precision itself. For example, Lüthi et al. (2010) show that there exists true small scale variability in $CO_2$ concentrations in the ice below the Bubble Clathrate Transition Zone, which could be accounted for by using replicates, especially for small sample sizes. Due to the diffusion effect, this small variation of atmospheric $CO_2$ is smoothed to some degree. In our study, large sample sizes (40g) of the ball mill system were used to reconstruct atmospheric $CO_2$, so a low-noise signal from the ice core is extracted (the smaller measurements used in other systems would be noisier in theory). The standard deviation of the measurement is estimated from the 5 injections, but system precision was calculated from blank measurements, which were performed after every 10 measurements accounting for the possible sources of $CO_2$ contamination with our analytical procedure.

To verify our new dataset, we made a composite data set using by aligning previous sets of measurements made over the MIS 6 period on the EDC ice core to our dataset. First, we compared to two existing $CO_2$ data sets and two new $CO_2$ data sets from EDC (Figure 1 and Table 1). There are two published $CO_2$ datasets for EDC during MIS6—the first measured by the ball mill system at IGE (Lourantou et al., 2010) and the second by the sublimation system at CEP (Schneider et al., 2013). We also compared unpublished atmospheric $CO_2$ measurements from EDC by a novel centrifugal ice microtome (CIM) system, a needle cracker and a ring mill system (Shin, 2019). All records are on the AICC2012 air age scale (Bazin et al., 2013). All data sets is corrected for the gravitational fractionation effect using the new $\delta^{15}N$ data in our study.

[Figure]

**Figure 1:** Atmospheric $CO_2$ from EDC and Vostok ice cores, compared to the δD of water at EDC (temperature proxy) during 190–135 kyr BP. Blue dots: Atmospheric $CO_2$ from EDC by ball mill system (this study). Yellow dots: Atmospheric $CO_2$ from EDC by ball mill system (Lourantou et al., 2010). Purple dots: Atmospheric $CO_2$ from EDC by ring mill system. Red equilateral triangles: Atmospheric $CO_2$ from EDC by needle cracker. Black inverted triangles: Atmospheric $CO_2$ from EDC by CIM. Green rhombuses: Atmospheric $CO_2$ from EDC by sublimation. Grey dots: Atmospheric $CO_2$ from the Vostok ice core (Petit et al., 1999). Grey line: δD of water at EDC (Jouzel et al., 2007).

Because of the limited amount of samples available, the data reconstructed by both ball mill and ring mill methods are single measurements from the depth interval. $CO_2$ records by CIM, needle cracker and the sublimation methods were reconstructed from 2–5 replicates from individual depth intervals. The error bars of data without replicate indicate that the standard deviation of five consecutive injections of the gas extracted from each sample into the gas chromatography (Lourantou et al., 2010; Petit et al., 1999). The error bars of data with replicate indicate the standard deviation of the mean of replicates from the same depth interval (Schneider et al., 2013). Figure 1 shows $CO_2$ concentrations measured by the ball mill system, the ring system, the sublimation, the CIM and the needle cracker. These $CO_2$ concentrations by the ball mill system (Lourantou et al., 2010), the ring system, the sublimation (Schneider et al., 2013), the CIM and the needle cracker are systematically higher than $CO_2$ concentrations measured by the ball mill system in our study (Table 1 and Figure 1). Atmospheric $CO_2$ during the MIS 6 period shows an offset between $CO_2$ data in this study and other $CO_2$ sets, which might be related with different analytical methods.

Where the additional datasets have enough resolution, the millennial-scale variations shown in our MIS 6 dataset are reproduced. Nevertheless, the measurements in the different datasets cannot be immediately aggregated because of offsets between their absolute $CO_2$ values. Offset residuals show that these offsets do not present any significant temporal evolution over MIS 6, but rather appear to be constant. In order to estimate these offsets while accurately accounting for both measurement uncertainty and uncertainty in the offsets themselves, we rely on a Monte Carlo procedure, which is run for 1000 iterations. At each iteration, the data from all datasets is resampled within its measurement uncertainty. Then, a Savitsky-Golay filter with an approximate cutoff period of 150 years (using a 7-point sliding window and cubic fit, sampled at 250-year resolution) is applied to the new EDC data

from this study. The offsets between each additional dataset and our data are calculated. At the end of the stochastic procedure, mean and standard deviations of each offset are calculated, and used to adjust each dataset to create the composite.

In order to test the sensitivity of the stack to the interpolation methods, Monte Carlo procedures were also run using linear interpolation, cubic spline filtering, and enting spline filtering in place of the Savitsky-Golay filter. The mean calculated offsets did not vary by more than 0.2 ppm depending on the method, well within the uncertainty ranges calculated for the offsets themselves.

[Figure]

**Figure 2:** Atmospheric $CO_2$ from EDC and Vostok ice cores, compared to the δD of water at EDC (temperature proxy) during 190—135 kyr BP. Blue dots: Atmospheric $CO_2$ from EDC by ball mill system (this study). Yellow dots: Atmospheric $CO_2$ from EDC by ball mill system (Lourantou et al., 2010). Purple dots: Atmospheric $CO_2$ from EDC by ring mill system. Red equilateral triangles: Atmospheric $CO_2$ from EDC by needle cracker. Black inverted triangles: Atmospheric $CO_2$ from EDC by CIM. Green rhombuses: Atmospheric $CO_2$ from EDC by sublimation. Grey dots: Atmospheric $CO_2$ from the Vostok ice core (Petit et al., 1999). Grey line: δD of water at EDC (Jouzel et al., 2007).

There are two main sources of uncertainty in the composite dataset--the measurement uncertainty of the data, and the uncertainty of the offset itself. The offset uncertainty is not independent for each point, but should rather have very high covariance, which we cannot account for exactly since there are no exact replicates, and we are limited to estimating a mean offset for each data set. Therefore, these two sources of uncertainty are presented separately, and not aggregated.

[Figure]

**Figure 3:** Atmospheric $CO_2$ from EDC and Vostok ice cores, compared to the δD of water at EDC (temperature proxy) during 190−135 kyr BP. Blue dots: Atmospheric $CO_2$ from EDC by ball mill system (this study). Yellow dots: Atmospheric $CO_2$ from EDC by ball mill system (Lourantou et al., 2010). Purple dots: Atmospheric $CO_2$ from EDC by ring mill system. Red equilateral triangles: Atmospheric $CO_2$ from EDC by needle cracker. Black inverted triangles: Atmospheric $CO_2$ from EDC by CIM. Green rhombuses: Atmospheric $CO_2$ from EDC by sublimation. Grey dots: Atmospheric $CO_2$ from the Vostok ice core (Petit et al., 1999). Grey line: δD of water at EDC (Jouzel et al., 2007).

We also use this procedure to estimate an offset between our data and the data measured on the Vostok ice core. However, this offset does appear to evolve over time, changing during late MIS 6. Additionally, uncertainties in the alignment of the Vostok and EDC age scales over MIS 6 make it unclear if the variations in the two data series are indeed contemporaneous. We therefore do not include the Vostok data in the composite.

The composite dataset confirms the millenial-scale variations shown in the data from this study (Figure 2). Although none of the individual additional datasets is of high enough resolution to show millenial-scale variations with accuracy, when aligned to our data the new data follow the millenial-scale variations with very few outliers.

Finally, the uncertainty with respect to the absolute $CO_2$ value should be noted. The offsets between the multiple datasets are in large part likely due to differences in extraction efficiency between the measurement methods. The sublimation and ring mill systems have high extraction efficiency on clathrates, and should therefore present more unbiased baseline $CO_2$ values. However, since these datasets are as of now incomplete, we have aligned all datasets to the baseline absolute value of our ball mill dataset, and the absolute $CO_2$ values are reported within an uncertainty of ~5 ppm. We emphasize that the conclusions in this paper are only made with respect to relative values, and absolute values are only considered within their uncertainties.

**Table 1:** Existing $CO_2$ data sets from EDC and Vostok ice core and new $CO_2$ data from EDC during MIS 6.

| Ice core | Method (Reference) | $CO_2$ difference with $CO_2$ from EDC by ball mill in this study (ppm) | Contamination correction | Number of replicates | Number of sample |
|---|---|---|---|---|---|
| EDC | Sublimation at CEP Schneider et al. (2013) | 4.7± 1.7 (1σ) | O | 2–5 | 14 |
| | Ball mill at IGE Lourantou et al. (2010) | 2.4±2.1 (1σ) | X | 1 | 11 |
| | Ring mill at IGE (In this study) | 8.2±1.1 (1σ) | O | 1 | 11 |
| | Needle cracker at CEP (In this study) | 7.8± 1.1 (1σ) | O | 2–4 | 35 |
| | CIM at CEP (In this study) | 5.4± 1.0 (1σ) | O | 2–4 | 26 |
| Vostok | Ball mill at CEP Petit et al. (1999) | 4.6± 3.0 (1σ) | X | 1 | 49 |

The information about the composite dataset is now given in section 3.1: Data compilation in the revised manuscript and 'A composite data set during MIS 6' in the SI. The composite dataset now replaces the ball mill dataset in all calculations and figures. We note that the composite dataset still supports our original conclusions about millenial scale variability.

The authors state that they "make use of contrasting boundary conditions during the last two glacial periods to gain insight into the co-occurring carbon cycle changes." They then note that those boundary conditions are only "slightly different", and in the end, never explain what those differences are or why they might be expected to matter. This undermines the rationale for proceeding with this study and should be much better explained in a revised manuscript.

P3L1-P3L11 revised. Please read P2L20-P2L35 .

2) It appears that a potentially significant conclusion of the manuscript derives from the observations associated with a single small millennial event. This hardly seems justified and should be bolstered either by theoretical arguments or indications of similar behavior in existing data from another time interval.

Many similar events are present in MIS3 (see Figure 7 in the main text), bolstering our conclusion. Please see Figure 7 in the revised manuscript.

The division and labeling of sub-events is neither referenced nor adequately described, much less explained. Such division is understandable and can be helpful, but only if clearly delineated and consistently applied. Are the divisions related to marine oxygen isotopes, and should they be, or to something else? What is the justification for 6c, 6d, and 6e, when there is no 6a or 6b?

We now number the Carbon Dioxide Maxima based on the sub-event numbering of Margari et al. (2010) and Gottschalk et al. (2020). The sub-events are based on the six isotopic maxima identified by these authors over MIS 6, which correspond with the CDMs (6i-6vi from oldest to youngest). We believe that this numbering system helps maintain greater consistency across studies concerning MIS 6.

All figures in the text, along with the text itself, have been revised to reflect this change.

Smaller points to be considered and addressed:

Page 2 line 15 – "opposite behaviour"

Revised

Page 2 line 17 – Do the authors really infer that CO2 changes are "in response" to temperature?

This is removed.

Page 7 line 8 - The half cycle between minimum and maximum values is not the definition of an inflection point, nor is it any point at all.

Revised to: "The midpoint of the stadial transitions in both $\delta^{18}O$ of planktonic foraminifera and tree pollen in MD01–2444 were used to identify the NH stadial stadial transitions."

Page 14-15 – Data do appear to be limited, although Helmke, 2003, Kandiano, 2003,

Obrochta 2014, Mokkeddem 2016, and Barker 2015 come to mind.

These datasets, while valuable for discussing MIS6, are too scattered to observe exact variations on millennial time scales and have larger age uncertainties with respect to EDC, making it complicated to use them to comment on the relationship between climate and prominent $CO_2$ variation during the early MIS 6 on millennial time scales.

Page 15, line 16 – "available"

Revised

Figure 3 – What are the "Six variations on millennial time scales: : :"?

Revised to: The durations of the six NA stadials during MIS 6.

Figure 5 – Golay should be capitalized in the legend.

Revised

---

## Referee Report (RR1)

In my first review, I asked the authors to state clearly in the main text the number of rejected data points.  This is important in the case of difficult measurements, like CO2 in ice cores, so that the reader has a clear understanding of any potential risk of data selection-induced bias.  The authors responded that two samples had been rejected for experimental reasons, and I thank them for that and indeed am reassured that there is virtually no risk of selection bias.  But I saw no place in the revised manuscript where this was stated.  The authors still need to put this in the main text.  A good place would be page 7, line 7, after the sentence "In total, the datasets contain 237 CO2 measurement points."

On page 5, line 12. the text states:  "Moreover, the convective zone was confirmed to be very thin during the last deglaciation by Parrenin et al. (2012).  Thus, we assume that h conv is negligible during MIS 6."

There is now new information on the timing of the onset of CO2 rise during the last deglaciation from the WAIS Divide ice core (Marcott et al., Nature, 2014).  This new timing information shows that CO2 began its *sustained* rise about 300 years after the abrupt warming inflection point that marks the onset of deglaciation.  The situation is admittedly confusing, because one earlier CO2 point appears to be high (above the LGM baseline), but later points are back down at the LGM baseline.  So it appears likely that this single CO2 point does NOT represent the onset of *sustained* CO2 rise.

These observations call into question the conclusion of Parrenin et al. (2013) that the onset of CO2 rise may have been synchronous with the onset of deglacial warming in Antarctica.  By logical extension, the Parrenin et al. (2012; 2013) convective-zone conclusion can no longer be true, that there was no convective zone at EDC during the last deglaciation.  The 300-year lag of CO2 behind temperature instead now requires a modest convective zone (~10 m), at least according to the d15N method used by Parrenin et al. (2012; 2013).

Indeed, if one adopts the stated age uncertainty estimates of Parrenin et al. in the timiing of CO2 onset relative to temperature onset, one must in any case state a *range* in implied convective zone thickness rather than a single value.  My understanding of Parrenin et al. (2013) is that the convective zone thickness is only implied to be zero if CO2 and temperature rose synchronously, and the stated uncertainties in timing (up to 500 years later CO2 rise) allowed a convective zone of up to 15 m or so,
if my memory is correct.

The use of d15N for age control in the present manuscript therefore should include an uncertainty corresponding to the rather limited knowledge of MIS 6 convective zone thickness (perhaps a range of 0 -15 m?).  This may require a small timescale expansion of the stated age uncertainty.

For this reason, I would recommend modifying the sentence in quotes above, to "Moreover, the convective zone was confirmed to be thin (<15 m) during the last deglaciation by Parrenin et al. (2012; 2013), when taking into account subsequently-published CO2 data from WAIS Divide (Marcott et al., 2014).  Thus, we assume that h conv is effectively negligible during MIS 6, with a range of +/ -…… m."

Technical corrections:

Page 9, line 6  "….firn column-induced smoothing…."
Page 9, line 7   there seems to be a missing period at the end of this sentence ("…short stadials")

Page 9, line 17  this sentence is quite long and difficult to read.  Perhaps you can simplify and cut one clause so that it reads: "Due to the lack of an MIS 6 temperature proxy in Greenland, and due to the difficulty of placing marine temperature proxies on a precise common chronology with the EDC ice core, in this work CH4 measurements on the EDC ice core were used as markers of rapid warming in the NH…."

Page 12, line 13 " Some studies (for example, Menviel et al. 2008) mention…"

Page 12, line 18 cut extra "during"

Page 12, line 24-25   "…low accumulation at EDC and its wider age distribution, the estimation…"

Page 13, line 3   "compared to"

Page 13 line 6  "..conducted during the MIS 6 period are needed."

---

## Author Response (AR2)

Dear editor and reviewers,

We are grateful for the detailed suggestions, and your continuous attention to this manuscript. Please find author reply to reviewer 3 and the revised manuscript (with point-by-point responses) attached below. We address the comments in blue.

All the best, on behalf of all co-authors,

Jinhwa Shin

Reviewer #3, re-review of Shinwa et al.

In my first review, I asked the authors to state clearly in the main text the number of rejected data points. This is important in the case of difficult measurements, like $CO_2$ in ice cores, so that the reader has a clear understanding of any potential risk of data selection-induced bias. The authors responded that two samples had been rejected for experimental reasons, and I thank them for that and indeed am reassured that there is virtually no risk of selection bias. But I saw no place in the revised manuscript where this was stated. The authors still need to put this in the main text. A good place would be page 7, line 7, after the sentence "In total, the datasets contain 237 $CO_2$ measurement points."

Added

On page 5, line 12. the text states: "Moreover, the convective zone was confirmed to be very thin during the last deglaciation by Parrenin et al. (2012). Thus, we assume that h conv is negligible during MIS 6." There is now new information on the timing of the onset of $CO_2$ rise during the last deglaciation from the WAIS Divide ice core (Marcott et al., Nature, 2014). This new timing information shows that $CO_2$ began its *sustained* rise about 300 years after the abrupt warming inflection point that marks the onset of deglaciation. The situation is admittedly confusing, because one earlier $CO_2$ point appears to be high (above the LGM baseline), but later points are back down at the LGM baseline. So it appears likely that this single $CO_2$ point does NOT represent the onset of *sustained* $CO_2$ rise. These observations call into question the conclusion of Parrenin et al. (2013) that the onset of $CO_2$ rise may have been synchronous with the onset of deglacial warming in Antarctica. By logical extension, the Parrenin et al. (2012; 2013) convective-zone conclusion can no longer be true, that there was no convective zone at EDC during the last deglaciation. The 300-year lag of $CO_2$ behind temperature instead now requires a modest convective zone (~10 m), at least according to the d15N method used by Parrenin et al. (2012; 2013). Indeed, if one adopts the stated age uncertainty estimates of Parrenin et al. in the timiing of $CO_2$ onset relative to temperature onset, one must in any case state a *range* in implied convective zone thickness rather than a single value. My understanding of Parrenin et al. (2013) is that the convective zone thickness is only implied to be zero if $CO_2$ and temperature rose synchronously, and the stated uncertainties in timing (up to 500 years later $CO_2$ rise) allowed a convective zone of up to 15 m or so, if my memory is correct. The use of d15N for age control in the present manuscript therefore should include an uncertainty corresponding to the rather limited knowledge of MIS 6 convective zone thickness (perhaps a range of 0 -15m?). This may require a small timescale expansion of the stated age uncertainty. For this reason, I would recommend modifying the sentence in quotes above, to "Moreover,

the convective zone was confirmed to be thin (<15 m) during the last deglaciation by Parrenin et al. (2012; 2013), when taking into account subsequently-published CO2 data from WAIS Divide (Marcott et al., 2014). Thus, we assume that h conv is effectively negligible during MIS 6, with a range of +/ -…… m."

5    We respectfully disagree with Reviewer 3's assertion that the assumption that the height of the convective zone was negligible at the onset of the last termination is contradicted by the WAIS Divide CO2 record. This assumption was verified by Parrenin et al. (2013) by comparing their d15N-based delta-depth estimate (which relies on the assumption) to 1) an estimate derived from synchronization in the air and ice phases to the EDML and TALDICE cores, and 2) an estimate based on the bipolar seesaw hypothesis. Neither of these two estimates required any

10   information about the convective zone height. Additionally, this verification did not depend in any way on the estimate of the lag between CO2 and Antarctic temperature, which has evolved with the detail provided by the higher-resolution WAIS Divide CO2 record (Chowdhry Beeman et al., 2019), but with no implications concerning the height of the convective zone at EDC.

15   **Technical corrections:**

Page 9, line 6 "….firn column-induced smoothing…."

Revised

Page 9, line 7 there seems to be a missing period at the end of this sentence ("…short stadials")

Revised

20   Page 9, line 17 this sentence is quite long and difficult to read. Perhaps you can simplify and cut one clause so that it reads: "Due to the lack of an MIS 6 temperature proxy in Greenland, and due to the difficulty of placing marine temperature proxies on a precise common chronology with the EDC ice core, in this work CH4 measurements on the EDC ice core were used as markers of rapid warming in the NH…."

Revised

25   Page 12, line 13 " Some studies (for example, Menviel et al. 2008) mention…"

Revised

Page 12, line 18 cut extra "during"

Revised

Page 12, line 24-25 "…low accumulation at EDC and its wider age distribution, the estimation…"

30   Revised

Page 13, line 3 "compared to"

Revised

Page 13 line 6 "..conducted during the MIS 6 period are needed."

Revised

35

Chowdhry Beeman, J., Gest, L., Parrenin, F., Raynaud, D., Fudge, T. J., Buizert, C., and Brook, E. J.: Antarctic temperature and CO$_2$: near-synchrony yet variable phasing during the last deglaciation, Climate of the 
[revised manuscript text omitted]